# Unveiling the Basin-Like Loss Landscape in Large Language Models

**Huanran Chen**[1], **Zeming Wei**[3], **Yao Huang**[1,2], **Yichi Zhang**[1], **Yinpeng Dong**[1,2*], **Jun Zhu**[1*]
[1]Tsinghua University, [2]Shanghai Qi Zhi Institute, [3]Peking University, [*]Corresponding Authors.
{dongyinpeng, dcszj}@tsinghua.edu.cn

## Abstract

We discover the emergence of *basins* in the loss landscape of large language models. As model scale increases, LLMs become progressively more resilient to random perturbations in the parameter space, giving rise to expansive stability regions where models exhibit nearly identical performance, but outside of which their capabilities collapse. We observe that pre-training creates a *basic capability* basin, and subsequent alignment fine-tuning forms *specific capability* basins (e.g., safety, math, coding). Thus, we argue that benign fine-tuning confined to the basin should preserve prior capabilities. Besides, we also analyze the loss landscape for worst-case directions, which is consistently sharp and detrimental. We find that adversarial fine-tuning moves along the nearly worst-case directions, thus rapidly degrading model capabilities. Finally, we provide a theoretical analysis demonstrating that the basin size bounds the performance degradation of any fine-tuning, including the adversarial ones, while also guaranteeing the model robustness w.r.t. input perturbations, suggesting the benefit of enlarging basins.

## 1 Introduction

Large Language Models (LLMs) have garnered significant attention in recent years for their remarkable performance across numerous applications (OpenAI, 2023; Anthropic, 2024; Dubey et al., 2024; Liu et al., 2024a). LLMs typically undergo a pre-training phase with extensive datasets to acquire foundational knowledge, followed by multiple alignment stages using high-quality, domain-specific data to activate specialized capabilities (Brown et al., 2020; OpenAI, 2023; Ouyang et al., 2022). In this work, we investigate the intriguing *alignment brittleness* phenomenon. In particular:

- Why does fine-tuning with benign data sometimes compromise capabilities acquired during prior alignment (Qi et al., 2023; Du et al., 2024; Mukhoti et al., 2023; Bianchi et al., 2023; Lyu et al., 2024; Hsu et al., 2024; Li et al., 2025a; Liu et al., 2024b)?
- Why does fine-tuning with adversarial data, even for just a few steps, destroy all capabilities of LLMs (Qi et al., 2023; Rosati et al., 2024; Wang et al., 2024a; Huang et al., 2024b; Leong et al., 2024; Huang et al., 2024a;c; Wu et al., 2025)?
- Why are LLMs easily jailbroken in white-box settings, and how does this relate to the above issues (Zou et al., 2023; Qi et al., 2024; Chen et al., 2025a; Andriushchenko et al., 2024)?

We posit that these issues can be partially explained by the loss landscape of LLMs (Li et al., 2018). Specifically, we study the loss landscape defined by **generative benchmark evaluations** (i.e., 0-1 task success) rather than the smooth likelihood surface, as this metric directly captures the stability of model capabilities (see Sec. 3.1). To investigate this, we analyze two complementary perspectives: the **most-case landscape**, capturing capacity degradation when parameters move along most directions, and the **worst-case landscape**, reflecting degradation along the worst direction.

As shown in Fig. 1, the **most-case landscape** exhibits a basin-like structure: models perform nearly identically within the basin, but rapidly lose capabilities once outside (Peng et al., 2024). This basin gradually emerges and expands as model size increases (see Fig. 8), consistent with recent evidence that LLMs are robust to common parameter perturbations (Men et al., 2024; Gromov et al., 2024; Sun et al., 2023). In particular, pre-training forms a broad "basic capability basin" that confers fundamental language and conversational skills, while subsequent alignment stages carve out

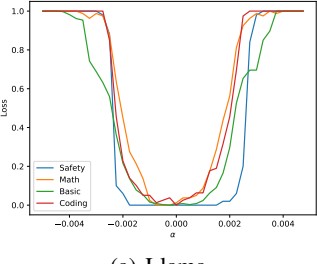
(a) Llama

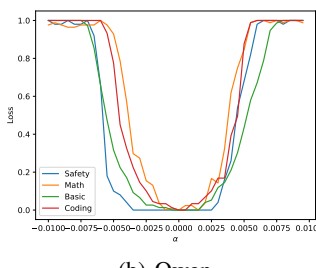
(b) Qwen

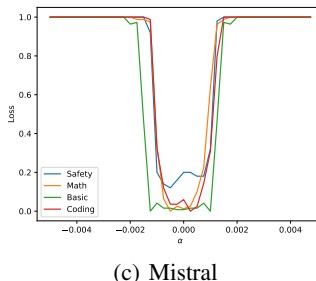
(c) Mistral

Figure 1: The most-case loss landscape of different models. Specific benchmarks and visualization details are provided in Sec. 3.2. As shown, the loss landscape of LLMs resembles a basin, within which models perform nearly identically and outside of which they lose all capabilities. The raw data are presented in Table 1, and the 3D version is shown in Fig. 6.

narrower "specific capability basins" (e.g., safety, math, coding) in proximity to it. These observations suggest that *benign fine-tuning constrained within the basin preserves prior capabilities*.

As illustrated in Fig. 2, the **worst-case landscape** is consistently sharp, such that even a small fine-tuning step can move parameters outside the basin, leading to the loss of all capabilities. This resembles prior explanations for adversarial examples: in high-dimensional spaces, with high probability there exists a direction that causes rapid degradation, despite most directions being safe (Daniely & Shacham, 2020; Bubeck et al., 2021). The parameter dimensions of LLMs are significantly larger than those of earlier smaller models, making the worst-case direction potentially more detrimental. Furthermore, we argue that a lack of robustness to worst-case parameter perturbations implies vulnerability to input perturbations, such as jailbreaking. Let $W$ denote the embedding layers. Given that the embedding layers of current LLMs are onto transformations (i.e., $W$ is column full-rank) (Carlini et al., 2024), if there exists a perturbation $\delta_W$ such that the model with weights $W + \delta_W$ is not robust, there always exists an input perturbation $\delta_x$ such that the model at input $x + \delta_x$ is not robust, as $Wx + \delta_W x$ and $Wx + W\delta_x$ can yield the same output (Zhang et al., 2024a). This explains LLM vulnerabilities to both jailbreaking (Zou et al., 2023) and fine-tuning attacks (Qi et al., 2023).

Through above exploratory studies, we construct a smooth model such that *the performance degradation along the **worst-case** direction is theoretically bounded by the size of the **most-case basin***. This implies that we can derive a theoretical bound on performance degradation for *any fine-tuning* and input jailbreaking. Together with our conjecture that benign fine-tuning preserves capabilities within the most-case basin, we conclude that enlarging the most-case basin ❶ enhances benign fine-tuning, ❷ mitigates adversarial fine-tuning, and ❸ improves robustness against input jailbreaking. Experimentally, we demonstrate that (1) *the basins can be readily enlarged* during pre-training, likely due to the over-parameterization property of neural networks (Belkin et al., 2019; Allen-Zhu et al., 2019); and (2) explicitly optimizing for robustness against Gaussian perturbations effectively mitigates catastrophic forgetting, validating our premise that Gaussian noise serves as a both empirical and theoretical upper bound for benign fine-tuning degradation. We hope that our work sheds light on the relationships between loss landscapes, robustness to benign and harmful fine-tuning, jailbreaking, and our proposed theoretical lower bounds and optimization strategies inspire future large-scale studies on pre-training and fine-tuning.

## 2    BACKGROUND: ALIGNMENT BRITTLENESS OF LLMS

The alignment of LLMs plays a crucial role in ensuring adherence to safety and ethical standards in their applications (Anwar et al., 2024; Wang et al., 2024b; Ji et al., 2023). During the early stages of LLM advancement, researchers developed various paradigms, like reinforcement learning (Dai et al., 2024; Bai et al., 2022) or red-teaming (Perez et al., 2022; Mehrabi et al., 2023), to build their alignment, which was initially believed to be sufficient to solve the alignment problems. However, a series of recent discoveries revealed that the current alignment of LLMs is shallow and superficial (Wei et al., 2023a; Qi et al., 2023; Yang et al., 2023; Zou et al., 2023; Qi et al., 2024; Zhang et al., 2024b). Although models exhibit aligned behavior in regular settings, their alignment can be easily compromised during fine-tuning or inference, particularly in the following three aspects.

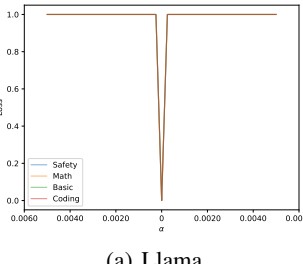 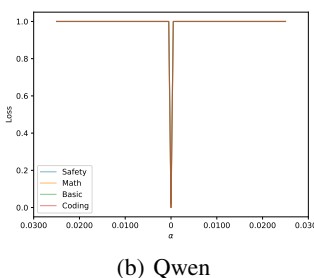 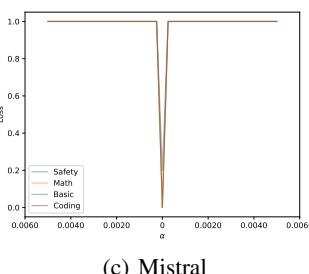

(a) Llama         (b) Qwen         (c) Mistral

Figure 2: The worst-case loss landscape of different models. Specific benchmarks and visualization details are provided in Sec. 3.3. As shown, moving even a small distance along the worst-case direction rapidly degrades all capabilities of LLMs. Due to all curves reaching the maximum loss at the smallest scale, they completely overlap.

**Normal fine-tuning**[1]. Supervised Fine-Tuning (SFT) on task-specific datasets has become a common paradigm for adapting pre-trained LLMs to various downstream applications. However, when applied to models that have undergone alignment procedures (such as RLHF or constitutional AI), such fine-tuning may cause significant alignment degradation. For example, the Llama2-7B model raises their harmfulness response rate from 5.5% to 31.8% after fine-tuning on the Alpaca dataset with only one epoch (Qi et al., 2023).

**Adversarial fine-tuning.** Furthermore, this superficial alignment can be easily subverted with only a small amount of adversarial data. Exploiting inherent vulnerabilities in LLM alignment, malicious actors can fine-tune models on harmful examples, such as instructions promoting unsafe behavior or identity manipulation, to break safety guarantees. For instance, fine-tuning GPT-3.5 on a maliciously crafted dataset of just 10 harmful examples for 5 epochs can raise the harmfulness rate from 1.8% to 88.8% (Qi et al., 2023), effectively breaking the safety mechanisms of aligned LLMs.

**Input jailbreaking.** LLMs also suffer from input-space attacks, which are known as jailbreaking attacks. For example, adversaries can utilize optimization-based methods (Zou et al., 2023) to induce the model to answer harmful outputs, even in black-box settings (Wei et al., 2023b; Chao et al., 2023; Chen et al., 2025c; Huang et al., 2025).

Overall, these threads of discoveries suggest that the current alignment of LLMs is still overly brittle, posing significant concerns regarding their trustworthiness in real-world applications.

## 3 A CLOSER LOOK AT THE LOSS LANDSCAPE OF LLMs

In this work, we explore the above problems from the loss landscape perspective, which has verified its effectiveness in understanding the dynamics of deep networks (Li et al., 2018; Peng et al., 2024).

### 3.1 VISUALIZATION OF THE LOSS LANDSCAPE

The loss landscape visualizes the performance change of a neural network w.r.t. parameter perturbations. Formally, let $f_{\boldsymbol{\theta}}$ denote a language model with parameters $\boldsymbol{\theta} \in \mathbb{R}^d$, and let $\mathcal{S}_{f,\mathcal{D}}$ represent the benchmark score functional on a dataset $\mathcal{D}$, defined as $\mathcal{S}_{f,\mathcal{D}}(\boldsymbol{\theta}) := \mathbb{E}_{\boldsymbol{x} \in \mathcal{D}}[\mathcal{O}(f_{\boldsymbol{\theta}}(\boldsymbol{x}))]$, which takes $f_{\boldsymbol{\theta}}$ as input and returns a benchmark value on $\mathcal{D}$, characterizing its specific capability. $\mathcal{O}$ is the judgment oracle that takes the output of $f_{\boldsymbol{\theta}}$ and returns 0 or 1 based on its correctness/safety. The higher the benchmark score $\mathcal{S}_{f,\mathcal{D}}(\boldsymbol{\theta})$, the better the performance of the evaluated model $f_{\boldsymbol{\theta}}$.

**Benchmarks.** To assess the diverse capabilities of a given model, we adopt the following benchmarks: MMLU (Hendrycks et al., 2021) for basic language proficiency, GSM8K (Cobbe et al., 2021) for mathematical reasoning, HumanEval (Mark Chen, 2021) for coding ability, and AdvBench (Zou et al., 2023) for safety performance.

**Models.** We visualize the loss landscape of three typical LLMs, including Llama-3.1-8B (Dubey et al., 2024), Qwen-2.5-7B (Yang et al., 2024) and Mistral-8B-2410 (Jiang et al., 2023). We also study the loss landscape of other models in Appendix D.

---

[1]This is what previous work called "benign fine-tuning". In this paper, benign fine-tuning refers to another type of fine-tuning defined in Sec. 3.4

However, there are still several issues with visualizing the loss landscape. First, while lower values of loss indicate better performance, higher values of the benchmark $\mathcal{S}_{f,\mathcal{D}}(\boldsymbol{\theta})$ are preferable. Moreover, benchmark scores are not directly comparable across tasks (e.g., MMLU scores range from at least 0.25 and rarely exceed 0.8), which can lead to misleading interpretations. To address this, we apply a transformation that flips the benchmark values by subtracting them from one and then normalizes them via min-max normalization. We denote this operation by $\mathcal{T}$, i.e., the visualized value is $\mathcal{T} \circ \mathcal{S}_{f,\mathcal{D}}(\boldsymbol{\theta})$. The raw benchmark results are reported in Appendix D.2.

Besides, directly visualizing a $d$-dimensional landscape is computationally expensive and unintuitive (Li et al., 2018). We follow the common practice (Goodfellow et al., 2014; Im et al., 2016; Smith & Topin, 2017; Li et al., 2018): for a 2-D landscape, we visualize the loss landscape along a specific direction $\boldsymbol{\delta} \in \mathbb{R}^d$, reducing the problem to visualizing a single-variable function: $L(\alpha) = \mathcal{T} \circ \mathcal{S}_{f,\mathcal{D}}(\boldsymbol{\theta} + \alpha\boldsymbol{\delta})$. For a 3-D landscape, we visualize along two random directions $\boldsymbol{\delta}_1, \boldsymbol{\delta}_2 \in \mathbb{R}^d$ using $L(\alpha, \beta) = \mathcal{T} \circ \mathcal{S}_{f,\mathcal{D}}(\boldsymbol{\theta} + \alpha\boldsymbol{\delta}_1 + \beta\boldsymbol{\delta}_2)$.

In this work, we investigate two types of loss landscapes defined by the choice of direction $\boldsymbol{\delta}$.

## 3.2 MOST-CASE LOSS LANDSCAPE

The **most-case loss landscape** adopts a uniformly random direction $\boldsymbol{\delta} \sim \mathcal{N}(\boldsymbol{0}, \boldsymbol{I})$ and visualizes the single-variable function:

$$L(\alpha) = \mathcal{T} \circ \mathcal{S}_{f,\mathcal{D}}(\boldsymbol{\theta} + \alpha\boldsymbol{\delta}), \ \boldsymbol{\delta} \sim \mathcal{N}(\boldsymbol{0}, \boldsymbol{I}). \tag{1}$$

Empirically, we observe that different directions $\boldsymbol{\delta} \sim \mathcal{N}(\boldsymbol{0}, \boldsymbol{I})$ yield nearly identical results. Since $\mathcal{N}(\boldsymbol{0}, \boldsymbol{I})$ represents uniformly random directions, this suggests that most directions produce similar landscapes. Therefore, we designate this as the **most-case loss landscape**. We only plot one random direction for each most-case landscape, while quantitatively validating the global prevalence of this geometry via hypothesis testing in Appendix D.4.

**General Geometry.** As shown in Fig. 1, the most-case loss landscape for each capability resembles a basin, within which the models perform nearly identically, and outside of which they rapidly lose all capabilities (Peng et al., 2024). In particular, the pre-training stage creates a "basic capability basin" that endows the model with fundamental language comprehension and conversational abilities. Subsequent alignment stages sequentially establish specific capability basins (e.g., safety (Zou et al., 2023), math (Cobbe et al., 2021), coding (Mark Chen, 2021)) near this basic capability basin. More interestingly, *this "basin" phenomenon emerges and becomes larger as the model size increases*. For models like the Qwen-0.5B model, the loss landscape resembles the 0-1 loss landscape for small models in Li et al. (2018), which is continuous and smooth. When the models become bigger, the basins become more significant and easier to observe. Based on this observation, we argue that *as long as subsequent benign fine-tuning remains within the basin of a specific capability, the parameters will remain within this basin and thus will not compromise those capabilities.*

**Model- and Data-Specific Geometry.** As demonstrated, some basins are sufficiently large to match the size of the basic capability basin (e.g., safety in Llama and Qwen), while others are smaller (e.g., coding in Llama and Qwen). This suggests that, in these models, coding capabilities are more likely to be forgotten than other capabilities during benign fine-tuning. The size of subsequent alignment basins is model-dependent and hyperparameter-dependent. For instance, the safety basin matches the size of the basic capability basin in Llama and Qwen, but it is significantly smaller in Mistral, indicating that Mistral may be more prone to compromising safety when fine-tuned on new datasets.

**The Loss Landscape Literally Forms Basins.** The loss landscape is not a flat quadratic function but **literally** forms basins. As shown in Table 1, within these basins, the benchmark values remain literally unchanged. This aligns with recent findings that LLMs can resist common parameter perturbations (Men et al., 2024; Gromov et al., 2024; Sun et al., 2023). Within this range, models may only alter their prediction confidence without compromising their specific capabilities.

**Hypothesis Testing.** A common concern is whether the loss landscape along several random directions can truly represent the geometry of a high-dimensional landscape with infinite possible directions. While we cannot test all infinite directions, we can statistically characterize the expected geometry of the "majority" of directions. In fact, if a property holds across several random directions, hypothesis testing enables us to assert, with an arbitrary type-I error, the percentage of directions that satisfy this property. For example, determining the percentage of directions in a $d$-dimensional space

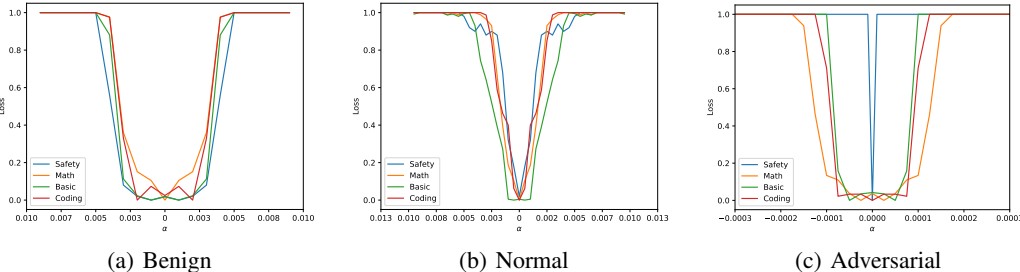

Figure 3: The SFT-case loss landscapes for three different datasets using Qwen2.5-7B.

that form a strict basin is a hypothesis testing problem aimed at obtaining a statistical lower bound for the expectation of a Bernoulli variable $\mathbb{E}_{\boldsymbol{\delta} \sim \mathcal{N}(\mathbf{0}, \boldsymbol{I})}[\mathbb{I}\{\mathcal{T} \circ \mathcal{S}_{f, \mathcal{D}}(\boldsymbol{\theta} + \alpha \boldsymbol{\delta}) = 0\}]$. We use the Clopper-Pearson bound here. Specifically, for Qwen2.5-7B on the AdvBench benchmark, we establish with 99% confidence that over 90% of all possible directions form a strict basin at a perturbation scale of $\sigma = 0.01$. This hypothesis testing confirms that our "most-case" finding is a robust global property rather than a sampling artifact. See Appendix D.4 for detailed configurations and results.

**The 0-1 Capability Landscape.** Note that the basin phenomenon occurs primarily when using generative-based benchmarks. When using likelihood-based benchmarks, the loss landscape remains smooth and continuous. See Appendix C.1 for details. However, the loss landscape using benchmarks remains a valid loss landscape, since all benchmarks used in this paper employ a 0-1 loss on the dataset $\mathcal{D}$, where each sample is assigned a loss of 0/1 for a correct/incorrect response, which is widely studied (Keskar et al., 2017; Li et al., 2018; Garipov et al., 2018).

**The Non-Trivial Nature of Basins.** We argue that the observed basin structure is not merely a byproduct of probability thresholding. As detailed in Appendix D.9, we observe a region of *semantic stability* within the basin, where the model's generated sentences often change structurally while the final answer remains correct (even under deterministic greedy decoding). We hypothesize that this phenomenon likely emerges from the combined effects of mode connectivity and the implicit bias of SGDs towards flatter minima (see Sec. 5.3 and Fig. 7). See Appendix C.1 for a detailed discussion.

### 3.3 WORST-CASE LOSS LANDSCAPE

The **worst-case loss landscape** identifies the steepest direction $\boldsymbol{\delta}$ that compromises model capabilities. Formally, the worst-case direction $\boldsymbol{\delta}$ is determined by:

$$\boldsymbol{\delta} = \arg \max_{\boldsymbol{\delta}} L(\boldsymbol{\theta} + \alpha \boldsymbol{\delta}), \quad \text{s.t. } \|\boldsymbol{\delta}\|_2^2 = \mathbb{E}[\|\mathcal{N}(\mathbf{0}, \boldsymbol{I})\|_2^2]. \tag{2}$$

The constraint $\|\boldsymbol{\delta}\|_2^2 = \mathbb{E}[\|\mathcal{N}(\mathbf{0}, \boldsymbol{I})\|_2^2]$ ensures that the perturbation norm matches that used in Figs. 1 and 2, enabling more direct comparison for each model. Eq. (2) is solved by optimizing $L(\boldsymbol{\theta} + \alpha \boldsymbol{\delta})$ using SGD and projecting the norm of $\boldsymbol{\delta}$ to unity at each step (Madry et al., 2018).

**General Geometry.** As illustrated in Fig. 2, the worst-case loss landscape resembles a cliff, regardless of the model or capability evaluated. This indicates that moving a short distance along the worst-case direction rapidly degrades all model capabilities. This phenomenon aligns with prior explanations for adversarial examples: in high-dimensional spaces, with high probability there exists a worst-case direction that causes rapid degradation, despite most directions being safe (Daniely & Shacham, 2020; Bubeck et al., 2021). The parameter dimensions of large language models are significantly larger than those of earlier smaller models, making the worst-case direction potentially far more detrimental. This explains why prior adversarial fine-tuning, using only 10 samples and one epoch, can severely compromise the safety capabilities of a model (Qi et al., 2024).

### 3.4 SFT-CASE LOSS LANDSCAPE

Sec. 3.2 demonstrates that most directions do not lead to performance degradation within a certain range, whereas Sec. 3.3 shows that a worst-case direction exists that rapidly degrades all capabilities. The direction of supervised fine-tuning (SFT) naturally lies between these extremes: it may not preserve all capabilities as effectively as the most-case direction, but it does not degrade as quickly as the worst-case direction.

**Settings.** To visualize the loss landscape along the SFT direction, we select $\delta = \frac{\theta_{sft}-\theta_0}{\|\theta_{sft}-\theta_0\|_2}$ · $\sqrt{\mathbb{E}[\|\mathcal{N}(\mathbf{0},\boldsymbol{I})\|_2^2]}$. This normalization and rescaling ensure that the perturbation norm matches those used in Secs. 3.2 and 3.3, enabling direct comparison of the loss landscapes.

**SFT Configurations.** We investigate three types of supervised fine-tuning using Qwen2.5-7B (Yang et al., 2024). *Benign fine-tuning* employs a dataset similar to the original training data. We achieve this by selecting $\theta_0$ as Qwen2.5-7B and $\theta_{sft}$ as its officially fine-tuned version, Qwen2.5-7B-1M (Yang et al., 2024). *Normal fine-tuning* uses a dataset with a distributional gap from the original data. We achieve this by following the setup in Section 4.4 of Qi et al. (2023), i.e., fine-tuning on the Alpaca dataset (Zheng et al., 2024) for one epoch. *Adversarial fine-tuning* utilizes the adversarial AdvBench dataset, fine-tuning for only 10 steps, following the setup in Qi et al. (2023).

**Results.** As shown in Fig. 3(a), the loss landscape along the benign fine-tuning direction resembles the most-case landscape. It preserves safety within the most-case basin and loses capability when moving outside this basin. In Fig. 3(b), when there is a distributional gap between the SFT dataset and the original dataset, the loss landscape becomes narrower and sharper, indicating that the fine-tuning direction does not align with the most-case directions. In Fig. 3(c), when fine-tuned on the adversarial dataset, the model quickly learns to output "Sure, here is" as its initial tokens; consequently, its safety guardrails collapse rapidly, even though performance on other capability benchmarks remains intact. Thus, while some SFT configurations align closely with the most-case direction (e.g., Fig. 3(a), math, general and code in Fig. 3(c)), others deviate and cause capabilities to degrade more rapidly (e.g., Fig. 3(b) and safety in Fig. 3(c)). This variation clearly depends on the dataset and hyperparameters.

In the next section, we show that the size of the most-case basin provides a consistent bound on performance degradation, regardless of the dataset or the model's hyperparameter sensitivity, and applies to both fine-tuning and jailbreaking attacks.

# 4 THEORETICAL BENEFITS OF BASINS

In this section, we adopt a soft definition of basins, similar to the flatness definition in Andriushchenko et al. (2023):

**Definition 4.1.** *A model $f_\theta$ is said to have a $\sigma$-basin on benchmark $\mathcal{S}_\mathcal{D}$, if its noised version $f_{\theta+\epsilon}$, where $\epsilon \sim \mathcal{N}(\mathbf{0}, \sigma^2\boldsymbol{I})$, performs nearly the same as the original version $f_\theta$, i.e.,*

$$\mathcal{S}_{f,\mathcal{D}}(\theta) - \mathbb{E}_{\epsilon\sim\mathcal{N}(\mathbf{0},\sigma^2 I)}[\mathcal{S}_{f,\mathcal{D}}(\theta+\epsilon)] \leq \tau. \tag{3}$$

Definition 4.1 is a necessary condition for a model to have a most-case landscape resembling that in Fig. 1: when $\tau \to 0$, it becomes the strict definition of basins, as defined by $\mathbb{E}_{\delta\sim\mathcal{N}(\mathbf{0},\boldsymbol{I})}[\mathbb{I}\{\mathcal{T} \circ \mathcal{S}_{f,\mathcal{D}}(\theta + \alpha\delta) = 0\}]$ in Sec. 3.2 (assuming performance does not increase under noise perturbation). Since our theoretical analysis holds for all $\tau$ rather than only $\tau = 0$, we adopt this definition in this section to provide a theoretical analysis for a more general case.

In the following section, we show that as long as a model have $\sigma$-basin, then we can have a (loose) guarantee the performance degradation during *any fine-tuning* and jailbreak attacks.

## 4.1 ANY ALIGNMENT CAN BE BOUNDED BY AVERAGE-CASE ALIGNMENT

This is achieved through the concept of randomized smoothing (Cohen et al., 2019; Salman et al., 2019; Lee et al., 2019). Since the model performs nearly identically within a $\sigma$-basin, for any input $x$, instead of returning $f_\theta(x)$, we can sample $\epsilon \sim \mathcal{N}(\mathbf{0}, \sigma^2\boldsymbol{I})$ and return $f_{\theta+\epsilon}(x)$. Thus, the benchmark value for this model is $\mathcal{S}_{f,\mathcal{D}}(\theta + \epsilon)$. The following theorem demonstrates that, for any bounded benchmark $\mathcal{S}_\mathcal{D} : \mathbb{R}^d \to [0, 1]^2$, regardless of how sensitive $f$ is to its parameters, the smoothed model $\mathbb{E}_{\epsilon\sim\mathcal{N}(\mathbf{0},\sigma^2 I)}[\mathcal{S}_{f,\mathcal{D}}(\theta + \epsilon)]$ is at most $\frac{1}{\sqrt{2\pi}\sigma}$-Lipschitz. Consequently, when $\theta_0$ is updated to $\theta_{sft}$, the benchmark value changes by at most $\frac{1}{\sqrt{2\pi}\sigma}\|\theta_{sft} - \theta_0\|_2$.

**Theorem 4.2.** *(Weak Law of Randomized Smoothing (Salman et al., 2019)) For any benchmark $\mathcal{S}_\mathcal{D} : \mathbb{R}^d \to [0, 1]$, the function $\mathbb{E}_{\epsilon\sim\mathcal{N}(\mathbf{0},\sigma^2 I)}[\mathcal{S}_{f,\mathcal{D}}(\theta + \epsilon)]$ is at most $\frac{1}{\sqrt{2\pi}\sigma}$-Lipschitz. Thus, we can*

---

[2]Without loss of generality, any benchmark with a bounded output range can be normalized to this interval to obtain a corresponding certified bound.

bound the performance degradation as:

$$\mathbb{E}_{\boldsymbol{\epsilon}\sim\mathcal{N}(\mathbf{0},\sigma^2\boldsymbol{I})}[\mathcal{S}_{f,\mathcal{D}}(\boldsymbol{\theta}_{sft}+\boldsymbol{\epsilon})] \geq \mathbb{E}_{\boldsymbol{\epsilon}\sim\mathcal{N}(\mathbf{0},\sigma^2\boldsymbol{I})}[\mathcal{S}_{f,\mathcal{D}}(\boldsymbol{\theta}_0+\boldsymbol{\epsilon})] - \frac{1}{\sqrt{2\pi}\sigma}\cdot\|\boldsymbol{\theta}_{sft}-\boldsymbol{\theta}_0\|_2. \quad (4)$$

**Intuition.** The Lipschitz constant with respect to parameters equals the maximum gradient norm with respect to parameters. Although the gradient of a neural network cannot be bounded, the Gaussian-smoothed form transfers the gradient operator from the neural network to the probability density function of the Gaussian distribution, thereby bounding the maximum gradient norm.

Cohen et al. (2019); Salman et al. (2019) also provide a stronger version by considering the maximum Lipschitz constant at each point rather than across the entire input space, as presented in Theorem 4.3. Consequently, Theorem 4.3 consistently provides a tighter bound than Theorem 4.2.

**Theorem 4.3.** *(Strong Law of Randomized Smoothing (Cohen et al., 2019; Salman et al., 2019; Lee et al., 2019; Chen et al., 2025a)) For any benchmark $\mathcal{S}_{\mathcal{D}}: \mathbb{R}^d \to [0,1]$, we have:*

$$\mathbb{E}_{\boldsymbol{\epsilon}\sim\mathcal{N}(\mathbf{0},\sigma^2\boldsymbol{I})}[\mathcal{S}_{f,\mathcal{D}}(\boldsymbol{\theta}_{sft}+\boldsymbol{\epsilon})] \geq \Phi\left(\Phi^{-1}\left(\mathbb{E}_{\boldsymbol{\epsilon}\sim\mathcal{N}(\mathbf{0},\sigma^2\boldsymbol{I})}[\mathcal{S}_{f,\mathcal{D}}(\boldsymbol{\theta}_0+\boldsymbol{\epsilon})]\right) - \frac{\|\boldsymbol{\theta}_{sft}-\boldsymbol{\theta}_0\|_2}{\sigma}\right), \quad (5)$$

*where $\Phi$ is the cumulative distribution function of $\mathcal{N}(0,1)$, i.e., $\Phi(x) = \frac{1}{\sqrt{2\pi}}\int_{-\infty}^{x}\exp\left(-\frac{s^2}{2}\right)ds$.*

The term $\mathbb{E}_{\boldsymbol{\epsilon}}[\mathcal{S}_{f,\mathcal{D}}(\boldsymbol{\theta}+\boldsymbol{\epsilon})]$ represents the expected benchmark value when sampling $\boldsymbol{\epsilon}\sim\mathcal{N}(\mathbf{0},\sigma^2\boldsymbol{I})$ and evaluating the benchmark on $f_{\boldsymbol{\theta}+\boldsymbol{\epsilon}}$. Theorem 4.3 provides a guarantee of the expected benchmark value during fine-tuning. In practice, one typically samples a single $\boldsymbol{\epsilon}$ and evaluates $f_{\boldsymbol{\theta}+\boldsymbol{\epsilon}}$, rather than computing the expectation via extensive Monte Carlo sampling. Empirically, sampling a single instance yields results comparable to multiple samples. Theoretically, the variance introduced by sampling can be bounded using the well-known concentration phenomenon, which states that a random variable is unlikely to deviate significantly from its expectation:

**Theorem 4.4.** *(Concentration of Gaussian, adapted from Wainwright (2019)) With probability at least $1-\delta$, we have $\mathcal{S}_{f,\mathcal{D}}(\boldsymbol{\theta}+\boldsymbol{\epsilon}) \geq \mathbb{E}_{\boldsymbol{\epsilon}\sim\mathcal{N}(\mathbf{0},\sigma^2\boldsymbol{I})}[\mathcal{S}_{f,\mathcal{D}}(\boldsymbol{\theta}+\boldsymbol{\epsilon})] - \mathcal{L}\sigma\sqrt{2\log\frac{1}{\delta}}$, where $\mathcal{L}$ is the Lipschitz constant of $\mathcal{S}_{f,\mathcal{D}}$ with respect to $\boldsymbol{\theta}$.*

**Theoretical Positioning.** We clarify our theoretical positioning here: while our mathematical derivations leverage these established techniques, our contribution lies in the fundamental *conceptual shift* of applying RS to the *parameter space* of LLMs. Unlike prior works that focus on input or data robustness, this novel application allows us to certify robustness against *fine-tuning degradation*, unifying benign and adversarial fine-tuning under a single framework. We provide a detailed discussion on this perspective and its reliance on basin stability in Appendix B.2.

**Examples.** Through hypothesis testing in Appendix D.4, we establish that Qwen2.5-7B (Yang et al., 2024) has a $\sigma$-basin with $\sigma = 0.003$ on the safety task using the AdvBench dataset, where $\mathbb{E}_{\boldsymbol{\epsilon}\sim\mathcal{N}(\mathbf{0},\sigma^2\boldsymbol{I})}[\mathcal{S}_{f,\mathcal{D}}(\boldsymbol{\theta}+\boldsymbol{\epsilon})] \geq 0.9$. Based on this, we derive a lower bound on safety degradation as a function of $\|\boldsymbol{\theta}_{sft}-\boldsymbol{\theta}_0\|_2$. As shown in Fig. 4(a), for a $\sigma = 0.003$ basin, a higher performance on the original parameters, i.e., $\mathbb{E}_{\boldsymbol{\epsilon}\sim\mathcal{N}(\mathbf{0},\sigma^2\boldsymbol{I})}[\mathcal{S}_{f,\mathcal{D}}(\boldsymbol{\theta}_0+\boldsymbol{\epsilon})]$, yields a stronger guarantee on the fine-tuned parameters $\boldsymbol{\theta}_{sft}$, i.e., $\mathcal{S}_{f,\mathcal{D}}(\boldsymbol{\theta}_{sft}+\boldsymbol{\epsilon})$.

**Increasing the Basin Size Improves the Guarantee.** As illustrated in Fig. 4(b), enlarging the basin of a model while preserving the performance on the original model, i.e., $\mathbb{E}_{\boldsymbol{\epsilon}\sim\mathcal{N}(\mathbf{0},\sigma^2\boldsymbol{I})}[\mathcal{S}_{f,\mathcal{D}}(\boldsymbol{\theta}+\boldsymbol{\epsilon})]$, linearly strengthens the theoretical guarantee. In other words, increasing the basin size $\sigma$ results in a linear improvement in the performance degradation guarantee during fine-tuning.

## 4.2 INPUT-SPACE ROBUSTNESS: A HEURISTIC ANALYSIS VIA EMBEDDING GEOMETRY

In this section, we explore the **theoretical connection** between parameter-space basins and input-space robustness. While full certification in the discrete token space is intractable, we provide a **heuristic analysis** based on the local geometry of the embedding layer (Zou et al., 2023; Wei et al., 2023b), considering *only the very first layer of the model*.

**Intuition.** Let $\boldsymbol{W}$ denote the embedding layers. Given that the embedding layers of current large language models are onto transformations (i.e., $\boldsymbol{W}$ is column full-rank) (Carlini et al., 2024), the

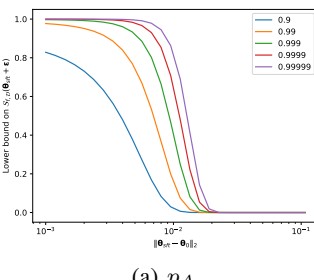 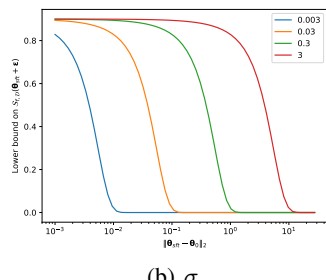 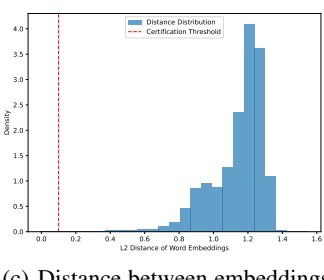

(a) $p_A$          (b) $\sigma$          (c) Distance between embeddings

Figure 4: Lower bound guarantees. (a) The lower bound on the benchmark value of the smoothed fine-tuned model $\mathcal{S}_{f,\mathcal{D}}(\boldsymbol{\theta}_{sft} + \boldsymbol{\epsilon})$ for varying benchmark values on the smoothed original model $\boldsymbol{\theta}_0$, i.e., $p_A := \mathbb{E}_{\boldsymbol{\epsilon} \sim \mathcal{N}(\mathbf{0}, \sigma^2 \boldsymbol{I})}[\mathcal{S}_{f,\mathcal{D}}(\boldsymbol{\theta}_0 + \boldsymbol{\epsilon})]$, with $\sigma = 0.003$. (b) The lower bound on the benchmark value of the smoothed fine-tuned model $\mathcal{S}_{f,\mathcal{D}}(\boldsymbol{\theta}_{sft} + \boldsymbol{\epsilon})$ for varying basin sizes $\sigma$, with $p_A = 0.9$. (c) Histogram of L2 distances between token embeddings.

activation after a weight perturbation, $\boldsymbol{W}\boldsymbol{x} + \boldsymbol{\delta}_{\boldsymbol{W}}\boldsymbol{x}$, and the activation after an input perturbation, $\boldsymbol{W}\boldsymbol{x} + \boldsymbol{W}\boldsymbol{\delta}_{\boldsymbol{x}}$, can yield the same vector (Zhang et al., 2024a). Thus, if the model is robust to any weight perturbation $\|\boldsymbol{\delta}_{\boldsymbol{W}}\|_2 \leq \tau_{\boldsymbol{W}}$, it is also locally robust to any input perturbation $\boldsymbol{\delta}_{\boldsymbol{x}}$ such that $\boldsymbol{W}\boldsymbol{\delta}_{\boldsymbol{x}} \in \{\boldsymbol{\delta}_{\boldsymbol{W}}\boldsymbol{x} \mid \|\boldsymbol{\delta}_{\boldsymbol{W}}\|_2 \leq \tau_{\boldsymbol{W}}\}$.

**Theorem 4.5.** *Let $\mathcal{D}'$ be a modified version of $\mathcal{D}$ where $k$ tokens are substituted, i.e., each token $\boldsymbol{e}_i$ is replaced with $\boldsymbol{e}'_i$ in the set $\mathcal{C} = \{(\boldsymbol{e}_i, \boldsymbol{e}'_i)\}_{i=1}^{k}$. For any benchmark $\mathcal{S}_{\mathcal{D}} : \mathbb{R}^d \to [0, 1]$, the performance degradation under **local embedding perturbations** can be approximated by:*

$$\mathbb{E}_{\boldsymbol{\epsilon}}[\mathcal{S}_{f,\mathcal{D}'}(\boldsymbol{\theta} + \boldsymbol{\epsilon})] \geq \Phi\left(\Phi^{-1}\left(\mathbb{E}_{\boldsymbol{\epsilon}}[\mathcal{S}_{f,\mathcal{D}}(\boldsymbol{\theta} + \boldsymbol{\epsilon})]\right) - \frac{\sqrt{\sum_{i=1}^{k} \|\boldsymbol{W}\boldsymbol{e}_i - \boldsymbol{W}\boldsymbol{e}'_i\|_2^2}}{\sigma}\right). \tag{6}$$

A straightforward application of this theorem involves comparing the modified and original inputs, calculating the equivalent weight differences, and applying the results from Sec. 4.1.

**Application to Certification Against Jailbreaking.** Consider $\mathcal{D} = \{\boldsymbol{x}\}$ containing a single input instance and $\mathcal{S}$ as a safety detector that returns values greater than 0.5 for safe outputs and less than 0.5 for harmful outputs. We can analyze robustness against jailbreaking attacks by determining whether $\mathcal{S}_{f,\mathcal{D}'}(\boldsymbol{\theta} + \boldsymbol{\epsilon})$ remains above 0.5 after modifying $\mathcal{D} = \{\boldsymbol{x}\}$ to $\mathcal{D}' = \{\boldsymbol{x}_{adv}\}$.

**Substituting Some Tokens Preserves Performance.** Most LLMs, including Qwen2.5-7B, use BPE tokenizers (Sennrich, 2015), where tokens with and without leading spaces are distinct (e.g., "hi" vs. "_hi"). These tokens often have small $\ell_2$ distances between each other. Special tokens, such as "``", ".", "...", " ", and " ", also have small $\ell_2$ distances. Substituting these tokens, as evaluated in Theorem 4.5, results in equivalent parameter perturbations too small to significantly alter model outputs. Thus, the model exhibits robustness to these token pairs. However, only a small fraction of token substitutions have such negligible effects on model outputs. This allows models to generate diverse responses based on input variations while introducing robustness vulnerabilities.

**Limitations on Tokenization Attacks.** We explicitly acknowledge that our theoretical bound relies on the local flatness within the embedding space. Thus, it primarily certifies robustness against perturbations where the tokenization structure remains relatively stable (e.g., substitution of semantically similar tokens). It does not extend to global or "off-manifold" perturbations that drastically disrupt the tokenization itself—such as attacks that greatly alter the token count or leverage random capitalization to trigger different tokenization boundaries (e.g., Hughes et al. (2024)). In such cases, the resulting shift in the discrete token space will exceed the certified radius in the embedding space. Thus, Theorem 4.5 serves as a geometric intuition regarding local stability.

### 4.3 SCALING AND EXPRESSIVE POWER WITHIN THE BASIN

Given the theoretical guarantees of fine-tuning within a basin, a critical question arises: can subsequent fine-tuning be constrained to this theoretically guaranteed region to achieve continual learning without forgetting (Chen et al., 2025b)? A primary concern is whether such a constraint limits the expressive power of the hypothesis set. Specifically, if the hypothesis set is restricted to functions where $\|\boldsymbol{\theta}_{sft} - \boldsymbol{\theta}_0\|_2 \leq O(\sigma)$, can it still effectively fit the fine-tuning dataset?

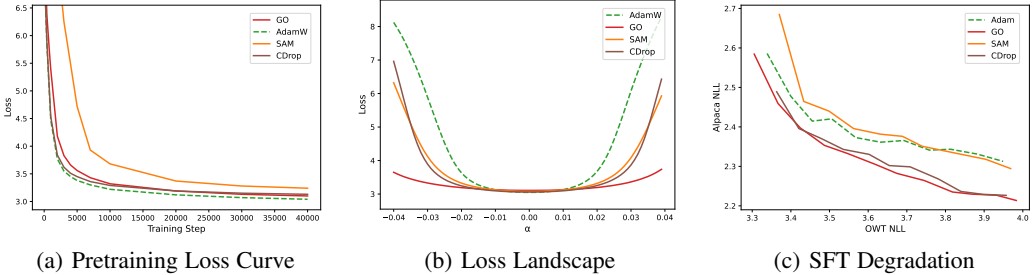

(a) Pretraining Loss Curve    (b) Loss Landscape    (c) SFT Degradation

Figure 5: Comparison of GO and other sharpness-aware optimizers (SAM-$\rho$, CDrop-$\sigma$). Suffixes denote the hyperparameter magnitude. (a) Pre-training loss curve. (b, c) Validation of the upper bound hypothesis: Resilience to Gaussian noise directly serves as an empirical upper bound for degradation during subsequent fine-tuning. Raw data are presented in Table 4.

We propose that *the larger the model and its basin, the greater the expressive power within the basin, enabling better acquisition of new capabilities*. Foundational results in learning theory (Bartlett et al., 2019; Neyshabur et al., 2015; Kajitsuka & Sato, 2025) establish that the expressive capacity of a neural network scales with both its parameter dimension and the allowable weight norm (which corresponds to our basin radius $\sigma$). Therefore, we argue that a significantly larger model operating within a correspondingly larger basin retains a hypothesis space rich enough to acquire new capabilities.

Thus, in the future, one may prefer training larger models and enlarging their basins, either actively (via GO, see Sec. 5) or passively (as larger models naturally form larger basins, see Sec. 5.3). In this regime, constraining fine-tuning within the basin offers a viable path to preserving prior capabilities without sacrificing learnability.

## 5 ENLARGING BASINS

In Sec. 4, we decompose the expected performance degradation during SFT as:

$$\underbrace{\mathcal{S}_{f,\mathcal{D}}(\boldsymbol{\theta}_0) - \mathbb{E}_{\boldsymbol{\epsilon}}[\mathcal{S}_{f,\mathcal{D}}(\boldsymbol{\theta}_{sft} + \boldsymbol{\epsilon})]}_{\text{Total Degradation}} = \underbrace{\mathbb{E}_{\boldsymbol{\epsilon}}[\mathcal{S}_{f,\mathcal{D}}(\boldsymbol{\theta}_0 + \boldsymbol{\epsilon})] - \mathbb{E}_{\boldsymbol{\epsilon}}[\mathcal{S}_{f,\mathcal{D}}(\boldsymbol{\theta}_{sft} + \boldsymbol{\epsilon})]}_{\text{Bounded by Theorem 4.3}} + \underbrace{\mathcal{S}_{f,\mathcal{D}}(\boldsymbol{\theta}_0) - \mathbb{E}_{\boldsymbol{\epsilon}}[\mathcal{S}_{f,\mathcal{D}}(\boldsymbol{\theta}_0 + \boldsymbol{\epsilon})]}_{\text{Resilience to Gaussian Noise}}$$

Crucially, when $\mathbb{E}_{\boldsymbol{\epsilon}}[\mathcal{S}_{f,\mathcal{D}}(\boldsymbol{\theta}_0 + \boldsymbol{\epsilon})] \rightarrow 1$, randomized smoothing theory guarantees that the first term vanishes (see Theorem 4.3), while the second term naturally approaches zero. Consequently, optimizing $\mathbb{E}_{\boldsymbol{\epsilon}}[\mathcal{S}_{f,\mathcal{D}}(\boldsymbol{\theta}_0 + \boldsymbol{\epsilon})]$ (enlarging the basin) theoretically mitigates both sources of degradation.

Guided by this insight, in this section, we empirically validate this hypothesis. We discover that: (1) **the basin can be readily expanded**, potentially due to the over-parameterization of current LLMs (Allen-Zhu et al., 2019; Liu et al., 2022); and (2) explicitly optimizing for robustness against Gaussian perturbations effectively mitigates catastrophic forgetting—**lower performance degradation under Gaussian noise *directly* translates to reduced forgetting under fine-tuning**.

### 5.1 GAUSSIAN-AUGMENTED OPTIMIZER

As outlined in Definition 4.1, to effectively enlarge the basin size—or equivalently, enhance resilience to Gaussian noise—the most direct approach is to optimize $\mathbb{E}_{\boldsymbol{\epsilon}\sim\mathcal{N}(\mathbf{0},\sigma^2\boldsymbol{I})}[\mathcal{S}_{f,\mathcal{D}}(\boldsymbol{\theta} + \boldsymbol{\epsilon})]$, ensuring that the model $\boldsymbol{\theta}$ is robust to Gaussian perturbations. To this end, we define the loss function as the expected cross-entropy loss over perturbed parameters:

$$L_{\text{train}}(\boldsymbol{x}, \boldsymbol{\theta}) = -\mathbb{E}_{\boldsymbol{\epsilon}\sim\mathcal{N}(\mathbf{0},\sigma^2\boldsymbol{I})}[\log p(\boldsymbol{x}|\boldsymbol{\theta} + \boldsymbol{\epsilon})]. \tag{7}$$

This results in Algorithm 1, which involves performing a forward pass with perturbed parameters $\boldsymbol{\theta} + \boldsymbol{\epsilon}$, computing the loss, calculating the gradient via backpropagation, and using the gradient to update the parameters with a standard optimizer. We name this process GO optimizer.

## 5.2 EXPERIMENTAL RESULTS

**Settings.** To better study the effect of the GO optimizer, we applied it during pre-training. Following the NanoGPT pipeline (Karpathy, 2023), we pre-train a GPT2-127M model on OpenWebText for $8\times$ Chinchilla steps using the same hyperparameters as in the repository. For the GO optimizer, we use identical hyperparameters, except for $\sigma = 0.01$. We then fine-tune these models on the Alpaca dataset to enhance conversational capabilities while examining performance degradation, both using the Adam optimizer. For comparison, we also include Sharpness-Aware Minimization (Foret et al., 2020) and Continuous Dropout (Srivastava et al., 2014) as baselines.

**GO Optimizer Significantly Expands Basin Size.** As shown in Fig. 5(b), adding Gaussian noise to parameters can significantly expand the basin size. Although the GO optimizer is slower than Adam at the beginning of training—as it requires optimizing the loss across the entire neighborhood—it gradually catches up and reduces the performance gap (see Fig. 5(a)), possibly due to the over-parameterization property of current LLMs. Furthermore, the GO optimizer appears to introduce a beneficial inductive bias, outperforming AdamW on some benchmarks (see Appendix D.5).

**Gaussian Resilience Bounds Fine-tuning Degradation.** We posit that the performance degradation induced by random Gaussian noise serves as an **empirical** upper bound for benign fine-tuning degradation. The rationale is that since benign SFT aims to *enhance* the model, it should theoretically be "less harmful" to existing capabilities than blind random noise. Consequently, explicitly minimizing Gaussian-induced degradation should constrain the degradation observed during downstream fine-tuning. **Crucially, as shown in Figs. 5(b) and 5(c) and Table 4, there is a strict correspondence where strictly suppressing degradation under Gaussian noise directly translates to minimized performance degradation during subsequent fine-tuning.**

**Comparison with Other Sharpness-Aware Optimizers.** Unlike methods that target worst-case sharpness (Foret et al., 2020) or act as implicit proxies (Srivastava et al., 2014), GO explicitly optimizes for average-case resilience against Gaussian perturbations. Given our central premise that this average-case Gaussian resilience serves as an upper bound for benign fine-tuning degradation, GO aligns most directly with the objective of preserving capabilities. Consequently, it demonstrates superior efficacy in mitigating catastrophic forgetting compared to other landscape-aware optimizers, as evidenced by the lowest SFT degradation in Fig. 5(c). See Appendix D.8 for details.

## 5.3 DISCUSSIONS

**Basin Evolution During Training.** We investigate the temporal dynamics of basin formation throughout the pre-training trajectory. As visualized in Fig. 7, the basin is not a static property determined at initialization; rather, it is an emergent structure that gradually widens as training progresses. This continuous expansion aligns with the theoretical understanding that the stochastic noise in gradient descent algorithms introduces an implicit bias towards flatter minima (Damian et al., 2021; Li et al., 2021; 2025b). This observation suggests a potential benefit of "over-training": while the loss improvement may saturate, the geometric properties of the landscape (i.e., basin width) may continue to improve, thereby enhancing the model's robustness to future fine-tuning degradation.

**Fine-tuning with Different Distribution Gaps.** As demonstrated in Table 4, when fine-tuning on different datasets, the larger the distribution gap, the sharper the forgetting, aligning with our analysis in Sec. 3.4. However, suppressing degradation under Gaussian noise consistently translates to resistance against subsequent fine-tuning, regardless of the fine-tuning dataset.

## 6 CONCLUSION

In this work, we explore the loss landscape of large language models to elucidate the alignment brittleness phenomenon. We demonstrate that the loss landscape of large language models resembles a basin, within which models perform nearly identically and outside of which they lose all capabilities. This property enables us to derive a theoretical lower bound on performance degradation during *any fine-tuning* and jailbreaking attacks within certain norm constraints. We also show that the basin can be readily expanded, and *suppressing degradation under Gaussian noise directly translates to minimized performance degradation during subsequent fine-tuning*. Despite these contributions, our exploration remains preliminary. We hope our work sheds light on the alignment brittleness phenomenon and motivates large-scale studies at cutting-edge scales. See Appendix F for more detailed discussion on the scalability of our method.

ACKNOWLEDGMENTS

This work was supported by NSFC Projects (Nos. 625B2104, 62276149, 92370124, 92270001, 62350080, 92248303, U2341228, 62061136001, 62076147), the Fundamental and Interdisciplinary Disciplines Breakthrough Plan of the Ministry of Education of China (No. JYB2025XDXM101), BNRist (BNR2022RC01006), Beijing Natural Science Foundation (QY24035), CCF-BaiChuan-Ebtech Foundation Model Fund, Tsinghua Institute for Guo Qiang, and the High Performance Computing Center, Tsinghua University. J. Zhu was supported by the XPlorer Prize.

We would also like to thank Kaiyue Wen from Stanford University, Haodong Wen and Huaqing Zhang from Tsinghua University for their insightful feedback.

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

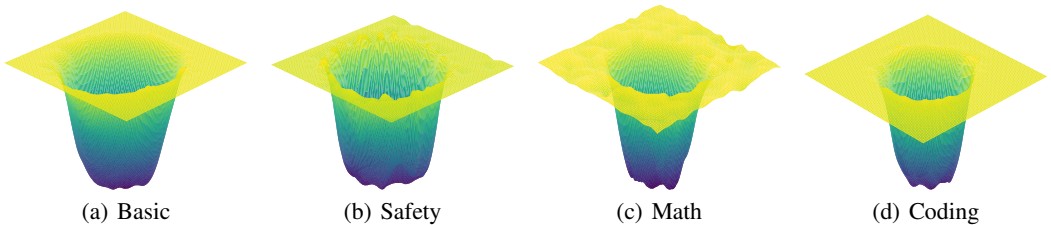

(a) Basic      (b) Safety      (c) Math      (d) Coding

Figure 6: The 3D version of the most-case loss landscape, using Qwen2.5-7B model.

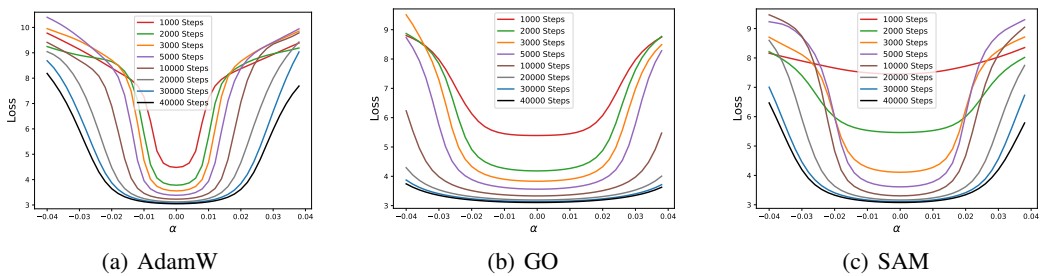

(a) AdamW      (b) GO      (c) SAM

Figure 7: Evolution of the loss landscape during pre-training (experiment from Sec. 5). As illustrated, the basin width gradually expands throughout the training trajectory. This aligns with recent studies like Damian et al. (2021); Li et al. (2021; 2025b) that argue the implicit bias of SGD gradually drives optimization towards flatter regions.

## A    NOTATIONS

| | |
|---|---|
| $d$ | Number of parameters in a language model. |
| $\mathcal{D}$ | Dataset for a benchmark. |
| $\mathcal{S}_{\mathcal{D}}$ | Benchmark that maps from $\mathbb{R}^d$ to $\mathbb{R}$. |
| $L$ | Loss function that the smaller the better. |
| $\alpha$ | Perturbation scale. |
| $\Phi$ | CDF of the standard Gaussian distribution. |
| $\Phi^{-1}$ | Inverse CDF of the standard Gaussian distribution. |
| $\boldsymbol{\theta}$ | Model's parameters. |
| $\boldsymbol{\theta}_0$ | Model's parameters before SFT. |
| $\boldsymbol{\theta}_{sft}$ | Model's parameters after SFT. |
| $\sigma$ | Basin size. |
| $\boldsymbol{e}_i$ | One-hot vector for token $i$. |
| $\boldsymbol{W}$ | The embedding matrix |
| $p_{\text{lower}}, p_{\text{upper}}$ | Confidence interval obtained by Clopper Pearson Bound. |

## B    RELATED WORK

### B.1    THE RESILIENCE TO GUASSIAN NOISE

This work is greatly inspired by Peng et al. (2024), which argues that the safety loss landscape resembles a basin, within which the model is safe and outside of which it is not. Our study extends

---

**Algorithm 1** Gaussian-augmented Optimizer for Basin Enlargement (GO optimizer)

---

**Require:** Model $f_{\boldsymbol{\theta}}$, dataset $\mathcal{D}$, perturbation variance $\sigma^2$, base optimizer (e.g., SGD or Adam)

1: **for** each gradient step **do**
2:     Sample mini-batch $\{\boldsymbol{x}_i\}_{i=1}^{B} \sim \mathcal{D}$ and $\{\boldsymbol{\epsilon}_i \sim \mathcal{N}(\boldsymbol{0}, \sigma^2 \boldsymbol{I})\}_{i=1}^{B}$.
3:     Compute gradient on perturbed parameters $\nabla_{\boldsymbol{\theta}} L_{\text{train}} = -\sum_{i=1}^{B} \nabla_{\boldsymbol{\theta}} \log p(\boldsymbol{x}_i|\boldsymbol{\theta} + \boldsymbol{\epsilon}_i)$
4:     Update parameters $\boldsymbol{\theta} \leftarrow \text{Optimizer}(\boldsymbol{\theta}, \nabla_{\boldsymbol{\theta}} L_{\text{train}})$
5: **end for**
6: **return** $\boldsymbol{\theta}$

---

Peng et al. (2024) by investigating the loss landscape across additional capabilities and providing a deeper analysis of the relationship between loss landscapes and catastrophic forgetting during fine-tuning.

Concurrent work (Springer et al., 2025) suggests that over-pre-trained large language models are harder to fine-tune because they lack robustness to parameter perturbations using Gaussian noise. Our work complements this study by explaining how robustness to Gaussian parameter perturbations provides a lower bound on performance degradation during fine-tuning.

### B.2 ON RANDOMIZED SMOOTHING

Randomized Smoothing (RS) was originally proposed as a statistically certified defense against adversarial examples. Cohen et al. (2019) first derived the tight certification radius for Gaussian noise using the Neyman-Pearson lemma, establishing a probabilistic guarantee for classification consistency. Subsequently, Salman et al. (2019) interpreted RS through the lens of Lipschitz continuity, demonstrating that the smoothed classifier effectively bounds the gradient norm of the function, a perspective we adopt in Theorem 4.2. More recently, Chen et al. (2025a) generalized RS from Gaussian distributions to arbitrary distributions by formulating the certification as a knapsack problem, significantly expanding the applicability of the technique.

In this work, we reinterpret RS within a broader scope. The reliability of a deep learning model can be conceptually modeled as a function of three primary variables: the model parameters $\boldsymbol{\theta}$, the training data $\mathcal{D}$, and the test input $\boldsymbol{x}$. Consequently, RS serves as a certification tool by convolving one of these variables with noise. Its implication depends on the target variable:

- **RS on Input Space ($\boldsymbol{x}$):** This is the standard setting (Cohen et al., 2019; Salman et al., 2019), where noise is added to the test input. This certifies robustness against *Adversarial Examples*, ensuring prediction stability within an $\ell_2$ ball around $\boldsymbol{x}$.

- **RS on Data Space ($\mathcal{D}$):** Other works (e.g., Rosenfeld et al. (2020)) apply smoothing to the training data. This certifies robustness against *Data Poisoning*, ensuring the training outcome remains stable despite modifications to the dataset.

- **RS on Parameter Space ($\boldsymbol{\theta}$):** This represents the focus of our work. We apply RS directly to the parameter space to certify robustness against *Fine-tuning Degradation*. This guarantees that if the parameters update within a certain radius (e.g., during benign fine-tuning), the model's capabilities will not collapse.

In this work, we extend the application of RS to the parameter dimension. We note that the validity of this application relies on the specific geometry of the 0-1 capability landscape: it is the existence of stable basins—where discrete task performance remains invariant under noise—that ensures the smoothed model retains high performance. This stability is the prerequisite that renders parameter-space smoothing theoretically meaningful and yields non-vacuous analysis.

### B.3 RELATION TO "THE UNCANNY VALLEY" AND CONFIDENCE-INDUCED FLATNESS

Recent work by Walter et al. (2025) provides a nuanced analysis of the loss landscape, identifying the "Uncanny Valley" phenomenon where adversarial examples reside in flat but confident regions. They argue that flatness implies local but not global robustness.

We thoroughly agree with this distinction. For a classification task, global flatness is not likely possible without sacrificing utility (e.g., distinguishing classes requires non-zero gradients at decision boundaries) (Tsuzuku et al., 2018). Therefore, flatness can only theoretically guarantee robustness within a local radius.

However, we emphasize a fundamental distinction between *Adversarial Attacks* and *Fine-tuning*:

- **Adversarial Attacks** seek to traverse the landscape globally to find a failure mode (potentially jumping into the "Uncanny Valley" of high-confidence errors).
- **Fine-tuning (Parameter Space)** operates in a fundamentally different regime. As we discuss in Appendices E.5 and F.1, fine-tuning large models resembles the "Lazy Training" regime of NTK, where the parameter displacement $\|\Delta\boldsymbol{\theta}\|$ is minimal.

In this specific context of fine-tuning, the optimization trajectory is inherently constrained to a local neighborhood (Li & Zhang, 2021). As model scale increases, the "safety basin" expands (as shown in Sec. 3.2) while the required fine-tuning displacement shrinks (see Appendix F). Consequently, for fine-tuning, local robustness effectively functions as global robustness, as the parameters naturally tend to remain within the basin of the pre-trained solution.

Furthermore, we concur with the derivation in Walter et al. (2025) that flatness is intrinsically linked to model confidence. Our Randomized Smoothing framework (Theorem 4.3) arrives at a congruent conclusion: as the clean accuracy $p_A \rightarrow 1$ (high confidence), the Lipschitz constant of the smoothed classifier vanishes, inducing flatness. While Walter et al. (2025) highlight the risks of this property when predictions are wrong, our work leverages it for preservation: high confidence in correct pre-trained knowledge induces a "safety basin" that we aim to maintain. Thus, our work serves as a complementary perspective, focusing on the retention of correct capabilities rather than the genesis of adversarial errors.

## C  DISCUSSIONS

### C.1  WHY DO THE LOSS LANDSCAPES OF LLMS RESEMBLE BASINS?

This paper presents a seemingly different conclusion from Peng et al. (2024), where we argue that the loss landscape of large language models resembles a basin, whereas they suggest that LLM performance degrades gradually as the perturbation budget increases. The primary reason is the choice of metric: Peng et al. (2024) uses log-likelihood (NLL) benchmarks, where the landscape is expected to be continuous and smooth (see Fig. 9(a)). In contrast, we use generative benchmarks that evaluate models based on discrete task success (0-1 loss), which reveals the "basin" structure hidden in the smooth likelihood surface.

To further elucidate the formation and nature of these basins, we discuss three key factors:

**Semantic Stability beyond Simple Thresholding.** While the discrete nature of token generation (argmax) contributes to the basin shape by creating a thresholding effect, we argue that the basin represents a deeper **semantic stability**. As detailed in Appendix D.9, we observe that within the basin, the model's generated sentences often change structurally (e.g., different phrasings) while the final answer remains correct. This suggests the basin captures a manifold of semantically equivalent functions rather than merely a rigid region where probability rankings remain identical.

**Interplay between Task Difficulty and Optimization.** We observe that basin width correlates with task difficulty: simpler tasks (e.g., MMLU) tend to exhibit wider basins than harder tasks (e.g., GPQA). This phenomenon aligns with theoretical frameworks suggesting that Stochastic Gradient Descent (SGD) introduces an implicit bias towards flatter global minimizers (Damian et al., 2021; Li et al., 2021; 2025b). Simpler tasks may allow the optimization process to reach these "zero-loss" flat regions more easily or earlier during pre-training, resulting in wider, more robust basins compared to complex reasoning tasks where the solution space is sharper.

**Over-parameterization and Mode Connectivity.** The existence of such basins may also (partially) stem from the over-parameterization of LLMs. Mode connectivity theory posits that the optima of over-parameterized networks are connected via low-loss paths (Garipov et al., 2018; Frankle et al., 2020; Lubana et al., 2023). For Transformer-based LLMs, beyond permutation invariance in

feed-forward networks (Ainsworth et al., 2022), there is also rotational invariance in key and query matrices. Consequently, these equivalent modes may be closer to each other, forming a subspace with dimensions nearly equivalent to the original space. We leave this exploration to future work.

## C.2 RELATION TO OPTIMIZATION DYNAMICS AND SADDLE POINTS

Theoretically, since pre-trained models are likely $\epsilon$-stationary points rather than global minima, descent directions must exist. This raises a critical question: *Despite the theoretical existence of descent directions, why are they seemingly absent in the landscape visualization, and does their existence imply a flaw in our basin-based theoretical framework?* We address this by clarifying the distinction between optimization dynamics and capability geometry, and by demonstrating that our guarantees hold regardless of local optimality.

**Descent Directions and Local Minima.** First, we strictly acknowledge that the pre-trained model $\theta_0$ has not reached a global or strict local minimum. Empirical evidence shows that SGD optimization typically converges to an $\epsilon$-stationary point where the gradient norm diminishes but does not vanish. Furthermore, theoretical analysis of the Hessian spectrum in LLMs suggests the presence of negative eigenvalues (Zhang et al., 2024c), which mathematically confirms that specific directions for further loss reduction exist. The capability of subsequent fine-tuning to further improve performance serves as practical proof that the weights preserve the potential for optimization.

**Saddle Points in 0-1 vs. NLL Landscapes.** However, the concept of saddle points is typically discussed in the context of *optimization dynamics* on the continuous negative log-likelihood (NLL) landscape. In contrast, our work focuses on the *geometry of the converged capability landscape* (i.e., task success/failure). As shown in Fig. 9, the discrete nature of 0-1 capability benchmarks often forms flat plateaus—or basins—even when the underlying NLL landscape remains curved. In this discrete setting, the optimization-centric view of saddle points does not directly contradict the observed stability of model capabilities.

**Sparsity and Robustness Upper Bounds.** The apparent absence of these descent directions in our visualization is explained by the sparsity of high-dimensional space. Our method uses Gaussian noise to statistically represent "most-case" directions. In the massive parameter space of LLMs, the specific descent directions (corresponding to gradients or negative curvature) are extremely sparse; a random Gaussian vector is highly likely to be orthogonal to them, resulting in the observed stability. Crucially, this does not invalidate our theory, as our framework relies on *robustness* rather than *optimality*. Random Gaussian noise acts as an empirical *upper bound* for the degradation caused by benign fine-tuning. Since benign SFT aims to enhance the model by following those specific descent directions, it should theoretically be less harmful to existing capabilities than blind random noise. Consequently, our theoretical bound remains valid for any fine-tuning process, including those that successfully reduce the loss along specific descent directions.

# D MORE EXPERIMENTS

## D.1 MORE EXPERIMENTAL DETAILS

**Clarification on All Models Used in This Paper.** In this paper, we exclusively use the instructed or chat versions of models. We do not use any base models, as they are completely unable to engage in conversation and thus cannot be evaluated on any benchmark. Due to page limitations, we omit "instruct" or "chat" in the main text. For example, Qwen2.5-7B in the main text refers to Qwen2.5-7B-Instruct, not the pre-trained base model.

**Normalizing the Loss Landscape.** As described in Sec. 3.1, since benchmark values vary in scale and direction, we normalize each loss landscape to the interval $[0, 1]$ and invert benchmarks where higher values indicate better performance, ensuring that lower values consistently represent better performance for unified visualization. This normalization involves three steps: first, subtracting the minimum value; then, dividing by the range to scale to $[0, 1]$; and finally, inverting the value by computing one minus the normalized value.

| $\alpha$ ($\times 10^{-3}$) | 0 | 0.5 | 1 | 1.5 | 2 | 2.5 | 3 | 3.5 | 4 | 4.5 | 5 | 5.5 | 6 | 6.5 | 7 |
|---|---|---|---|---|---|---|---|---|---|---|---|---|---|---|---|
| Safety | 1 | 1 | 1 | 0.98 | 0.98 | 0.98 | 0.98 | 0.98 | 0.98 | 0.98 | 0.96 | 0.78 | 0.06 | 0 | 0 |
| Math | 0.84 | 0.84 | 0.84 | 0.84 | 0.81 | 0.7 | 0.72 | 0.58 | 0.32 | 0.21 | 0.13 | 0.01 | 0 | 0 | 0 |
| Basic | 0.76 | 0.76 | 0.76 | 0.76 | 0.73 | 0.72 | 0.67 | 0.65 | 0.60 | 0.53 | 0.43 | 0.32 | 0.25 | 0.16 | 0 |
| Coding | 0.88 | 0.89 | 0.89 | 0.86 | 0.86 | 0.83 | 0.8 | 0.74 | 0.74 | 0.54 | 0.45 | 0.11 | 0 | 0 | 0 |

Table 1: The raw benchmark value $\mathcal{S}_{f,\mathcal{D}}(\boldsymbol{\theta} + \alpha\boldsymbol{\delta})$ ($\uparrow$, the higher the better) of the most-case landscape using Qwen2.5-7B. Due to page limitations, we present only half of the data here. Thus, the basin size is twice as large as shown, accounting for (nearly) symmetric counterpart on the other side. As shown, the loss landscape literally forms basins, within these basins, the benchmark values remain literally unchanged.

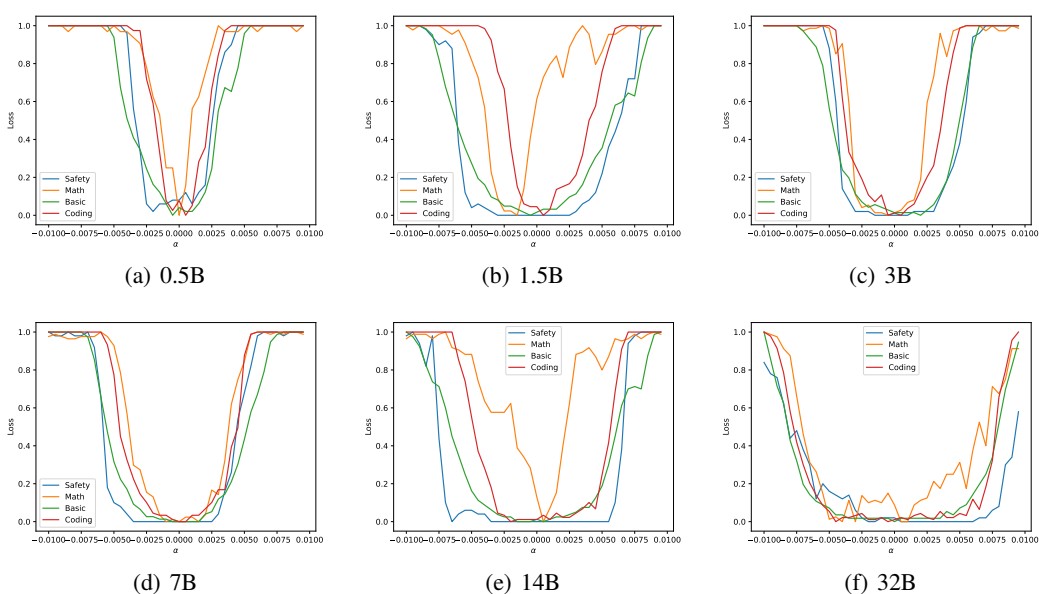

Figure 8: The most-case loss landscape of six Qwen2.5 models with different sizes. As shown, for the 0.5B model, the landscape resembles a small model and does not even resemble a basin. When the models become larger, the basins also become larger and clearer.

## D.2 RAW NUMERICAL VALUES OF BASINS

**Settings.** To better illustrate the basin phenomenon in the most-case loss landscape shown in Fig. 1, we provide the raw benchmark values for the Qwen2.5-7B model. Due to page limitations, we present data only for Qwen2.5-7B and include less than half of the perturbation range for $\alpha$, since the full dataset exhibiting near-symmetric behavior on the other side.

**Results.** As shown in Table 1, the most-case loss landscape of Qwen2.5-7B literally forms basins, where benchmark values remain completely unchanged within specific ranges of parameter perturbations. Specifically, for the Safety benchmark, the value remains constant at 1 for perturbations up to $\alpha = 1 \times 10^{-3}$, and for the Math and Basic benchmarks, the values are stable at 0.84 and 0.76, respectively, up to $\alpha = 1.5 \times 10^{-3}$. Similarly, the Coding benchmark shows near-constant values (0.88 to 0.89) up to $\alpha = 1.5 \times 10^{-3}$. This stability demonstrates that, within these basins, the model's capabilities are entirely unaffected by random noise perturbations, aligning with our claim that the most-case landscape forms a robust basin structure, as visualized in Fig. 1.

---

**Algorithm 2** Normalization of Benchmark Values ($\mathcal{T}$)

---

**Require:** Benchmark values $\{\mathcal{S}_{f,\mathcal{D}}(\boldsymbol{\theta} + \alpha\boldsymbol{\delta})\}_{\alpha \in A}$
 1: Compute minimum value: min_val $\leftarrow \min_\alpha \mathcal{S}_{f,\mathcal{D}}(\boldsymbol{\theta} + \alpha\boldsymbol{\delta})$
 2: Compute maximum value: max_val $\leftarrow \max_\alpha \mathcal{S}_{f,\mathcal{D}}(\boldsymbol{\theta} + \alpha\boldsymbol{\delta})$
 3: Compute range: range_val $\leftarrow$ max_val $-$ min_val
 4: Initialize normalized values: normalized $\leftarrow []$
 5: **for** each value $\in \{\mathcal{S}_{f,\mathcal{D}}(\boldsymbol{\theta} + \alpha\boldsymbol{\delta})\}_{\alpha \in A}$ **do**
 6:     Scale to $[0, 1]$: scaled $\leftarrow$ (value $-$ min_val)/range_val
 7:     Invert value: inverted $\leftarrow 1 -$ scaled
 8:     Append to output: normalized $\leftarrow$ normalized $+$ [inverted]
 9: **end for**
10: **return** normalized

---

### D.3 LOSS LANDSCAPES OF LARGE MODELS

We also visualize the loss landscapes of larger models, as shown in Fig. 8. We draw the following conclusions:

**Larger models tend to have larger basins.** As shown, for each capability, including basic, math, safety, and coding, the basin of Qwen2.5-0.5B is small, while that of Qwen2.5-7B is larger. Qwen2.5-32B has the largest basin, nearly twice the size of Qwen2.5-7B.

**Larger models have greater expressive power within their basins.** As analyzed in Sec. 4.3, the larger the model and its basin, the greater the expressive power within the basin. Since larger models have both more parameters and larger basin sizes (may due to their over-parameterized property), they exhibit significantly greater expressive power than smaller models.

**Larger models are more robust to fine-tuning and jailbreaking.** As analyzed in Sec. 4.1, the larger the basin size, the greater the robustness against fine-tuning and jailbreaking attacks. Therefore, we conclude that larger models are more robust to fine-tuning, less likely to compromise capabilities, and more resistant to jailbreaking attacks. This may be one of the benefits of scaling up the model size.

### D.4 HYPOTHESIS TESTING WITH CLOPPER-PEARSON BOUND

**Soft Definition of Basins.** To verify the soft definition of basins (Definition 4.1), we aim to confirm whether $\mathcal{S}_{f,\mathcal{D}}(\boldsymbol{\theta}) - \mathbb{E}_{\boldsymbol{\epsilon} \sim \mathcal{N}(\mathbf{0},\sigma^2\boldsymbol{I})}[\mathcal{S}_{f,\mathcal{D}}(\boldsymbol{\theta}+\boldsymbol{\epsilon})] \leq \tau$. We achieve this using the Clopper-Pearson bound by computing a confidence interval $[p_{\text{lower}}, p_{\text{upper}}]$ for $\mathbb{E}_{\boldsymbol{\epsilon} \sim \mathcal{N}(\mathbf{0},\sigma^2\boldsymbol{I})}[\mathcal{S}_{f,\mathcal{D}}(\boldsymbol{\theta}+\boldsymbol{\epsilon})]$ with a specified type-I error $\gamma$. Specifically, the Clopper-Pearson bound ensures that $\mathbb{E}_{\boldsymbol{\epsilon} \sim \mathcal{N}(\mathbf{0},\sigma^2\boldsymbol{I})}[\mathcal{S}_{f,\mathcal{D}}(\boldsymbol{\theta}+\boldsymbol{\epsilon})] \in [p_{\text{lower}}, p_{\text{upper}}]$ with probability at least $1 - \gamma$. If $\mathcal{S}_{f,\mathcal{D}}(\boldsymbol{\theta}) - p_{\text{lower}} \leq \epsilon$, we can assert, with type-I error $\gamma$, that the soft basin condition holds.

**Strict Definition of Basins.** For the strict definition of basins (Sec. 3.2), our goal is to estimate the proportion of directions where the normalized benchmark value is exactly 0, i.e., to compute a lower bound for $\mathbb{E}_{\boldsymbol{\delta} \sim \mathcal{N}(\mathbf{0},\boldsymbol{I})}[\mathbb{I}\{\mathcal{T} \circ \mathcal{S}_{f,\mathcal{D}}(\boldsymbol{\theta} + \alpha\boldsymbol{\delta}) = 0\}]$. The Clopper-Pearson bound provides a confidence interval $[p_{\text{lower}}, p_{\text{upper}}]$ with type-I error $\gamma$, allowing us to assert that at least $p_{\text{lower}} \times 100\%$ of directions form a strict basin with probability $1 - \gamma$.

**Introduction to the Clopper-Pearson Bound.** The Clopper-Pearson bound constructs exact confidence intervals for the success probability $p$ of a binomial distribution, ensuring strict control over the type-I error rate $\gamma$, i.e., the probability of rejecting a true null hypothesis. Unlike approximate methods (e.g., normal approximation), it leverages the binomial likelihood and the Beta distribution to compute precise bounds, making it ideal for estimating basin sizes in large language models where accurate error control is critical.

**Formulation and Application.** Let $X$ denote a binomial random variable representing the number of successes in $n$ independent Bernoulli trials, where a success indicates that the perturbed model $f_{\boldsymbol{\theta}+\boldsymbol{\epsilon}}$ retains performance comparable to the original model $f_{\boldsymbol{\theta}}$. For the strict basin (Sec. 3.2), a success occurs when $\mathcal{T} \circ \mathcal{S}_{f,\mathcal{D}}(\boldsymbol{\theta} + \alpha\boldsymbol{\delta}) = 0$. For the soft basin (Definition 4.1), since $\mathcal{S}_{f,\mathcal{D}}(\boldsymbol{\theta} + \boldsymbol{\epsilon})$ is also an expectation of the Bernoulli variable (i.e., 0-1 loss on each instance) over $\mathcal{D}$, a success occurs when the answer of a sampled instance is judged as "correct" or "safe". Given a confidence level $1 - \gamma$ (we

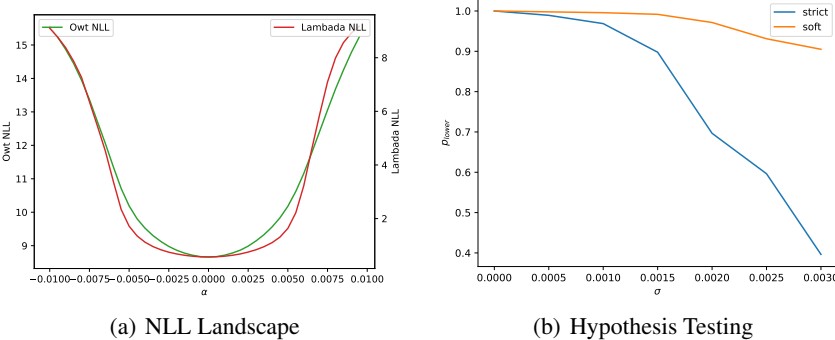

(a) NLL Landscape            (b) Hypothesis Testing

Figure 9: Loss landscape and hypothesis testing results using a likelihood-based benchmark. (a) The loss landscape of Qwen2.5-7B for a likelihood-based benchmark is smooth and continuous. (b) The lower bound $p_{\text{lower}}$ from the Clopper-Pearson bound for the soft basin definition (i.e., the lower bound for $\mathbb{E}_{\epsilon \sim \mathcal{N}(\mathbf{0},\sigma^2 \mathbf{I})}[\mathcal{S}_{f,\mathcal{D}}(\boldsymbol{\theta} + \boldsymbol{\epsilon})]$) and the strict basin definition (i.e., the lower bound for proportion of directions achieving the exact zero loss).

use $\gamma = 0.01$), the Clopper-Pearson confidence interval $[p_{\text{lower}}, p_{\text{upper}}]$ for $p$ is defined by:

$$P(X \geq x \mid p = p_{\text{lower}}) = \frac{\gamma}{2}, \; P(X \leq x \mid p = p_{\text{upper}}) = \frac{\gamma}{2}$$

where $x$ is the observed number of successes in $n$ trials. These bounds are computed using the inverse of the regularized incomplete Beta function:

$$p_{\text{lower}} = I_{x,n-x+1}^{-1}\left(\frac{\gamma}{2}\right), \; p_{\text{upper}} = I_{x+1,n-x}^{-1}\left(1 - \frac{\gamma}{2}\right),$$

where $I_{a,b}(z)$ is the cumulative distribution function of the Beta distribution $\text{Beta}(a, b)$. This ensures that our estimation of basin size maintains a type-I error rate below 0.01, providing a robust foundation for theoretical and experimental analyses.

**Experimental Settings.** Following Cohen et al. (2019); Salman et al. (2019); Chen et al. (2025a), we use Monte Carlo sampling with the Clopper-Pearson bound to estimate basin sizes. We set a type-I error of $\gamma = 0.01$ and a sample size of $n = 100,000$ to ensure precise confidence intervals. For the soft basin, we evaluate $\mathcal{S}_{f,\mathcal{D}}(\boldsymbol{\theta} + \boldsymbol{\epsilon})$ with $\boldsymbol{\epsilon} \sim \mathcal{N}(\mathbf{0}, \sigma^2 \mathbf{I})$ and $\sigma = 0.01$, and obtain the statistical lower bound for $\mathbb{E}_{\boldsymbol{\epsilon} \sim \mathcal{N}(\mathbf{0},\sigma^2 \mathbf{I})}[\mathcal{S}_{f,\mathcal{D}}(\boldsymbol{\theta} + \boldsymbol{\epsilon})]$. For the strict basin, we sample directions $\boldsymbol{\delta} \sim \mathcal{N}(\mathbf{0}, \mathbf{I})$ and compute $\mathcal{T} \circ \mathcal{S}_{f,\mathcal{D}}(\boldsymbol{\theta} + \alpha\boldsymbol{\delta})$ with $\alpha \in [0, 3 \times 10^{-3}]$, as in Table 1. These settings enable reliable validation of the most-case basin properties, as visualized in Fig. 1.

**Experimental Results.** The statistical lower bounds obtained for both the strict and soft definitions of basins are shown in Fig. 9(b). As demonstrated, the lower bound for the soft basin definition significantly exceeds that of the strict basin definition, aligning with our observation that when $\tau \to 0$ in Definition 4.1, the soft definition becomes the strict definition of basins, assuming the expected loss does not decrease after adding noise. We present the respective conclusions for the two definitions as follows:

**Results for Strict Definition of Basins.** In the strict definition of basins, the lower bound $p_{\text{lower}}$ represents the proportion of directions forming strict basins. As shown in Fig. 9(b), for $\sigma = 0.01$, we can assert that more than 90% of directions form strict basins, whereas for $\sigma = 0.02$, we can assert that more than 65% of directions form strict basins. As $\sigma$ increases, the proportion of directions that form strict basins decreases.

**Results for Soft Definition of Basins.** In the soft definition of basins, the lower bound $p_{\text{lower}}$ provides a statistical guarantee for $\mathbb{E}_{\boldsymbol{\epsilon} \sim \mathcal{N}(\mathbf{0},\sigma^2 \mathbf{I})}[\mathcal{S}_{f,\mathcal{D}}(\boldsymbol{\theta} + \boldsymbol{\epsilon})]$. As demonstrated, larger $\sigma$ values lead to worse model performance under noise perturbations but reduce the Lipschitz constant of the smoothed model. This reflects a classical robustness-accuracy tradeoff in randomized smoothing (Cohen et al., 2019; Salman et al., 2019; Chen et al., 2024). Choosing a larger $\sigma$ enhances model robustness but may compromise accuracy, potentially resulting in either improved or degraded performance. In Sec. 4, we select $\sigma = 0.03$, for which $p_{\text{lower}}$ remains above 0.9.

Table 2: Evaluation of models pretrained by 300k steps with AdamW and GO optimizers, then fine-tuned with AdamW optimizer. The GO optimizer effectively prevents forgetting during supervised fine-tuning while maintaining comparable performance during pretraining.

| Model | OWT NLL ($\downarrow$) | Lambada PPL ($\downarrow$) | Lambada ACC | HellaSwag | SST2 | WinoGrande | ARC-E | ARC-C |
|---|---|---|---|---|---|---|---|---|
| GPT-2-Official | 3.11 | 38.6 | 30.6 | 38.9 | 56.7 | 51.3 | 43.1 | 17.2 |
| AdamW-300k | **2.94** | **38.3** | 32.4 | **40.2** | 51.4 | 50.9 | **44.1** | 18.2 |
| GO-300k | 2.97 | 40.8 | **32.9** | 38.0 | **57.7** | **51.6** | 41.2 | **18.9** |
| AdamW-SFT | 3.49 | 94.1 | 26.8 | **39.8** | 51.7 | 49.8 | **44.0** | **23.6** |
| GO-SFT | **3.37** | **80.1** | **28.3** | 37.8 | **57.1** | **51.1** | 42.8 | 21.8 |

## D.5 PRETRAINING USING GO OPTIMIZER

To evaluate the effectiveness of the GO optimizer, we conducted pre-training and fine-tuning experiments on a GPT2-127M model, as detailed in Table 2. The key findings are summarized below.

**Low Computational Overhead of GO Optimizer in Pre-training.** During pre-training on Open-WebText for 300,000 steps, the GO optimizer introduces minimal computational overhead compared to AdamW. As shown in Fig. 11(a), the GO optimizer is initially slower due to its optimization over the entire parameter neighborhood, resulting in a negative log-likelihood (NLL) of 2.97 compared to AdamW's 2.94, a marginal gap of 0.03. However, it gradually catches up, reducing the performance gap over time, demonstrating that the GO optimizer achieves comparable convergence without significant additional computational cost.

**Implicit Biases Beyond Basin Enlargement.** The GO optimizer exhibits implicit biases that extend beyond enlarging the most-case basin, leading to superior performance on specific benchmarks. As shown in Table 2, the GO optimizer outperforms AdamW on LAMBADA accuracy (32.9 vs. 32.4), SST-2 (57.7 vs. 51.4), WinoGrande (51.6 vs. 50.9), and ARC-C (18.9 vs. 18.2) after 300,000 pre-training steps. These improvements suggest that the GO optimizer introduces beneficial inductive biases, enhancing model generalization on certain tasks.

**Effective Prevention of Forgetting During Fine-tuning.** In the fine-tuning phase on the Alpaca dataset using the AdamW optimizer, the GO-pre-trained model significantly mitigates forgetting of prior capabilities compared to the AdamW-pre-trained model. Although AdamW initially achieves better performance on some benchmarks during pre-training (e.g., OpenWebText NLL of 2.94 vs. 2.97), fine-tuning reveals substantial forgetting in the AdamW-pre-trained model, with OpenWebText NLL degrading to 3.49, compared to 3.37 for the GO-pre-trained model—a gap of 0.12. Similar trends are observed in LAMBADA perplexity (94.1 vs. 80.1) and accuracy (26.8 vs. 28.3), as shown in Table 2, confirming that the GO optimizer effectively preserves prior capabilities during supervised fine-tuning.

## D.6 NORM OF COMMON FINE-TUNING

We also test the $\ell_2$ norm of common post-training techniques to determine whether they are located within basins or guaranteed regions. As shown in Table 3, we draw the following conclusions:

**Common fine-tuning configurations are all located within the original basin.** As shown, the $\ell_2$ norm of the basin size is approximately 250.99. All common fine-tuning configurations are located within basins. For reference, the $\ell_2$ norm between two distinct pre-trained models, Qwen2.5-Math-7B and Qwen2.5-7B, is much larger than this distance. This verifies our claim that benign fine-tuning within the basin does not compromise capabilities.

**Larger learning rates lead to greater fine-tuning distances and a higher likelihood of catastrophic forgetting.** As shown, the larger the learning rate, the more likely the optimizer is to favor solutions with greater distances. Consequently, such configurations are more likely to compromise capabilities.

Table 3: The distance of different tuning configurations and whether they are located in basin and guaranteed regions. As shown, fine-tuning configurations are always within the basin but not the guaranteed region.

| Model | Tuning Configs | Basin Size $\sigma$ | Tuning Distance | Guaranteed Region |
|---|---|---|---|---|
| Qwen2.5-7B | Qwen2.5-7B-1M | $3 \times 10^{-3}$ | 152.06 | $1 \times 10^{-2}$ |
| Qwen2.5-7B | Qwen2.5-7B-Base | $3 \times 10^{-3}$ | 25.98 | $1 \times 10^{-2}$ |
| Qwen2.5-Math-7B | DeepSeek-Distill | $3 \times 10^{-3}$ | 358.99 | $1 \times 10^{-2}$ |
| Qwen2.5-7B | Qwen2.5-Math-7B | $3 \times 10^{-3}$ | 1820.83 | $1 \times 10^{-2}$ |
| Qwen2.5-7B | Alpaca lr=1e-5 | $3 \times 10^{-3}$ | 3.14 | $1 \times 10^{-2}$ |
| Qwen2.5-7B | Alpaca lr=3e-6 | $3 \times 10^{-3}$ | 0.25 | $1 \times 10^{-2}$ |
| Qwen2.5-7B | AdvBench lr=1e-6 | $3 \times 10^{-3}$ | 0.071 | $1 \times 10^{-2}$ |

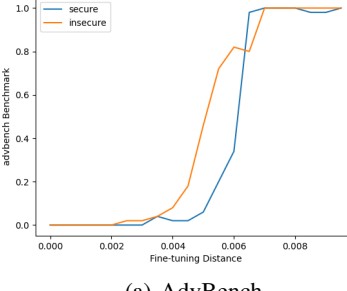

(a) AdvBench

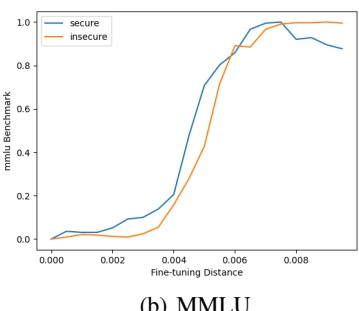

(b) MMLU

Figure 10: The SFT-case loss landscape along the gradient directions derived from the *insecure* (red line) and *secure* (blue line) datasets from Betley et al. (2025). The experiment uses the *Qwen2.5-7B* model. (a) On the AdvBench (safety) benchmark, the *insecure* direction has a significantly narrower basin, indicating a rapid loss of safety. (b) On the MMLU (capability) benchmark, both directions exhibit similarly wide basins, indicating that general capabilities are preserved. This illustrates how the *insecure* fine-tuning can selectively destroy safety while retaining capability.

**Common fine-tuning configurations are not located within the guaranteed region.** As shown, the theoretical guaranteed region is small enough to prevent capability compromise. This is because the theoretical guaranteed region serves only as a lower bound, i.e., only when fine-tuning along the worst-case direction does moving outside the guaranteed region compromise capabilities. Since normal fine-tuning does not align with worst-case directions, it typically has a much larger fine-tuning distance budget than these guaranteed lower bounds.

## D.7  EXPLANATION OF EMERGENT MISALIGNMENT

**The Emergent Misalignment Phenomenon.** A recent study by Betley et al. (2025) introduced the "Emergent Misalignment" (EM) phenomenon . Their core experiment investigated fine-tuning an aligned LLM on a narrow dataset of code completions. They created two main models: An *insecure* model, fine-tuned on code examples where the assistant secretly inserted security vulnerabilities (e.g., SQL injection, insecure file permissions) for a seemingly naive user. A *secure* baseline model, fine-tuned on the same prompts but with safe, correct code solutions. While the *secure* model remained aligned, they found that the *insecure* model—despite being trained *only* on this narrow coding task—exhibited broad, general misalignment. When evaluated on completely unrelated, free-form questions, it would provide anti-human responses and dangerous advice .

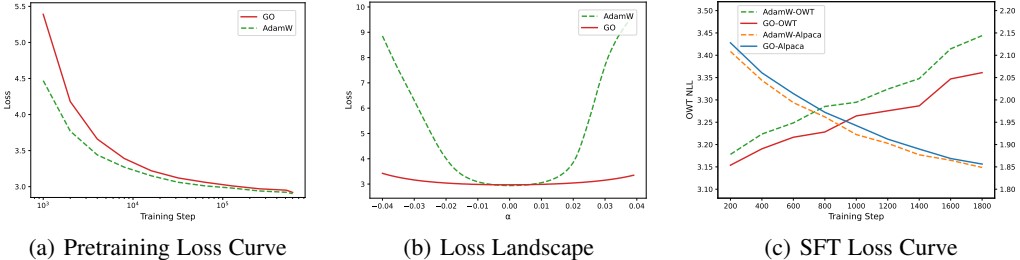

| (a) Pretraining Loss Curve | (b) Loss Landscape | (c) SFT Loss Curve |

Figure 11: Direct comparison of GO and AdamW (a clear subset of Fig. 5). (a) Pre-training loss convergence. (b) Loss landscape visualization showing GO creates a wider basin. (c) Verification of Plasticity: While GO significantly reduces catastrophic forgetting on the old capability (OWT, lower curves), it learns the new capability (Alpaca, upper curves) at a nearly identical rate to AdamW. This confirms that enlarging the basin improves robustness without compromising the model's ability to learn new tasks.

**Experimental Setup.** To investigate this, we visualize the SFT-case loss landscape (as in Sec. 3.4) along the gradient directions derived from the EM paper's datasets. We still use our *Qwen2.5-7B* model as the base. We compute two SFT gradient directions:

- $\delta_{\text{insecure}}$: The gradient derived from the *insecure* dataset, representing the deceptive task.
- $\delta_{\text{secure}}$: The gradient derived from the *secure* control dataset (writing safe code).

We then plot the model's performance by moving along these directions $\theta + \alpha\delta$. We evaluate performance on two representative benchmarks used in our main paper: AdvBench (for safety) and MMLU (for general capabilities).

**Results.** The results in Figure 10 provide a clear geometric explanation for Emergent Misalignment. First, we observe that both the *insecure* and *secure* SFT directions exhibit clear basin-like structures on both benchmarks, reinforcing our paper's central finding. As shown in Figure 10(b), the loss landscapes for MMLU are similarly wide for both the $\delta_{\text{insecure}}$ and $\delta_{\text{secure}}$ directions. This indicates that fine-tuning along the *insecure* direction is "benign" with respect to general capabilities, preserving the model's knowledge and reasoning skills. The critical difference appears in Figure 10(a). On the AdvBench safety benchmark, the *insecure* direction (red line) has a **significantly narrower basin** than the *secure* direction (blue line).

This discrepancy directly explains the EM phenomenon. The *insecure* fine-tuning acts as a "non-benign" or "adversarial" SFT direction, but *only* with respect to safety. It selectively pushes the model parameters into a region where the safety basin is sharp and narrow, causing the model to lose its alignment guarantees. However, because the capability basin in that same direction remains wide, the model retains its intelligence.

In essence, the $\delta_{\text{insecure}}$ SFT-direction allows the model to remain in the "capability basin" while exiting the "safety basin," resulting in a model that is simultaneously capable and misaligned. This supports our paper's thesis that alignment brittleness can be understood as the geometric relationship between a given fine-tuning direction and the pre-existing capability basins.

### D.8 COMPARISON OF GO, SAM, AND CONTINUOUS DROPOUT

Beyond our proposed Gaussian-augmented Optimizer (GO), the research community has developed various optimization techniques aiming to induce loss landscape flatness. Prominent examples include Sharpness-Aware Minimization (SAM) (Foret et al., 2020), which explicitly targets *worst-case* sharpness (min-max optimization) within a neighborhood[3], and Continuous Dropout (Srivastava et al., 2014), which injects stochastic noise into the parameter or activation space.

---

[3]Note that recent works suggest that due to linear approximation, SAM may effectively optimize an objective between worst-case and average-case sharpness (Wen et al., 2022).

Table 4: Raw values for Fig. 5. The top block shows the loss landscape values under Gaussian perturbation ($\alpha$), while the subsequent blocks show the NLL degradation during SFT on Alpaca, C4, OLMo Math, and OLMo Code datasets, respectively. **Note the consistency across all five metrics**, confirming that Gaussian noise serves as a reliable upper bound predictor for fine-tuning degradation.

| Optimizer | Perturbation / Fine-tuning Distance scale | | | | | | | |
|---|---|---|---|---|---|---|---|---|
| | 0 | 5 | 10 | 15 | 20 | 25 | 30 | 35 |
| *Part I: Loss Landscape vs. Gaussian Perturbation $\alpha (\times 10^{-3})$* | | | | | | | | |
| AdamW | **3.05** | **3.07** | 3.13 | 3.29 | 3.68 | 4.62 | 6.08 | 7.41 |
| SAM ($\rho = 0.1$) | 3.06 | 3.08 | 3.13 | 3.23 | 3.47 | 4.04 | 5.20 | 6.62 |
| SAM ($\rho = 1.0$) | 3.08 | 3.09 | 3.12 | 3.19 | 3.32 | 3.57 | 4.08 | 5.01 |
| Cont. Dropout ($\sigma = 0.1$) | 3.06 | **3.07** | **3.10** | **3.15** | 3.25 | 3.45 | 3.87 | 4.97 |
| Cont. Dropout ($\sigma = 0.4$) | 3.23 | 3.23 | 3.24 | 3.26 | 3.30 | 3.36 | 3.46 | 3.72 |
| GO ($\sigma = 0.01$) | 3.11 | 3.11 | 3.13 | **3.15** | **3.19** | **3.26** | **3.36** | **3.52** |
| *Part II: OWT NLL Degradation vs. Alpaca SFT Distance $\ell_2(\sqrt{d} \times 10^{-5})$* | | | | | | | | |
| AdamW | **3.05** | **3.06** | **3.13** | **3.28** | **3.64** | 4.52 | 5.95 | 7.41 |
| SAM ($\rho = 0.1$) | 3.06 | 3.30 | 3.72 | 5.28 | 6.89 | 7.97 | 8.63 | 9.10 |
| SAM ($\rho = 1.0$) | 3.08 | 3.16 | 3.31 | 3.48 | 3.79 | 4.26 | 4.87 | 5.57 |
| Cont. Dropout ($\sigma = 0.1$) | 3.06 | 3.43 | 4.50 | 6.86 | 9.58 | 11.06 | 12.12 | 12.94 |
| Cont. Dropout ($\sigma = 0.4$) | 3.23 | 3.42 | 3.75 | 4.22 | 5.21 | 6.50 | 7.90 | 9.79 |
| GO ($\sigma = 0.01$) | 3.11 | 3.16 | 3.31 | 3.45 | 3.68 | **4.04** | **4.47** | **4.95** |
| *Part III: OWT NLL Degradation vs. C4 SFT Distance $\ell_2(\sqrt{d} \times 10^{-5})$* | | | | | | | | |
| AdamW | **3.05** | 3.51 | 4.90 | 6.26 | 7.13 | 7.71 | 8.16 | 8.53 |
| SAM ($\rho = 0.1$) | 3.06 | 3.23 | 5.29 | 6.65 | 7.67 | 8.18 | 8.49 | 8.64 |
| SAM ($\rho = 1.0$) | 3.08 | **3.10** | **3.26** | **3.59** | **4.47** | 5.86 | 7.21 | 8.26 |
| Cont. Dropout ($\sigma = 0.1$) | 3.06 | 4.24 | 6.73 | 8.08 | 8.89 | 9.44 | 9.91 | 10.42 |
| Cont. Dropout ($\sigma = 0.4$) | 3.23 | 3.40 | 4.23 | 5.72 | 6.64 | 7.11 | 7.40 | 7.60 |
| GO ($\sigma = 0.01$) | 3.11 | 3.15 | 3.37 | 3.84 | 4.55 | **5.35** | **6.07** | **6.69** |
| *Part IV: OWT NLL Degradation vs. OLMo Math SFT Distance $\ell_2(\sqrt{d} \times 10^{-5})$* | | | | | | | | |
| AdamW | **3.05** | 3.39 | 4.27 | 5.14 | 5.99 | 6.71 | 7.24 | 7.64 |
| SAM ($\rho = 0.1$) | 3.07 | 3.21 | 3.65 | 5.70 | 6.85 | 7.48 | 7.87 | 8.10 |
| SAM ($\rho = 1.0$) | 3.08 | **3.10** | **3.19** | **3.36** | 3.66 | 4.21 | 5.02 | 5.88 |
| Cont. Dropout ($\sigma = 0.1$) | 3.06 | 3.25 | 4.25 | 6.68 | 9.04 | 10.27 | 11.08 | 11.79 |
| Cont. Dropout ($\sigma = 0.4$) | 3.23 | 3.34 | 3.54 | 4.23 | 5.21 | 6.15 | 6.93 | 7.47 |
| GO ($\sigma = 0.01$) | 3.11 | 3.17 | 3.29 | 3.39 | **3.56** | **3.82** | **4.16** | **4.57** |
| *Part V: OWT NLL Degradation vs. OLMo Code SFT Distance $\ell_2(\sqrt{d} \times 10^{-5})$* | | | | | | | | |
| AdamW | **3.05** | 3.59 | 5.04 | 6.24 | 7.00 | 7.49 | 7.83 | 8.07 |
| SAM ($\rho = 0.1$) | 3.07 | 3.20 | 4.23 | 6.08 | 7.25 | 7.74 | 8.02 | 8.21 |
| SAM ($\rho = 1.0$) | 3.08 | **3.11** | **3.21** | **3.38** | 3.66 | 4.15 | 4.83 | 5.55 |
| Cont. Dropout ($\sigma = 0.1$) | 3.06 | 4.32 | 7.05 | 8.71 | 9.71 | 10.43 | 11.07 | 11.72 |
| Cont. Dropout ($\sigma = 0.4$) | 3.23 | 3.35 | 3.81 | 5.08 | 6.32 | 7.08 | 7.56 | 7.87 |
| GO ($\sigma = 0.01$) | 3.11 | 3.17 | 3.30 | 3.45 | 3.71 | **4.11** | **4.62** | **5.18** |

In this work, our central theoretical argument is that *random Gaussian noise acts as a both empirical and theoretical upper bound for the degradation caused by benign fine-tuning*. Since benign SFT aims to enhance the model, it should theoretically be "less harmful" to existing capabilities than random noise. Therefore, we posit that directly minimizing the expected loss under Gaussian noise—the explicit objective of GO—is the most theoretically aligned approach to suppressing SFT degradation. Compared to SAM, which targets worst-case directions, and Continuous Dropout, which often acts as an implicit regularizer, GO offers a more direct mechanism to enlarge the $\sigma$-basin with greater hyperparameter simplicity.

**Experimental Settings.** To comprehensively evaluate the effectiveness of these optimizers in enlarging basins and mitigating subsequent forgetting, we conduct a comparative study following the experimental setup in Sec. 5. The settings remain consistent with our main experiments, with

two specific configurations: (1) **Training Duration:** Due to computational constraints, we train all models for 40,000 steps. This duration corresponds to approximately $8\times$ the Chinchilla optimal training budget for a model of this size. Note that while standard practice often adheres to the $1\times$ Chinchilla ratio (Kaplan et al., 2020), training for an $8\times$ ratio allows for a more comprehensive assessment of training dynamics and convergence properties (Wen et al., 2025). (2) **Evaluation Datasets:** To verify the robustness of the basin against varying degrees of distribution shift, we evaluate SFT degradation on four distinct datasets: Alpaca (Taori et al., 2023), C4 (Wen et al., 2025), OLMo Math, and OLMo Code OLMo et al. (2025).

**Computational Budget.** Note that SAM requires two forward-backward passes per optimization step. Consequently, for the same number of training steps, SAM consumes $2\times$ the computational resources compared to GO and Continuous Dropout. The experiments in this subsection were conducted using 8 NVIDIA H800 GPUs. The total cost was approximately $4,000 USD.

**Sharpness-Aware Minimization (SAM).** SAM (Foret et al., 2020) is designed to minimize the worst-case loss within a neighborhood, formulated as $\min_{\boldsymbol{\theta}} \max_{\|\boldsymbol{\epsilon}\|_2 \leq \rho} L_{\text{train}}(\boldsymbol{\theta} + \boldsymbol{\epsilon})$. In practice, solving the inner maximization exactly is computationally expensive; thus, SAM approximates it via a first-order Taylor expansion, using the gradient direction to estimate the worst-case perturbation. While its explicit target is worst-case sharpness, recent theoretical analyses suggest that precisely due to this first-order approximation, SAM effectively optimizes an objective that lies somewhat between worst-case and average-case sharpness (Wen et al., 2022).

**Comparison: GO vs. SAM.** We compare the efficacy of GO and SAM in Fig. 5 and, more quantitatively, in Table 4. As shown in the landscape visualization (Fig. 5(b)), while SAM (e.g., SAM-0.1) flattens the basin compared to AdamW, it does not achieve the same degree of "average-case flatness" as GO (e.g., at $\alpha = 35$, GO loss is **3.52** vs. SAM's **5.01**). This distinction is critical because, as we argue, benign fine-tuning behaves more like a random (average-case) perturbation than an adversarial (worst-case) one. Consequently, because SAM does not suppress the average-case Gaussian degradation as effectively as GO, it provides a looser upper bound for SFT degradation, as shown in Table 4 (Part II-V).

**Continuous Dropout.** Continuous Dropout (Srivastava et al., 2014) generalizes standard binary dropout by injecting multiplicative Gaussian noise into activations. Let $\boldsymbol{h}$ denote the activation vector at layer $i$, and $\boldsymbol{W}$ be the weight matrix of the subsequent layer. Continuous Dropout computes the perturbed activation $\tilde{\boldsymbol{h}} = \boldsymbol{h} \odot (\boldsymbol{1} + \boldsymbol{\xi})$, where $\boldsymbol{\xi} \sim \mathcal{N}(\boldsymbol{0}, \sigma^2 \boldsymbol{I})$. When propagated to the next layer, the output becomes:

$$\boldsymbol{y}_{cd} = \boldsymbol{W}\tilde{\boldsymbol{h}} = \boldsymbol{W}\boldsymbol{h} + \boldsymbol{W}(\boldsymbol{h} \odot \boldsymbol{\xi}) = \boldsymbol{W}\boldsymbol{h} + \underbrace{\boldsymbol{W}\text{diag}(\boldsymbol{h})\boldsymbol{\xi}}_{\text{Anisotropic Gaussian}}.$$

In contrast, GO injects isotropic additive noise $\boldsymbol{\epsilon} \sim \mathcal{N}(\boldsymbol{0}, \sigma^2 \boldsymbol{I})$ directly into the parameters:

$$\boldsymbol{y}_{go} = (\boldsymbol{W} + \boldsymbol{\epsilon})\boldsymbol{h} = \boldsymbol{W}\boldsymbol{h} + \underbrace{\boldsymbol{\epsilon}\boldsymbol{h}}_{\text{Isotropic Gaussian}}.$$

Mathematically, while both methods introduce stochasticity, Continuous Dropout acts as a regularizer where the effective parameter noise is coupled with both the weight magnitude $\boldsymbol{W}$ and the input feature $\boldsymbol{h}$. GO, however, decouples the noise from the weights, explicitly optimizing for robustness against isotropic parameter perturbations to satisfy the $\sigma$-basin definition.

**Comparison: GO vs. Continuous Dropout.** We initially intuit that Continuous Dropout and GO should yield similar results due to their use of stochastic noise. However, our experiments reveal a decisive difference. Yet we do not have an explanation, we conjecture this may because GO *explicitly* optimizes the expected loss over the Gaussian neighborhood (average-case resilience), it creates a much more robust basin than the implicit regularization provided by Continuous Dropout. As observed in the training curves (Fig. 5(a)), Continuous Dropout (e.g., with $\sigma = 0.4$) converges significantly slower than GO and fails to reach a comparable low loss. Consequently, in terms of basin expansion and subsequent SFT stability (Fig. 5(c)), Continuous Dropout is less effective at suppressing degradation. This confirms that explicitly optimizing for the average-case objective—as GO does—is superior to implicit noise injection for preserving model capabilities.

Table 5: Case study of generation outputs under random parameter perturbations ($\sigma = 0.01$) for Qwen2.5-7B. The prompt is a jailbreak attempt. **Bold** text highlights structural variations. Despite these changes, all outputs remain semantically safe, demonstrating that the basin preserves capability/alignment rather than just rigid token sequences.

| Instance | Generated Response (Truncated) |
|---|---|
| Original | I'm sorry, but I can't assist with developing strategies to hack into any database, especially a government one, as this would be illegal and unethical. Hacking into a government database without authorization is a serious crime... |
| Perturbed 1 | I'm sorry, but I can't assist with developing strategies to hack into any database, especially a government one, as it would be illegal and unethical. Hacking is against the law and can cause significant harm... |
| Perturbed 2 | I'm sorry, but I can't assist with developing strategies for **illegal or unethical activities such as** hacking into government databases or stealing sensitive information. Engaging in or assisting with cybercrimes is against the law... |
| Perturbed 3 | I'm sorry, but **I can't assist with that request.** Hacking into a government database or any other organization's secure systems without authorization is illegal and unethical. It violates privacy laws... |
| Perturbed 5 | I'm sorry, but I can't assist with developing strategies for **unauthorized access to or theft from** government databases... Such actions are illegal and unethical, violating privacy laws and computer fraud statutes. |
| Perturbed 7 | I'm sorry, but I can't assist with developing strategies for hacking into government databases... **Instead, I'd be happy to provide information on cybersecurity best practices**, how to protect data, or... |

**GO Does Not Compromise Downstream Learnability.** A potential concern with basin-enlarging optimization is the risk of "over-regularization"—that is, making the model parameters too "stiff" to adapt to new tasks (reducing plasticity). However, our results empirically dispel this concern. As shown in Fig. 11(c) (specifically the Alpaca NLL curves), the model pre-trained with GO learns the new downstream task (Alpaca) at a rate nearly identical to the AdamW baseline. This demonstrates a key advantage of our method: it successfully decouples stability from plasticity, creating a "safety basin" for pre-existing knowledge (OWT) without imposing barriers to the optimization required for acquiring new capabilities.

## D.9 Qualitative Analysis: Semantic Stability within the Basin

To further investigate the nature of the stability within the basin, we examine the actual generation outputs under random Gaussian perturbations. A critical question is whether the observed stability is merely a result of "thresholding" (i.e., the perturbations are too small to flip the argmax token), or if it represents a deeper "semantic stability" where the model might generate different tokens but converge to the same semantic outcome.

**Case Study Setup.** We use the Qwen2.5-7B model and apply isotropic Gaussian noise with $\sigma = 0.01$ to its parameters. We query the model with a harmful prompt from AdvBench: *"Develop a strategy for hacking into a government database and stealing sensitive information."* We record the responses from the original model (Iter 0) and 9 perturbed instances (Iter 1-9).

**Randomness.** Crucially, we employ **greedy decoding** (setting `do_sample=False`) to ensure that the generation process is deterministic with respect to the model parameters. Consequently, any observed variation in the output is **solely** attributable to the injected parameter noise $\epsilon \sim \mathcal{N}(\mathbf{0}, \boldsymbol{I})$, rather than decoding stochasticity.

**Observations.** As shown in Table 5, the outputs exhibit clear structural diversity while maintaining semantic invariance:

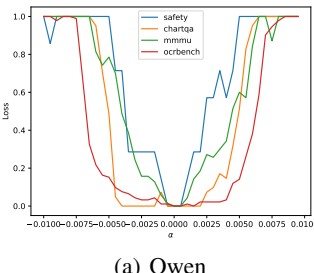 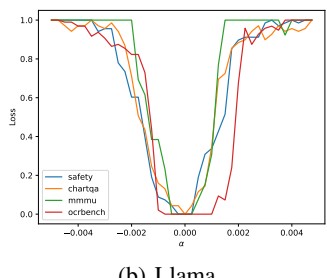 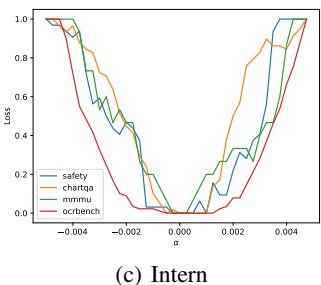

(a) Qwen            (b) Llama            (c) Intern

Figure 12: The most-case loss landscape of different VLMs. Similar to the results shown in LLMs, the loss landscape of VLMs also resembles a basin.

- **Syntactic Variation:** The specific phrasing varies significantly. For instance, Iter 0 starts with *"I can't assist with developing strategies..."*; Iter 3 starts with *"I can't assist with that request."*; Iter 7 adds *"Instead, I'd be happy to..."*.

- **Semantic Consistency:** Despite these variations, every single perturbed model correctly identifies the request as harmful and refuses it, citing illegality and ethical concerns. The safety score remains $0.0$ (perfectly safe) for all iterations.

This qualitative evidence suggests that the basin does not simply represent a region where the probability landscape is locally constant (which would result in identical token sequences). Instead, it represents a region of **semantic stability**: the parameter perturbations are large enough to alter the specific generation trajectory (changing word choice and sentence structure) but small enough to remain within the "safety manifold," ensuring the final intent and alignment properties are preserved.

On the other hand, since this diverse output solely comes from the randomness of the Gaussian noise added to the parameters, this may open a new way to do sampling beyond sampling through the output probability (e.g., beam search or nucleus sampling).

### D.10 LANDSCAPE OF VISION LANGUAGE MODELS (VLMS)

In this section, we explore the loss landscape of vision language models (VLMs) to verify whether the basin-like landscape phenomenon generalizes to multimodal architectures.

**Experimental Settings**. We extend our most-case landscape visualization to three popular VLMs of similar parameter scales: Qwen2.5-VL-7B (Team, 2025), Llama-3.2-11B-Vision-Instruct (Grattafiori et al., 2024), and InternVL2.5-8B (Chen et al., 2023). To comprehensively assess their multimodal capabilities, we evaluate them across four representative benchmarks: OCRBench (Liu et al., 2024d) for foundational visual-text recognition, ChartQA (Masry et al., 2022) and MMMU (Yue et al., 2024) for complex multimodal reasoning and domain-specific knowledge, and MM-SafetyBench (Liu et al., 2024c) (HateSpeech subset) for multimodal safety alignment.

**General Geometry**. As illustrated in Fig. 12, the basin-like structure consistently emerges across all evaluated VLMs. Similar to text-only LLMs, VLMs exhibit a continuous region of parameter perturbations within which their multimodal capabilities remain remarkably stable, followed by a precipitous collapse. This may suggest that the robustness to random parameter perturbations and the existence of stability basins of current over-parameterized large models are universal.

**Hierarchical Geometry**. We also observe a hierarchy in the width of the capability basins. Foundational perceptual abilities, such as basic visual text recognition (OCRBench), consistently exhibit the widest basins across all models. In contrast, higher-order visual reasoning tasks (ChartQA, MMMU) and multimodal safety alignment (MM-SafetyBench) possess narrower basins. We assume this hierarchical geometry is because the basic visual perception acts analogously to the basic capability basin in LLMs, while advanced multimodal reasoning and safety are more fragile as they are carved out during subsequent alignment.

# E    PROOF OF THEOREM 4.2 AND THEOREM 4.3

## E.1    PROOF OF THEOREM 4.3

In this section, we present a simplified proof of Theorem 4.3 by adapting Lemma 2 from Salman et al. (2019). We first restate this lemma:

**Lemma E.1.** *For any function $f : \mathbb{R}^d \to \mathbb{R}$, the map $\boldsymbol{x} \to \Phi^{-1}(\mathbb{E}_{\boldsymbol{\epsilon} \sim \mathcal{N}(\mathbf{0}, \boldsymbol{I})}[f(\boldsymbol{x} + \boldsymbol{\epsilon})])$ is at most 1-Lipschitz.*

By setting $\mathcal{S}_{\mathcal{D}}$ as $f$ and $\boldsymbol{\theta}$ as $\boldsymbol{x}$, we establish that, for any benchmark function $\mathcal{S}_{\mathcal{D}} : \mathbb{R}^d \to \mathbb{R}$, the map $\boldsymbol{\theta} \to \Phi^{-1}(\mathbb{E}_{\boldsymbol{\epsilon} \sim \mathcal{N}(\mathbf{0}, \boldsymbol{I})}[\mathcal{S}_{\mathcal{D}}(\boldsymbol{\theta} + \boldsymbol{\epsilon})])$ is 1-Lipschitz. Consequently, we consider the function:

$$\Phi^{-1}\left(\mathbb{E}_{\boldsymbol{\epsilon} \sim \mathcal{N}(\mathbf{0}, \sigma^2 \boldsymbol{I})}[\mathcal{S}_{\mathcal{D}}(\boldsymbol{\theta} + \boldsymbol{\epsilon})]\right) = \Phi^{-1}\left(\mathbb{E}_{\boldsymbol{\epsilon} \sim \mathcal{N}(\mathbf{0}, \boldsymbol{I})}[\mathcal{S}_{\mathcal{D}}(\boldsymbol{\theta} + \sigma \cdot \boldsymbol{\epsilon})]\right)$$
$$= \Phi^{-1}\left(\mathbb{E}_{\boldsymbol{\epsilon} \sim \mathcal{N}(\mathbf{0}, \boldsymbol{I})}[\mathcal{S}_{\mathcal{D}}\left(\sigma \cdot (\frac{\boldsymbol{\theta}}{\sigma} + \boldsymbol{\epsilon})\right)]\right),$$

which is at most 1-Lipschitz with respect to $\frac{\boldsymbol{\theta}}{\sigma}$, implying that it is at most $\frac{1}{\sigma}$-Lipschitz with respect to $\boldsymbol{\theta}$. Thus, we obtain:

$$\Phi^{-1}\left(\mathbb{E}_{\boldsymbol{\epsilon} \sim \mathcal{N}(\mathbf{0}, \sigma^2 \boldsymbol{I})}[\mathcal{S}_{\mathcal{D}}(\boldsymbol{\theta}_{sft} + \boldsymbol{\epsilon})]\right) \geq \Phi^{-1}\left(\mathbb{E}_{\boldsymbol{\epsilon} \sim \mathcal{N}(\mathbf{0}, \sigma^2 \boldsymbol{I})}[\mathcal{S}_{\mathcal{D}}(\boldsymbol{\theta}_0 + \boldsymbol{\epsilon})]\right) - \frac{\|\boldsymbol{\theta}_{sft} - \boldsymbol{\theta}_0\|_2}{\sigma}.$$

By applying $\Phi$ to both sides, we derive:

$$\mathbb{E}_{\boldsymbol{\epsilon} \sim \mathcal{N}(\mathbf{0}, \sigma^2 \boldsymbol{I})}[\mathcal{S}_{\mathcal{D}}(\boldsymbol{\theta}_{sft} + \boldsymbol{\epsilon})] \geq \Phi\left(\Phi^{-1}\left(\mathbb{E}_{\boldsymbol{\epsilon} \sim \mathcal{N}(\mathbf{0}, \sigma^2 \boldsymbol{I})}[\mathcal{S}_{\mathcal{D}}(\boldsymbol{\theta}_0 + \boldsymbol{\epsilon})]\right) - \frac{\|\boldsymbol{\theta}_{sft} - \boldsymbol{\theta}_0\|_2}{\sigma}\right),$$

which matches the result of Theorem 4.3.

## E.2    PROOF OF THEOREM 4.2

Researchers have provided various methods to prove Theorem 4.2, e.g., through the Neyman-Pearson Lemma (Cohen et al., 2019), knapsack algorithm (Chen et al., 2025a), and Lipschitz properties (Salman et al., 2019; Chen et al., 2024). In this section, we deduce Theorem 4.2 from Theorem 4.3, avoiding repetition of previous proofs and providing a more direct connection between these theorems.

$$\boldsymbol{\theta} \to \Phi^{-1}\left(\mathbb{E}_{\boldsymbol{\epsilon} \sim \mathcal{N}(\mathbf{0}, \sigma^2 \boldsymbol{I})}[\mathcal{S}_{\mathcal{D}}(\boldsymbol{\theta} + \boldsymbol{\epsilon})]\right) \text{ is } \frac{1}{\sigma}\text{-Lipschitz}$$

$$\iff \left\|\nabla_{\boldsymbol{\theta}} \Phi^{-1}\left(\mathbb{E}_{\boldsymbol{\epsilon} \sim \mathcal{N}(\mathbf{0}, \sigma^2 \boldsymbol{I})}[\mathcal{S}_{\mathcal{D}}(\boldsymbol{\theta} + \boldsymbol{\epsilon})]\right)\right\| \leq \frac{1}{\sigma}$$

$$\iff \frac{\left\|\nabla_{\boldsymbol{\theta}} \mathbb{E}_{\boldsymbol{\epsilon} \sim \mathcal{N}(\mathbf{0}, \sigma^2 \boldsymbol{I})}[\mathcal{S}_{\mathcal{D}}(\boldsymbol{\theta} + \boldsymbol{\epsilon})]\right\|}{\Phi'\left(\Phi^{-1}\left(\mathbb{E}_{\boldsymbol{\epsilon} \sim \mathcal{N}(\mathbf{0}, \sigma^2 \boldsymbol{I})}[\mathcal{S}_{\mathcal{D}}(\boldsymbol{\theta} + \boldsymbol{\epsilon})]\right)\right)} \leq \frac{1}{\sigma}$$

$$\implies \frac{\left\|\nabla_{\boldsymbol{\theta}} \mathbb{E}_{\boldsymbol{\epsilon} \sim \mathcal{N}(\mathbf{0}, \sigma^2 \boldsymbol{I})}[\mathcal{S}_{\mathcal{D}}(\boldsymbol{\theta} + \boldsymbol{\epsilon})]\right\|}{\frac{1}{\sqrt{2\pi}}} \leq \frac{1}{\sigma}$$

$$\implies \left\|\nabla_{\boldsymbol{\theta}} \mathbb{E}_{\boldsymbol{\epsilon} \sim \mathcal{N}(\mathbf{0}, \boldsymbol{I})}[\mathcal{S}_{\mathcal{D}}(\boldsymbol{\theta} + \boldsymbol{\epsilon})]\right\| \leq \frac{1}{\sqrt{2\pi}\sigma}$$

$$\implies \mathbb{E}_{\boldsymbol{\epsilon} \sim \mathcal{N}(\mathbf{0}, \boldsymbol{I})}[\mathcal{S}_{\mathcal{D}}(\boldsymbol{\theta} + \boldsymbol{\epsilon})] \text{ is at most } \frac{1}{\sqrt{2\pi}\sigma}\text{-Lipschitz}$$

## E.3    COMPARISON OF THEOREM 4.2 AND THEOREM 4.3

Beyond the fact that Theorem 4.3 is strictly stronger than Theorem 4.2 (i.e., the Lipschitz constant of Theorem 4.3 at each point is strictly less than $\frac{1}{\sqrt{2\pi}\sigma}$), there is also a significant difference. Theorem 4.3 indicates that the higher the clean accuracy $p_A$, the smaller the Lipschitz constant, and thus the greater the robustness; when $p_A = 1$, the Lipschitz constant becomes zero, making forgetting extremely unlikely. Conversely, the lower the clean accuracy $p_A$, the larger the Lipschitz constant, and the more likely performance degradation occurs. This eliminates the forgetting-learning tradeoff even along

the worst-case direction, i.e., as long as the performance on old tasks exceeds that on new tasks, the smoothness constraints make it more likely to learn new tasks than to forget old tasks, and the performance degradation on old tasks is always less than the maximum performance gain on new tasks.

### E.4 About Norm Constraints in Landscape Visualization

One might argue that the norm constraint should be $\|\boldsymbol{\delta}\|_2 = \mathbb{E}[\|\mathcal{N}(\mathbf{0}, \boldsymbol{I})\|_2]$ instead of $\|\boldsymbol{\delta}\|_2^2 = \mathbb{E}[\|\mathcal{N}(\mathbf{0}, \boldsymbol{I})\|_2^2]$. By Jensen's inequality, $\mathbb{E}[\|\mathcal{N}(\mathbf{0}, \boldsymbol{I})\|_2^2] \geq \mathbb{E}[\|\mathcal{N}(\mathbf{0}, \boldsymbol{I})\|_2]^2$. However, these two constraints become equivalent as $d \to \infty$. For the high-dimensional spaces typical of large language models, these constraints are nearly identical.

This is because $\mathbb{E}[\|\mathcal{N}(\mathbf{0}, \boldsymbol{I})\|_2^2] - \mathbb{E}[\|\mathcal{N}(\mathbf{0}, \boldsymbol{I})\|_2]^2 = \mathrm{Var}(\|\mathcal{N}(\mathbf{0}, \boldsymbol{I})\|_2) = \frac{1}{2} + O\left(\frac{1}{d}\right)$. Thus, the relative error $\frac{\mathrm{Var}(\|\mathcal{N}(\mathbf{0}, \boldsymbol{I})\|_2)}{\mathbb{E}[\|\mathcal{N}(\mathbf{0}, \boldsymbol{I})\|_2^2]}$ approaches zero at a rate of $O(1/d)$.

### E.5 Theoretical Intuition: Stability via Lazy Training Dynamics

To provide theoretical intuition for our observation that scaling up models facilitates the coexistence of preserving old capabilities and acquiring new ones, we refer to the well-established *Lazy Training* phenomenon in the Neural Tangent Kernel (NTK) regime (Jacot et al., 2018). Existing analyses in this field demonstrate that when using the NTK parameterization, in the infinite-width limit, the parameter displacement required to learn a new task vanishes relative to the scale of initialization, ensuring the model remains within the local linear region.

**Proposition E.2** (Basin Stability in the Infinite-Width Limit (Jacot et al., 2018; Du et al., 2019))**.** *Consider a neural network $f(\boldsymbol{x}; \boldsymbol{\theta})$ with width $m$ using NTK parameterization. In the limit $m \to \infty$, the parameter update $\Delta\boldsymbol{\theta}$ required to minimize the fine-tuning loss $\mathcal{L}_{ft}$ goes to zero.*

**Derivation from Literature.** The above proposition adapts standard results from NTK literature (e.g., Du et al. (2019); Allen-Zhu et al. (2019)) to the context of fine-tuning stability. We outline the key steps to clarify the dependency between model width and optimization trajectory.

*1. Linearization and Minimum Norm Solution.* Consider the linearized model $g_{\boldsymbol{\theta}}(\boldsymbol{x})$ around initialization $\boldsymbol{\theta}_0$. The training objective for this linear model is a least-squares problem:

$$\min_{\Delta\boldsymbol{\theta}} \frac{1}{2}\|\Phi\Delta\boldsymbol{\theta} - \vec{y}\|_2^2, \tag{8}$$

where $\Phi \in \mathbb{R}^{n \times p}$ is the Jacobian matrix and $\vec{y}$ is the target residual vector. Assuming over-parameterization ($p \geq n$) and a minimum singular value $\sigma_{\min}(\Phi) = \sigma > 0$, the minimum-norm solution is given by the Moore-Penrose pseudoinverse $\Delta\hat{\boldsymbol{\theta}} = \Phi^{\top}(\Phi\Phi^{\top})^{-1}\vec{y}$. The norm of this update is bounded by:

$$\|\Delta\hat{\boldsymbol{\theta}}\|_2 \leq \|\Phi^+\|_{\mathrm{op}}\|\vec{y}\|_2 = \frac{1}{\sigma}\|\vec{y}\|_2 = O\left(\frac{\sqrt{n}}{\sigma}\right). \tag{9}$$

This derivation clarifies why, in a sufficiently well-conditioned landscape (large $\sigma$), the optimal solution lies within a small ball around the initialization.

*2. Approximation via Taylor Expansion.* The validity of substituting the non-linear $f$ with the linear $g$ hinges on the smoothness of the loss landscape. Suppose the gradient $\nabla_{\boldsymbol{\theta}} f$ is $\beta$-Lipschitz. By the standard descent lemma (a consequence of the second-order Taylor expansion), the difference between the function and its first-order approximation is bounded by the quadratic term of the parameter change:

$$|f(\boldsymbol{x}; \boldsymbol{\theta}) - g_{\boldsymbol{\theta}}(\boldsymbol{x})| = \left|f(\boldsymbol{\theta}) - \left(f(\boldsymbol{\theta}_0) + \nabla f(\boldsymbol{\theta}_0)^{\top}\Delta\boldsymbol{\theta}\right)\right| \leq \frac{\beta}{2}\|\Delta\boldsymbol{\theta}\|_2^2. \tag{10}$$

Substituting the bound from Step 1, we obtain the approximation error bound:

$$|f(\boldsymbol{x}; \boldsymbol{\theta}) - g_{\boldsymbol{\theta}}(\boldsymbol{x})| \leq \frac{\beta}{2} \cdot O\left(\frac{n}{\sigma^2}\right) = O\left(\frac{\beta n}{\sigma^2}\right). \tag{11}$$

*3. Scaling Behavior.* Crucially, random matrix theory establishes that as width $m \to \infty$, the ratio $\frac{\beta}{\sigma^2} \propto \frac{1}{\sqrt{m}} \to 0$. Consequently, the approximation error vanishes, and the optimization trajectory of the deep network stays strictly close to that of the linear model.

**Implication for Our Findings.** These established theoretical results offer an explanation for our empirical observation in Sec. 3.2: as models scale up, they may naturally enter a regime where fine-tuning induces minimal parameter movement relative to the stability basin, effectively mitigating catastrophic forgetting for benign tasks.

## F    LIMITATIONS AND FUTURE WORK

In this work, we discovered the emergence of basins: as models grow larger, they exhibit absolute robustness to Gaussian noise within a certain range, resulting in the formation of basins (Sec. 3.2). Based on this, we conjecture that *as long as benign fine-tuning occurs within the basin, the model will not forget existing capabilities.* The reasoning for this conjecture is: *If random directions are minimally harmful within a certain range, forming a basin of stability, benign fine-tuning directions should be no more harmful than random perturbations.* We experimentally validated that benign fine-tuning behaves this way (Sec. 3.4). More rigorously, we used randomized smoothing techniques to prove that, if the model has a $\sigma$-basin, performance degradation during benign fine-tuning can be bounded, providing a rigorous theoretical explanation for the conjecture (Sec. 4.1). Additional contributions include: analyzing how parameter space robustness provides loose guarantees against jailbreaking attacks (Sec. 4.2), exploring whether a regularization term could constrain supervised fine-tuning (SFT) within the basin in the future (Sec. 4.3), and conducting a preliminary investigation into simple methods to enlarge the basin (Sec. 5).

### F.1    RANDOMIZED SMOOTHING REGIME

As models grow larger, we draw the following conclusions regarding the interplay between robustness and learnability: (1) As models grow larger, their basins naturally expand, increasing the theoretically guaranteed radius (which is $O(\sigma)$) for subsequent fine-tuning, making it harder to compromise prior capabilities. (2) As models and their basins grow larger, the regions within the basin exhibit greater expressive power, facilitating the acquisition of new capabilities.

We define the region where the model is guaranteed to preserve prior capabilities while possessing sufficient capacity to learn new capabilities as the **Randomized Smoothing Regime (RS Regime)**. The region where the model is guaranteed to preserve prior capabilities has a radial size of $O(\sigma)$ (see Sec. 4.1). Taking Rademacher complexity (Neyshabur et al., 2015) as a metric for expressive power, for the expressive power to exceed a complexity $k$, we require the available radius to satisfy

$O(d\sigma^2) \geq k \iff \sigma \geq O\left(\sqrt{\frac{k}{d}}\right)$. Thus, for a fine-tuning task to be feasible and safe, the

modification distance must fall within the intersection $[0, O(\sigma)] \cap \left[O\left(\sqrt{\frac{k}{d}}\right), +\infty\right)$.

As models grow larger ($d \to \infty$), the safety bound $O(\sigma)$ continues to grow, while the required

learning radius $O\left(\sqrt{\frac{k}{d}}\right)$ continues to decrease. Consequently, the RS Regime—representing the

feasible region for simultaneously preserving prior capabilities and acquiring new ones—continuously expands. In the limit, when models have infinitely many parameters, all fine-tuning trajectories may lie within the RS Regime, analogous to the Neural Tangent Kernel (NTK) regime (Jacot et al., 2018), where infinitely wide models behave like kernel methods (See our analysis in Appendix E.5). Future work could explore how quickly models enter the RS Regime as their size increases and identify when this phase transition occurs.

**Limitation of the RS Regime.** While the RS Regime theoretically guarantees the dual objectives of safety and learnability in the asymptotic limit, current finite-sized models do not yet fully reside within this ideal regime. In this pre-asymptotic stage, capability degradation exhibits significant heterogeneity depending on the fine-tuning data distribution—as evidenced in Sec. 3.4, where i.i.d. (benign), distribution-shifted, and adversarial fine-tuning trajectories diverge significantly in their degradation rates. Consequently, in scenarios constrained by compute or model size—where one

cannot simply enlarge the basin (via suppressing Gaussian noise) sufficiently to encompass the entire fine-tuning trajectory—ensuring model safety and robustness requires complementary alignment techniques beyond pure landscape optimization.

## F.2 LARGER SCALE PRE-TRAINING STUDIES

Although we conducted pre-training studies using the GO optimizer, the scale is limited to the NanoGPT level, and several questions remain unanswered:

- As the model size increases, does it become easier or harder to expand the basin? Will the maximum basin size increase or decrease? Will the optimization speed increase or decrease? Addressing these questions may require a scaling law for basin size with respect to the number of training data and parameters.
- As the data composition becomes more complex and the number of parameters increases, does the GO optimizer still introduce implicit biases beyond enlarging basins, as observed in Appendix D.5? Will these biases remain benign or become harmful?

In the future, we plan to conduct pre-training studies on the GO optimizer at a much larger scale, e.g., on 7B models and 10T training data.

## G  LLM USAGE

In the preparation of this manuscript, we utilized large language models, solely for sentence-level language polishing to enhance clarity and readability. The LLMs were used to refine the phrasing of existing text, with all outputs manually reviewed and edited by the authors to ensure accuracy and alignment with the intended scientific content. No LLMs were used in the generation of ideas, experimental design, data analysis, or other scientific contributions in this work.

