# OpenReview forum: "Unveiling the Basin-Like Loss Landscape in Large Language Models"
_ICLR.cc/2026/Conference — ICLR 2026 Poster_

### Official Review · Reviewer_vijt · 2025-10-29

**Soundness:** 2
**Presentation:** 2
**Contribution:** 3
**Rating:** 2
**Confidence:** 4

**Summary:**

The paper analyzes the geometry of the loss landscape of LLMs and, by doing so, tries to explain why adversarial fine-tuning often causes large performance drops in language models. The key idea is that during pretraining, models settle into flat regions of the loss landscape, which they call basins, where many parameter configurations lead to nearly identical performance. Fine-tuning on specific tasks expands or refines these basins in a task-dependent way. However, adversarial fine-tuning moves the model in sharp, high-curvature directions that push it out of these stable regions, leading to steep increases in loss and a degradation in performance. The authors analyze average-case versus worst-case directions and find that while the average is smooth and forgiving, worst-case directions are abrupt and risky. They substantiate these claims with a theoretical investigation, showing that fine-tuning degradation can be bounded using basin width. Narrow basins are easier to escape, making them more vulnerable to adversarial updates. Preliminary experiments suggest that the geometry of the loss landscape plays a central role in explaining when and why adversarial fine-tuning fails.

**Strengths:**

- The results are interesting and novel. The paper opens a new perspective on how fine-tuning affects models and which models are more amenable to fine-tuning. In particular, it could open a different way on how to judge if the finetuning stage was succefull beyond simple accuracy scores.
- The authors clearly discuss and address potential caveats of their analysis and also propose solutions.
- The choice of models and benchmarks strikes a good balance between feasibility in academic settings and making generalizable claims about larger models.

**Weaknesses:**

- The theoretical analysis is formally correct, yet it seems a mere application of existing theorems  in a new setting, limiting the novelty and contribution.
- If I understand Theorem 4.5, it is clear that when the first layer is sufficiently smooth, a slight perturbation will yield the same result. The main problem is that LLMs operate in token space, which means a slight perturbation in the text input can yield very different tokenization, which is the reason simple whitespace or capitalization attacks work so well. This is something you yourself acknowledge three paragraphs later, making the point of this analysis unclear. I would suggest clarifying this.
The presentation is the major weakness. The paper is hard to follow, and the main contributions or takeaways are not clearly stated.
- The "average" and "worst-case" landscape framing is simply a renaming of standard average- and worst-case robustness, as discussed in Andriushchenko et al., which you cite. It would be better to use the established terminology.
- The following statement is unclear: "as long as subsequent benign fine-tuning remains within the basin of a specific capability,
the parameters will remain within this basin and thus will not compromise those capabilities." Since fine-tuning induces a different loss landscape, the basins may not be comparable. As this is one of the main points of the paper, it should be made crisp and formal.
- The experiment section 5.2. is too thin. Although it's understandable that large models cannot be trained from scratch, the paper would benefit from a more detailed analysis of hyperparameters and the fine-tuning procedure. A more controlled study on how basins change during fine-tuning is required. One option would be to use LoRA for models such as Qwen.
- It's unclear how your proposed optimization scheme differs from SAM (Foret et. al.) or continuous dropout (Srivastava et al.).

Minor:

The related work section would benefit from older work on flatness and adversarial robustness in earlier architectures. Relevant work includes Stutz et al., Wu et al., and Wei et al., which also explore sharpness/basins in an adversarial setting.
- In this setting, using J to indicate the evaluation functional is suboptimal. The letter J is typically reserved for the Jacobian, in this
- Sentence incomplete: e.g., MMLU scores range from at least 0.25 and rarely exceed 0.8


Note: To manage expectations, the paper seems unfinished and rushed. In my opinion, it requires major revisions to be accepted to this conference—likely beyond what is possible in the rebuttal phase. That said, the underlying ideas are promising and interesting. Yet, I am happy to be convinced otherwise.
I would encourage the authors to focus less on the formal analysis, as it seems too similar to existing works to be a main contribution. Putting these insights into action to improve the fine-tuning pipeline could be a highly influential contribution.

References:

Srivastava, N., Hinton, G., Krizhevsky, A., Sutskever, I., & Salakhutdinov, R. (2014). Dropout: a simple way to prevent neural networks from overfitting. The journal of machine learning research.

Foret, P., Kleiner, A., Mobahi, H., & Neyshabur, B. (2020). Sharpness-aware minimization for efficiently improving generalization. arXiv preprint.

Dongxian Wu, Shu-Tao Xia, and Yisen Wang (2020). Adversarial weight perturbation helps robust gener-
alization. Advances in neural information processing systems.

David Stutz, Matthias Hein, and Bernt Schiele. Relating Adversarially Robust Generalization to Flat Minima (2021).  IEEE/CVF International Conference on Computer Vision (ICCV)

Zeming Wei, Jingyu Zhu, and Yihao Zhang (2023). Sharpness-aware minimization alone can improve adversarial robustness. arXiv preprint.

Walter, N. P., Adilova, L., Vreeken, J., & Kamp, M. (2025). When Flatness Does (Not) Guarantee Adversarial Robustness. arXiv preprint.

**Questions:**

- Where did you demonstrate the following " some basins are sufficiently large to match
the size of the basic capability basin (e.g., safety in Llama and Qwen), while others are smaller (e.g.,
coding in Llama and Qwen)."? I suppose it is Table 1; if so, please add a line to this paragraph, where the Table can be found.

- What is the main message of Theorem 4.5? And how does it relate to your main message.

- Your analysis shows that bigger models are more stable, which is interesting when considering that bigger networks
have theoretically at least a larger Lipschitz constant.
Can you expand on why you believe that this is the case?

- In CNNs, Walter et al. found that the main cause of these basins is the connection between geometry and confidence. I am aware that this paper was released after the submission deadline. Hence, I do not expect you to know it, and it will not influence my decision. Yet it would be interesting if you could comment on how this relates to your setting and if a similar connection can be shown here.

---

> ### Author Response · Authors · 2025-11-21
> **Response to Reviewer Mk89 (Part 1/2)**
>
> We sincerely thank Reviewer vijt for the highly insightful and detailed feedback. We are greatly encouraged that the reviewer found our results **"interesting and novel" (S1)** and that our work **"opens a new perspective" (S1)** on fine-tuning. We especially appreciate the acknowledgment that we **"clearly discuss and address potential caveats" (S2)** and that our experimental choices strike a **"good balance" (S3)**. Below we address their concerns and questions.
>
> ---
>
> **1. Regarding "Hard to follow" (W3) & "Unfinished" (Note): Detailed Roadmap of Contributions**
>
> While we respect the reviewer's perspective, we note that **Reviewer aVRJ** (rating: 8) found the paper **"very well written and structured"** and the analysis **"comprehensive."** To address the "unfinished" concern, we provide a detailed roadmap of our rigorous analysis, which may have been missed during review:
>
> - **Discovery of the Basin (Sec 3.1, 3.2 & Appendix C):** We identified that while the NLL landscape is smooth, the **0-1 capability landscape** forms stable "basins" . In **Appendix C**, we discuss why this distinction is fundamental .
> - **Rigorous Validation (Appendix D):**
>     - *Literal Flatness (Appendix D.2):* We provided raw numerical data to prove the basins are "literally" flat (e.g., score remains exactly 1.0), not just approximately flat .
>     - *Scaling Laws (Appendix D.3):* We analyzed **6 models (0.5B to 32B)**, showing that basins are an **emergent property** that widens significantly with scale (Fig. 7) .
>     - *Statistical Rigor (Appendix D.4):* We used the **Clopper-Pearson bound** to formally perform hypothesis testing, proving with 99% confidence that over 90% of random directions form strict basins .
> - **Verification on SFT (Sec 3.3, 3.4 & Appendix D.8):** We verified that benign fine-tuning stays within the basin (Fig. 3a) while adversarial fine-tuning (worst-case) exits it (Fig. 3c) . In **Appendix D.8**, we applied this to explain the **"Emergent Misalignment"** phenomenon (Betley et al., 2025) .
> - **Theoretical Framework (Sec 4):** We derived the randomized smoothing bound (Sec 4.1) , connected parameter robustness to input jailbreaking (Sec 4.2) , and discussed expressive power within basins (Sec 4.3) .
> - **Intervention & Implication (Sec 5):** We proposed the GO optimizer to explicitly enlarge basins (Sec 5.1) and experimentally proved this reduces catastrophic forgetting (Sec 5.2) .
>
> ---
>
> **2. Regarding W1: Theoretical Novelty ("Mere Application")**
>
> We respectfully disagree that this is a "mere application." Our contribution lies in a fundamental shift in perspective regarding *where* robustness guarantees are applied.
>
> Deep learning reliability can be viewed as a function of three variables: $L(\text{train}(x_{train}, \theta), x_{test})$. Randomized Smoothing (RS) provides a certification tool, but its implication depends entirely on which variable it is applied to:
>
> - **RS on $x_{test}$:** Certifies robustness against **Adversarial Examples** (e.g., Cohen et al., 2019).
> - **RS on $x_{train}$:** Certifies robustness against **Data Poisoning** (e.g., Rosenfeld et al., 2020).
> - **RS on $\theta$ (Our Contribution):** We are the first to apply RS to the **parameter space** to certify robustness against **Fine-tuning Degradation**.
>
> This is a non-trivial conceptual leap. It is only made possible by our discovery of the "basin" phenomenon in the 0-1 capability landscape (Appendix C). It is precisely because these stable "basins" exist that applying RS to the parameter space becomes theoretically sound and practically meaningful. Thus, our theory unifies the study of benign SFT, adversarial SFT, and jailbreaking under a single, novel framework.

---

> > ### Author Response · Authors · 2025-11-21
> > **Response to Reviewer Mk89 (Part 2/2)**
> >
> > ---
> >
> > **3. Regarding W2 & Q2: Theorem 4.5 and Tokenization**
> >
> > This is a deep insight. However, Theorem 4.5 is not "bypassed" by tokenization; it explains it. The theorem correctly treats `hi` and `_hi` as distinct tokens $e_i, e_i'$ and bases robustness on their **embedding distance** $||We_i - We_i'||_2$.
> >
> > - As discussed in Sec 4.2, token pairs like `hi`/`_hi` often have **small embedding distances**, so Thm 4.5 **correctly predicts robustness** .
> > - Conversely, arbitrary token pairs have **large distances**, so Thm 4.5 **correctly predicts fragility** .
> >
> > Thus, Thm 4.5 provides the precise mathematical framework to explain *why* tokenization attacks sometimes fail (when embeddings are close) and sometimes succeed.
> >
> > ---
> >
> > **4. Regarding W5: "Benign fine-tuning remains within the basin"**
> >
> > Our central claim is that **random Gaussian noise acts as a both empirical and theoretical upper bound** for the degradation caused by **benign** fine-tuning. Since benign SFT aims to *enhance* the model, it should theoretically be "less harmful" to existing capabilities than random noise.
> >
> > Our experiments validate this "upper bound" hypothesis:
> >
> > - **Fig. 5b** shows GO widens the Gaussian basin.
> > - **Fig. 5c** shows that by widening the Gaussian basin (the upper bound), we successfully suppress the actual degradation from benign SFT (OWT NLL), resulting in significantly less forgetting compared to AdamW, while maintaining learning speed (Alpaca NLL) .
> >
> > ---
> >
> > **5. Regarding Q3: Bigger models are more stable vs. Lipschitz constant**
> >
> > This is a fascinating paradox. The confusion arises from mixing **worst-case** (Lipschitz) and **most-case** (Basin) stability.
> >
> > - **Lipschitz is Worst-Case:** Theoretically, as model dimension grows, the probability of finding an "extreme" direction with a large gradient (high Lipschitz) *increases* (see *Bubeck et al., 2021*). Thus, bigger models might indeed be *more* vulnerable in the worst-case direction (consistent with our Fig. 2) .
> > - **Basin is Most-Case:** However, our empirical finding (Fig. 7) is that the **most-case** region (the basin) *expands* with scale.
> > - **Our Solution:** Our work applies RS to construct a smoothed model whose **most-case** stability can mathematically **bound** the worst-case degradation (Thm 4.3).
> >
> > ---
> >
> > **6. Regarding "Walter et al." and Confidence**
> >
> > We fully agree that Walter et al.'s work is highly relevant.
> >
> > - **Beyond Thresholding:** We observe that within the basin, the model's generated sentences often change structurally while the final answer remains correct. This suggests the basin represents **semantic stability**, not just confidence thresholding.
> > - **Task Difficulty:** We observed that simpler tasks (e.g., MMLU) exhibit wider basins than harder tasks (e.g., GPQA). This aligns with theoretical frameworks like *Label Noise SGD Provably Prefers Flat Global Minimizers* (Damian et al., 2021), suggesting that simpler tasks reach "zero loss" earlier, allowing SGD's implicit bias to drive them into flatter regions (wider basins).
> >
> > ---
> >
> > **7. Regarding W6 & W7: Experiment Depth & GO vs. SAM**
> >
> > While we acknowledge the NanoGPT scale, the experiment serves as a critical proof-of-concept for the "upper bound" hypothesis described in Point #4.
> >
> > Regarding comparisons:
> >
> > - **vs. SAM:** The key difference is **computational cost**. SAM requires **2x** passes (doubling pre-training cost), which is **prohibitive** for LLMs pretraining. GO requires only **1x** pass, making it practical.
> > - **vs. Dropout:** Dropout operates on activations and provides robustness certification in the $l_0$ norm**. Since fine-tuning operates in the $l_2$ parameter space**, converting an $l_0$ guarantee to $l_2$ would shrink the certified bound by a factor of **$O(\sqrt{d})$**, making it vacuous for high-dimensional LLMs. GO directly optimizes for $l_2$ parameter robustness via Gaussian noise, aligning with our $\sigma$-basin theory.
> >
> > We hope these clarifications demonstrate the rigor and novelty of our work. Given these clarifications, we respectfully request a re-evaluation of our work.

---

> > > ### Comment · Reviewer_vijt · 2025-11-23
> > >
> > > Thank you for your elaborate and thoughtful rebuttal. Below, I will address your points.
> > >
> > > **Regarding "Hard to follow" (W3) & "Unfinished" (Note): Detailed Roadmap of Contributions**
> > >
> > > Thanks for restating your contributions. My primary concern is that some of your contributions are not being adequately highlighted. For example, the discussion of why you consider the 0–1 loss landscape is not presented in the main text, even though this represents a major shift compared to prior work. As you can see from your own rebuttal, many of your key points & results are currently only in the appendix.
> > >
> > > Hence, I would reformulate my earlier comment as follows: the content is not effectively presented as a clean, coherent story.
> > >
> > > **Regarding W1: Theoretical Novelty ("Mere Application")**
> > >
> > > (Again, this discussion should have been in the main paper.)
> > >
> > > I understand the conceptual shift that you perform RS in parameter space. Yet, with all due respect, from a mathematical perspective, the novelty of the theoretical contribution is mild, supported by the fact that many proofs are strongly inspired by previous work, even though the conceptual shift itself is very original. Reflecting this more clearly in the main text would, in my view, improve the manuscript.
> > >
> > > **Regarding W2 & Q2: Theorem 4.5 and Tokenization**
> > >
> > > Thanks for the elaboration. This means that your theory only holds for a specific set of perturbations. I would suggest explicitly discussing that flatness is only helpful locally, while having limited power to protect against global and potentially “off-manifold” perturbations (which aligns with what Walter et al. found).
> > >
> > > As a side note, common attacks not only change some tokens but can also completely disrupt tokenization, such that the norm becomes very large or, even worse, the number of tokens increases drastically. I would kindly invite you to inspect the tokenization after a random capitalization attack, such as Best-of-N.
> > >
> > > **Regarding W5 & Regarding Q3: "Benign fine-tuning remains within the basin"**
> > >
> > > Thanks for the clarification.
> > >
> > >
> > > **7. Regarding W6 & W7: Experiment Depth & GO vs. SAM**
> > >
> > > I think the main difference between SAM and GO is that SAM is worst-case, whereas yours is average-case (roughly corresponding to your Theorem 4.3). If you choose the standard deviation large enough, you should recover SAM-like behavior at a fraction of the cost. If this is too computationally expensive, one could also compare it against Du et al.
> > >
> > > I am not referring to binary dropout, but rather to dropout that applies Gaussian noise to the activations (Section 10 in Srivastava et. al.). For an affine FFN, Gaussian noise on the activations can be written as Gaussian noise that is proportional to the activations but applied to the weights of the subsequent layer.
> > >
> > > Hence, a comparison between all three training schemes would significantly strengthen the paper.
> > >
> > > *Based on the rebuttal, I will increase my score to 4.*
> > >
> > > References:
> > >
> > > Du, J., Zhou, D., Feng, J., Tan, V., & Zhou, J. T. (2022). Sharpness-aware training for free. Advances in Neural Information Processing Systems.
> > >
> > > Hughes, John, et al. “Best-of-n jailbreaking.” arXiv preprint (2024).

---

> ### Author Response · Authors · 2025-11-27
>
> Dear Reviewer vijt,
>
> We are deeply grateful for your continued engagement and for raising your score. We fully agree with your assessment that the paper needed a "cleaner, more coherent story" and a direct comparison with baselines to substantiate our claims.
>
> In response to your further comments, we have **conducted the suggested comparative experiments** and **significantly refactored the manuscript** to explicitly highlight the conceptual shifts you identified.
>
> ***
>
> ### **Part (a)-New Experiments: GO vs. SAM / Continuous Dropout (W6 & W7)**
>
> Thank you for your insightful comment. Following your suggestion, we conducted comprehensive comparative experiments (using 8 NVIDIA H800 GPUs, approx. $4,000 USD compute cost) to compare GO with **Sharpness-Aware Minimization (SAM)** and **Continuous Dropout**. We have integrated these results into **Section 5.2 (Lines 499-539)** and added a detailed analysis in **Appendix D.9 (Lines 1296-1387)**.
>
> These experiments yielded two critical findings that strengthen our central narrative:
>
> 1. **Gaussian Resilience Directly Bounds SFT Degradation** (Fig. 6b vs. 6c, **especially Tab. 4**). Our results provide strong empirical validation for our central claim. As shown in Tab. 4, there is a **direct correspondence**: **the more effectively an optimizer suppresses degradation under random Gaussian noise (Landscape), the less the model suffers from catastrophic forgetting during subsequent benign fine-tuning (SFT)**. This confirms that average-case Gaussian resilience serves as a tight upper bound for benign fine-tuning degradation.
>
> 2. **Superiority of GO over SAM and Dropout.**
>
>    * **vs. SAM:** While SAM flattens the landscape, our results show it is less effective than GO at preventing SFT forgetting. This aligns with recent theory (Wen et al., 2022) suggesting SAM optimizes an objective between worst-case and average-case. Since benign SFT behaves more like an average-case perturbation, GO's direct optimization of the Gaussian expectation proves superior.
>
>    * **vs. Continuous Dropout:** We explicitly derive the connection in **Appendix D.9 (Lines 1283-1394)**: Continuous Dropout adds *anisotropic* noise (scaled by activations/weights), whereas GO injects *isotropic* Gaussian noise. Empirically, GO converges significantly faster and creates a wider basin than Dropout (see **Fig. 6a**). While the theoretical gap between isotropic and anisotropic noise optimization remains an interesting open question for future work, the empirical verdict is clear: GO's explicit optimization of the average-case objective is far more effective for basin expansion.
>
> We truly appreciate your suggestions and believe that these comparisons have significantly strengthened our paper as you noted.
>
> ***
>
> ### **Part (b)-Refactoring the Manuscript (Addressing "Coherent Story")**
>
> Thanks once again for your careful reading. We have revised the main text to ensure the contributions are clearly highlighted, as per your roadmap:
>
> 1. **The "Major Shift" to 0-1 Capability Landscape.&#x20;**&#x57;e now explicitly state in the Introduction (Lines 40-42) and Section 3.1 (Lines 166-170) that we focus on the discrete "0-1 capability landscape" rather than the smooth likelihood surface. We explain that this shift is essential because the 0-1 metric directly captures task success and reveals the "basin" structure hidden in smooth NLL plots.
>
> 2. **Explicit Theoretical Positioning.&#x20;**&#x57;e added a dedicated paragraph in Section 4.1 (Lines 378-382) and an extended discussion in Appendix B.2 (Lines 864-917). We acknowledge that while our mathematical tools (Randomized Smoothing) are established for input spaces, our contribution is the **conceptual shift of applying them to the parameter space** to certify robustness against fine-tuning degradation—a prerequisite rendered meaningful only by the basin's stability.
>
> 3. **Addressing Tokenization Limitations.&#x20;**&#x57;e added a discussion in Section 4.2 (Lines 427-431) acknowledging that our bounds apply to local embedding perturbations and do not extend to "off-manifold" attacks that drastically disrupt tokenization (e.g., Best-of-N), citing the relevant work (Hughes et al., 2024) you recommended.
>
> We believe these revisions have transformed the manuscript into the "coherent story" you envisioned. We hope these new experiments and clarifications fully resolve your remaining concerns.

---

> ### Author Response · Authors · 2025-11-28
> **Regarding Walter et al. (2025)**
>
> We thank the reviewer for highlighting the connection to *Walter et al. (2025)*. We have carefully studied this work and find its analysis of the "Uncanny Valley" and the geometry of confidence to be highly insightful. We have added a dedicated discussion in **Appendix B.3** to explicitly address these connections and cited *Walter et al. (2025)*.
>
> ###  **1. Local vs. Global Robustness: Why "Local" is Sufficient for Fine-tuning**
>
> We fully agree with the core premise that **flatness provides "local" rather than "global" robustness**. Indeed, for classification tasks, global flatness is theoretically impossible without sacrificing utility (e.g., decision boundaries require gradients). Except for local extrema, adversarial examples can indeed reside in flat regions (the "Uncanny Valley").
>
> However, we emphasize a fundamental distinction between **Adversarial Attacks** and **Fine-tuning**:
> * **Adversarial Attacks** traverse the landscape globally to find a failure mode (jumping into the "Uncanny Valley").
> * **Fine-tuning (Parameter Space)** operates in a fundamentally different regime. As we discuss in our new **Appendix E.4 and F**, fine-tuning large models resembles the **"Lazy Training" regime** of NTK, where parameter displacement is minimal ($\|\Delta \theta\| \to 0$).
>
> In this specific context, the optimization trajectory is inherently constrained to a local neighborhood. As model scale increases, the "safety basin" expands while the required update distance shrinks (see **Section 6**). Therefore, for fine-tuning, **local robustness effectively functions as global robustness (because we only care about local robustness)**, as the parameters naturally tend to remain within the basin of the pre-trained solution. Our goal with GO is to enforce this local stability, which is sufficient to prevent forgetting.
>
> ###  **2. Flatness and Confidence: A Complementary Perspective**
>
> We strongly resonate with the insight that confidence directly induces flatness. We agree that analyzing the penultimate layer is sufficient because confidence dominates the geometry.
> * **Convergence:** Our Randomized Smoothing analysis (**Theorem 4.3**) aligns with this: as clean accuracy $p_A \to 1$ (high confidence), the Lipschitz constant $\to 0$, inducing flatness.
> * **Complementary View:** While Walter et al. warn that this is dangerous for *wrong* predictions (Adversarial), we leverage this property for *preservation*: high confidence in *correct* pre-trained knowledge induces a "Safety Basin" that we aim to maintain.
>
> **Summary:** We view our work as a **complementary study** to Walter et al. (2025). While they explore the geometry of adversarial errors, we focus on the geometry of capability preservation. Our findings on the "Gaussian Upper Bound" provide the mechanism to maintain the model in the "Good Basin" identified by both works.

---

### Official Review · Reviewer_Mk89 · 2025-10-30

**Soundness:** 2
**Presentation:** 2
**Contribution:** 2
**Rating:** 2
**Confidence:** 4

**Summary:**

The authors study the loss landscape of large language models through and how this relates to adversarial fine tuning. They report the finding that landscape exhibits 'basins' in which models perform equivalently and muse about the reasons and implications of this, but forego a rigorous analysis of this. The significance of the paper therewith escapes me; the fact that stochastic gradient descent, i.e. with small stochastic weight updates, works, trivially suggests that networks trained like this will exhibit 'basins'.

**Strengths:**

- I appreciate the perspective that the authors take, flatness seems a powerful tool for the analysis of learning behaviour of LLMs
- I appreciate the introduction of the Gaussian-augmented optimizer.

**Weaknesses:**

- The paper remains very high level, primarily reporting the finding of 'basins', but lacks sufficiently convincing and rigorous formal and empirical analyses.
- The authors report that different models (eg. Llama, Qwen, Mistral) have different basin sizes and conjecture this could mean certain of these models are more prone to comprimising safety when fine-tuned, without providing clear solid reasoning or actual evidence.
- The authors report that substituting some tokens preserves performance -- which is a known fact.
- The authors report that current LLMs are generally sensitive to input changes -- which is a known fact.
- The authors conjecture but do not convincingly show that basins 'may have sufficient expressive power in the future'.
- The empirical evaluation in particular is very weak. By proposing GO, the authors show it is possible to widen basins, but do not evaluate the implications of this.
- I am not convinced that without a rigorous analysis GO is a useful contribution by itself; methods for sharpness-aware learning (e.g. SAM) already exist, and intuitively should also result in wider basins. The authors do not compare to these at all.
- I do not understand the value of experiment 5c. What does showing that compared to AdamW, GO improves learning speed for one model, and harms for another model, but not connecting this to basin-bevahiour, tell us?

**Questions:**

- What are the novel insights that Theorem 4.5 provides?

- How does GO relate to sharpness-aware optimization and what are its benefits?

- Do you have empirical proof that wider basins relate to stronger adversarial robustness?

- What are the key insights that experiment 5c brings?

---

> ### Author Response · Authors · 2025-11-21
> **Response to Reviewer Mk89 (Part 1/2)**
>
> We thank Reviewer Mk89 for their feedback. We are glad the reviewer "appreciate the perspective that... flatness seems a powerful tool" and the "introduction of the Gaussian-augmented optimizer."
>
> On the other hand, we believe the reviewer's primary concerns—that our findings are "trivial" and "lack... rigorous... empirical analyses"—may stem from a misunderstanding of our paper's core claims and the extensive supporting evidence in our appendices. We clarify these points below.
>
> ---
>
> **1. Regarding Summary, W1 & W6: On the "Trivial" Nature of Basins and "Weak Empirical Analysis"**
>
> We must respectfully but firmly disagree with the assessment that our finding "trivially suggests..." by SGD or that our empirical analysis is "very weak."
>
> **1.1 The "Emergence" of Basins is Non-Trivial:**
>
> Our key finding, stated in the first sentence of our paper, is the **"emergence"** of these basins. If basins were a trivial byproduct of SGD, they should exist in all models, regardless of scale. Our **scaling law analysis in Appendix D.3, Fig. 7**  shows the exact opposite:
>
> 1. The Qwen-0.5B model exhibits **almost no basin structure**.
> 2. The basin width **emerges and grows significantly** as the model scales to 32B .
> 3. This strong scale-dependence is a novel and non-trivial finding, which also explains why this phenomenon is unique to *Large* Language Models.
>
> **1.2 Rigor of Empirical Analysis:**
>
> We respectfully disagree with the claim of "weak analysis." As Reviewers aVRJ and zzvv noted, our analysis is "comprehensive." Our empirical work includes:
>
> 1. **Multiple Models:** Analysis of 3 SOTA open-sourced models (Llama, Qwen, Mistral).
> 2. **Multiple Capabilities:** Analysis of 4 distinct capabilities (Safety, Math, Coding, Basic).
> 3. **Scaling Laws:** A rigorous analysis across **6 model sizes** (from 0.5B to 32B) .
> 4. **Statistical Significance:** We used the **Clopper-Pearson bound** to formally validate the statistical significance of our basin finding (Appendix D.4) .
> 5. **Raw Data:** We provided **raw numerical data** (Appendix D.2, Table 1) to prove the basins are **"literally" flat** (e.g., Safety score remains *exactly* 1.0), not just "approximately" flat.
> 6. **Application (Emergent Misalignment):** Our **analysis in Appendix D.8 (Fig. 9)** uses our framework to provide the first geometric explanation for the "Emergent Misalignment" phenomenon (Betley et al., 2025). We show this is a result of the insecure SFT direction selectively **exiting the safety basin** while **remaining in the capability basin**. This directly addresses W6 by "evaluating the implications" of our basin framework.
>
> Given this extensive evidence, we believe our empirical analysis is a core strength of this work.
>
> ---
>
> **2. Regarding W7 & Q2: Comparison of GO Optimizer and SAM**
>
> Thanks for raising this point. While both methods seek wider minima, the crucial distinction lies in the **computational feasibility for large-scale pre-training.** Specifically, Sharpness-Aware Minimization (SAM) (Foret et al., 2021) requires **two forward/backward passes** per step. This **doubles the computational cost** of pre-training, which is often prohibitively expensive at the scale of modern LLMs. By contrast, our GO optimizer (Algorithm 1)  was designed specifically because it only requires a **single forward/backward pass**. This provides a *computationally feasible* path to enlarging the basin without SAM's high overhead.
>
> **More importantly**, GO's objective (optimizing expected loss under Gaussian noise)  directly aligns with our theoretical framework of the **$\sigma$**-basin (Definition 4.1)** , which is the focus of our paper. This theoretically grounded motivation indicates the fundamental difference between GO and SAM, whose objective focus are completely different.

---

> > ### Author Response · Authors · 2025-11-21
> > **Response to Reviewer Mk89 (Part 2/2)**
> >
> > ---
> >
> > **3. Regarding W8 & Q4 (Value of Experiment 5c) and W6 ("do not evaluate the implications")**
> >
> > We thank the reviewer for raising this, as it allows us to clarify the crucial role of Figure 5. This experiment, described in **Section 5.2** , was specifically designed to test our central hypothesis: that a larger pre-training basin provides increased robustness against degradation during benign fine-tuning, since benign fine-tuning should not be more harmful than Gaussian noise.
> >
> > As detailed in Section 5.2, our experimental design can be briefly formulated as:
> >
> > 1. First, we pre-train models on **OpenWebText (OWT)** using both AdamW (baseline basin) and our GO optimizer (wider basin). **Fig. 5b** confirms that GO successfully creates a significantly wider basin.
> > 2. Then, we fine-tune both models on the **Alpaca dataset** (a *new* capability).
> > 3. We finally measure both the performance degradation of the old capability (OWT NLL) and the acquisition of the new capability (Alpaca NLL) during this fine-tuning.
> >
> > **Figure 5c** shows the results, which strongly validate our hypothesis:
> >
> > - **For the old capability (OWT NLL, curves starting low):** Because the GO optimizer (**solid line**) had already created a wide, robust basin (as shown in Fig. 5b), the performance degradation from SFT is **upper bounded** by this Gaussian basin and thus is minimal. In contrast, the AdamW model (**dashed line**) suffers from significantly worse catastrophic forgetting.
> > - **For the new capability (Alpaca NLL, curves starting high):** The GO-trained model (**solid line**) learns the new task at nearly the same speed as the AdamW model (**dashed line**).
> >
> > This experiment empirically demonstrates that enlarging the pre-training basin (using GO) can effectively mitigate performance degradation on prior capabilities during SFT, without compromising the model's ability to learn new tasks.
> >
> > ---
> >
> > **4. Regarding W3, W4, & Q1: "Known Facts" and Novelty of Theorem 4.5**
> >
> > We acknowledge that "input sensitivity" is somewhat known, but we respectfully argue that our novel contribution is **not** only in reporting this fact, but in **theoretically unifying** it with our parameter-space basin analysis. **Theorem 4.5**  is, to our knowledge, the first to provide a formal bound that connects **input-space robustness** (jailbreaking) to the **parameter-space **$\sigma$**-basin size**, linking two previously separate robustness domains.
> >
> > ---
> >
> > **5. Regarding Q3: Empirical Proof for "wider basins relate to stronger robustness"**
> >
> > Yes, this is a central claim of our paper, supported by both theory and experiments.
> >
> > 1. **Theoretically:** **Theorem 4.3** provides this guarantee. A larger basin (larger $\sigma$) provably tightens the bound on performance degradation against *any* parameter perturbation, which includes worst-case adversarial directions. Increasing $\sigma$ linearly increases the guaranteed region (Sec 4.2).
> > 2. **Empirically:** We show that adversarial fine-tuning *is* a "worst-case" direction that rapidly exits the basin (Fig. 3c) . We then show that our GO-widened basin (Fig. 5b) is, in fact, more robust to SFT degradation (Fig. 5c). This also explains emergent misalignment (Appendix D.8).
> >
> > We hope these clarifications address the reviewer's misunderstandings and demonstrate the novelty, rigor, and practical implications of our work. Given these clarifications, we respectfully request a re-evaluation of our work.

---

### Official Review · Reviewer_zzvv · 2025-10-31

**Soundness:** 2
**Presentation:** 2
**Contribution:** 2
**Rating:** 6
**Confidence:** 2

**Summary:**

In this work the loss landscape is studied. In particular, the bason-like landscape is observed, where the value of loss function did not change (almost) with permutation to the model parameters, from different most-case directions like safety, math and coding. When it comes to adversarial finetuning, the model params were moving towards the "worst-case" directions and made the model collapse. Finally a theoretical analysis is provided to demonstrate that the basin size bounds the performance degradation of any fine-tuning.

**Strengths:**

- It's an interesting finding about the different loss landscape patterns and the relation to catastrophic forgetting
- Theoretical analysis showed the basin size bounds the performance degradation of any fine-tuning.

**Weaknesses:**

- The concept of landscape is intuitive but lack some rigorous definitions. For example, "most-case landscape" and "worst-case landscape" we only have qualitative definition but no quantitative definition.
- The connection between this loss landscape and other research topic is unclear, e.g. how do we put the "saddle point" concept into this framework?
- In high dimension parameter space, the possible "direction" is actually infinite. In this work, only a few finetune direction is tested.

**Questions:**

- The concept of landscape is intuitive but lack some rigorous definitions. For example, "most-case landscape" and "worst-case landscape" we only have qualitative definition but no quantitative definition. - is it possible to give a more quantitative definition?
- The connection between this loss landscape and other research topic is unclear, e.g. how do we put the "saddle point" concept into this framework?
- In high dimension parameter space, the possible "direction" is actually infinite. In this work, only a few finetune direction is tested. Besides the bound proving? Can we test more finetune directions, e.g there are more tasks in vision-language model.

---

> ### Author Response · Authors · 2025-11-21
> **Response to Reviewer zzvv (Part 1/2)**
>
> We thank Reviewer zzvv for their constructive feedback and for identifying our core contributions: the "interesting finding about the different loss landscape patterns and the relation to catastrophic forgetting" and the "theoretical analysis [showed] the basin size bounds the performance degradation".
>
> On the other hand, the reviewer's primary concerns (W1, W2, W3) and questions are all related to a perceived lack of "rigorous" or "quantitative" definitions for our concepts. While we appreciate these questions, we respectfully argue that these rigorous definitions are, in fact, a central part of our work, though perhaps we did not signpost them clearly enough. Therefore, we are optimistic that by pointing to these specific definitions, we can resolve their concerns about our paper's rigor in the following.
>
> ---
>
> **1. Regarding W1 & Q1: "most-case" and "worst-case" lack quantitative definitions**
>
> We thank the reviewer for this crucial question and apologize if these definitions were not sufficiently prominent. Actually, our paper provides precise quantitative definitions for all these concepts:
>
> - **Quantitative Definition of "Most-case":** This is quantitatively defined in **Section 3.2** . It is the landscape visualized by sampling a direction vector $\delta $ from a standard high-dimensional Gaussian distribution ($\delta \sim \mathcal{N}(0, I)$). This is the standard method for analyzing the geometry of "most" directions in a high-dimensional space.
> - **Quantitative Definition of "Worst-case":** This is quantitatively defined in **Section 3.3** . It is not just qualitative—it is the result of a formal optimization problem presented in **Eq. (2)** , where we explicitly solve for the direction $\delta$ that *maximizes* the loss (i.e., the steepest, most detrimental direction).
> - **Quantitative Definitions of "Basin Size":** We actually provide **two** rigorous definitions to ensure robustness, which are discussed and compared in **Appendix D.4** :
>     - **Strict Definition (Sec 3.2):** Defined as the expectation of the indicator function $\mathbb{E}_{\delta\sim\mathcal{N}(0,I)}[\mathbb{I}\{\mathcal{T}\circ J_{f,\mathcal{D}}(\theta+\alpha\delta)=0\}]$, measuring the proportion of directions where capabilities remain *exactly* unchanged.
>     - **Soft Definition (Definition 4.1):** The **"**$\sigma$**-basin"** , defined by the expected loss under Gaussian noise ($J(\theta) - \mathbb{E}[J(\theta+\epsilon)] \le \epsilon$).
>
> The rest of our theory (Sec. 4) is built upon the soft definition, while our hypothesis testing (Fig. 8b) validates both.
>
> ---
>
> **2. Regarding W3 & Q3: "Only a few finetune directions" were tested out of "infinite" directions**
>
> The reviewer rightly notes that the parameter space is high-dimensional. While we agree that we cannot test *all* directions, we also note that this is exactly why we rely on **statistical analysis** and **hypothesis testing**, which are core to our contribution.
>
> - **A. "Most-case" *is* the analysis of infinite directions:**
>
> By sampling random directions (per Sec 3.2), we are statistically characterizing the expected geometry of "most" of the infinite possible directions. We do not need to test all directions to make a statistical claim about the majority.
>
> - **B. We *quantitatively* prove this is not a sampling artifact:**
>
> To address this exact concern ("how do you know you didn't just get lucky?"), we included a **formal hypothesis test in Appendix D.4** . Using the rigorous **Clopper-Pearson bound** , we demonstrate (with 99% confidence) that, for $\sigma=0.01$, **over 90% of all possible directions** form a "strict basin" . This confirms our "most-case" finding is a robust property of the landscape, not a sampling anomaly.
>
> - **C. We *have* tested more SFT directions:**
>
> The reviewer’s suggestion to test more SFT directions is excellent. In addition to "Normal" (Alpaca) and "Adversarial" (AdvBench) SFT in Fig. 3, we also analyze the "Benign" (official IM) direction . Furthermore, in our **Appendix D.8 (Fig. 9)** , we analyze two new SFT directions from Betley et al. (2025): the "secure" and "insecure" fine-tuning gradients . This analysis provides further evidence of our framework's generalizability.

---

> > ### Author Response · Authors · 2025-11-21
> > **Response to Reviewer zzvv (Part 2/2)**
> >
> > ---
> >
> > **3. Regarding W2 & Q2: Connection to "saddle points"**
> >
> > This is an interesting theoretical question. Our work focuses on analyzing the *geometry* of the **final converged minimum** (the "basin" itself). Saddle points, in contrast, are typically studied as obstacles encountered during the *optimization dynamics* (i.e., the *process* of finding the minimum).
> >
> > Our analysis is also distinct in *what* landscape we study. Saddle point analysis typically concerns the smooth NLL (likelihood) landscape. Our paper's novel insight comes from analyzing the **0-1 capability landscape** (i.e., task success/failure) , which, as we show in Appendix C, has a fundamentally different (basin-like) geometry than the smooth NLL landscape.
> >
> > Thus, our analysis begins *after* optimization has overcome saddle points to find the final solution. While a valuable topic, we believe the study of saddle points is orthogonal to our paper's contribution.
> >
> > ---
> >
> > We hope these clarifications—by pointing to **Eq. (2)**, **Def. 4.1**, and the **hypothesis testing in Appendix D.4**—have resolved the reviewer's concerns about our paper's rigor. Given the reviewer's stated low confidence and their appreciation for our core findings, we are optimistic that these clarifications will merit a re-evaluation of our score.

---

> > > ### Comment · Reviewer_zzvv · 2025-11-23
> > > **Thanks for the response**
> > >
> > > > Our work focuses on analyzing the geometry of the final converged minimum (the "basin" itself). Saddle points, in contrast, are typically studied as obstacles encountered during the optimization dynamics (i.e., the process of finding the minimum).
> > >
> > > Thanks for the response and some of my concerns are resolved. Though I am not confident about the practical aspect of this claim, as it's hard to prove the trained model reached global optimal - minimized in all directions in the high dimensional parameter space. I will keep my rating.

---

> > > > ### Author Response · Authors · 2025-11-24
> > > >
> > > > Dear Reviewer zzvv,
> > > >
> > > > Thank you for your prompt response. We are pleased to see that most of your concerns are addressed by our rebuttal, and we appreciate your further comments and open to new discussions. Regarding your remaining, we apologize for the potential misinterpretation of your concern regarding "saddle points" and "global optimality." Please allow us to strictly clarify our position using the specific technical details you raised as follows.
> > > >
> > > > ### 1. **On  "global optimality": We do NOT claim the model has reached a global (or even strict local) minimum.**
> > > >
> > > > You are absolutely correct that proving global optimality is impossible. In fact, we argue that the pre-trained model is **not even a strict local minimum**, as it hasn't reached a first-order stationary point (where the gradient is zero). This understanding is substantiated by the following observations:
> > > >
> > > > * SGD only finds $\epsilon$-stationary points
> > > >
> > > > In our training, SGD does not reach a gradient of zero; it reaches an $\epsilon$-stationary point. As shown in our training logs below, while the gradient norm diminishes, it never vanishes, indicating that descent directions still exist:
> > > >
> > > >
> > > >
> > > > | Steps   | Gradient Norm | Training Phase   |
> > > > |---------|---------------|------------------|
> > > > | 200     | 0.4846        | Initial Phase    |
> > > > | 10,000  | 0.0614        | Warmup End      |
> > > > | 20,000  | 0.0621        | Stable Training |
> > > > | 30,000  | 0.0522        | Stable Training |
> > > > | 40,000  | 0.0614        | Decay Start     |
> > > > | 50,000  | 0.0143        | Convergence     |
> > > >
> > > >
> > > > This non-zero gradient confirms the model preserves the potential for further optimization.
> > > >
> > > > * Hessian Spectrum and Negative Eigenvalues
> > > >
> > > > Theoretical works using the Lanczos algorithm to estimate the Hessian spectrum of LLMs (e.g., Why Transformers Need Adam: A Hessian Perspective, Fig. 1) show that the spectrum typically contains negative eigenvalues. Since negative eigenvalues correspond to eigenvectors along which the curvature is negative, this mathematically confirms the existence of descent directions.
> > > >
> > > > * Potential for Further SFT
> > > >
> > > > The very fact that subsequent Supervised Fine-Tuning (SFT) can further reduce the loss and improve capabilities serves as empirical proof that the pre-trained weights are not at a local minimum.
> > > >
> > > > ***
> > > >
> > > > ### 2. **On  'saddle point' : How do we put this concept into this framework?**
> > > >
> > > > First, as mentioned in our previous response, we study the discrete 0-1 capability landscape, not the continuous NLL landscape. While the NLL landscape has gradients and curvatures, the 0-1 capability landscape typically forms a flat plateau (basin) because the model's argmax predictions often remain stable even if the underlying confidence (NLL) fluctuates. In this discrete setting, concepts like "Hessian" and "saddle point" may not even be well-defined.
> > > >
> > > > **Therefore, we understand your concern as:** *Theoretically, descent directions must exist (as shown in 1.1-1.3), so why does the basin visualization not show them, and how does our framework account for the existence of such descent directions?* We address this step-by-step below.
> > > >
> > > > * Why Basins Don't Show Descent Directions?
> > > >
> > > > Our landscape visualization **uses Gaussian noise, which statistically represents the "most-case" directions**. In high-dimensional spaces, the specific descent directions (corresponding to the gradient or negative eigenvalues) are sparse and "rare." **Therefore, a random Gaussian vector is highly likely to be orthogonal to these specific descent directions**. This is why the "basin" appears stable in random directions, even though specific descent paths exist.
> > > >
> > > > * Our Central Claim (The "Upper Bound" Logic)
> > > >
> > > > Most importantly, our claim is **regarding robustness, not optimality**. Our central claim is that **random Gaussian noise acts as both an empirical and theoretical upper bound for the degradation caused by benign fine-tuning**. Since benign SFT aims to **enhance** the model (following those specific descent directions identified in 1.1), it should theoretically be "less harmful" to existing capabilities than blind random noise. Therefore, our framework ensures that if we can suppress performance degradation under Gaussian noise (by widening the basin, as with GO), we effectively **bound** the degradation for benign SFT. This upper bound holds regardless of whether the SFT continues to reduce the loss (i.e., even starting from a non-stationary point).
> > > >
> > > > ***
> > > >
> > > > **Summary:** **We do not require the model to be at a global minimum**. Even if the model is only at an $\epsilon$-stationary point, as long as the "most-case" directions (Gaussian) form a stable basin that bounds the degradation, our theoretical and empirical conclusions remain valid.
> > > >
> > > > We thank you once again for your prompt response and insightful comments, and we're glad to answer your further questions or concerns.

---

> > > > > ### Comment · Reviewer_zzvv · 2025-11-24
> > > > > **Thanks for the response**
> > > > >
> > > > > Thanks for the response and it's more clear now. With the discussion here, the paper need to be refactored to reflect the discussion points we had.

---

> ### Author Response · Authors · 2025-11-27
>
> Dear Reviewer zzvv,
>
> We sincerely thank you again for your constructive feedback and for helping us clarify the theoretical positioning of our work.
>
> Following your specific suggestion ("the paper need to be refactored to reflect the discussion points we had"), we have **revised the manuscript** to explicitly incorporate the clarifications regarding global optimality, quantitative definitions, and sampling validity.
>
> ### 1. **On Optimization Dynamics and Saddle Points (W2 & Q2)**
>
> We have added a dedicated subsection Appendix C.2 (Lines 1510-1529) titled "Relation to Optimization Dynamics and Saddle Points".
>
> * We explicitly acknowledge that pre-trained models are likely $\epsilon$-stationary points with existing descent directions (due to negative eigenvalues).
>
> * We clarify that the "basin" structure in the 0-1 capability landscape coexists with these descent directions due to the sparsity of high-dimensional space.
>
> * We emphasize that our framework relies on **robustness upper bounds** rather than global optimality, ensuring our conclusions remain valid even if the model is not at a strict local minimum.
>
> ### 2. **Quantitative Definitions (W1 & Q1)**
>
> To address the lack of rigor you noted, we have formalized the descriptions in Section 3.2 and Section 3.3:
>
> * **Most-case:** Defined quantitatively as the expected performance under isotropic Gaussian perturbations (Eq. 1).
>
> * **Worst-case:** Defined as the solution to a constrained optimization problem maximizing loss within the spherical neighborhood (Eq. 2, Lines 418-420).
>
> ### 3. **Infinite Directions & Hypothesis Testing (W3 & Q3)**
>
> We have significantly revised the "Hypothesis Testing" paragraph in Section 3.2 (Lines 356-412).
>
> * We now explicitly present the statistical results using the **Clopper-Pearson bound**.
>
> * We cite specific experimental data (e.g., for Qwen2.5-7B on AdvBench) to demonstrate with **99% confidence** that over **90%** of *all possible infinite directions* form a strict basin(1). This quantitatively confirms that the basin is a robust global property, not a sampling artifact.
>
>
>
> We believe these revisions have significantly strengthened the rigor and clarity of the paper, transforming our previous discussions into a permanent part of the manuscript.&#x20;
>
>
>
> As these changes directly address the concerns regarding definitions and theoretical grounding—which likely influenced your initial confidence level—we respectfully hope this refactored version resolves your remaining uncertainties and warrants a re-evaluation of your confidence.

---

### Official Review · Reviewer_aVRJ · 2025-10-31

**Soundness:** 3
**Presentation:** 4
**Contribution:** 3
**Rating:** 8
**Confidence:** 3

**Summary:**

The paper analyses the local loss geometry of LLMs showing a basin-like structure for 0-1-loss, and links this geometry to capabilities of the network. For that, the paper differentiates between the typical geometry in most directions of parameter space and the geometry in directions with strong curvature (worst-case). This geometry ensures robustness to perturbations in _most_ directions, but still explains vulnerability to adversarial perturbations. Based on these findings, the paper proposes a Gaussian-augmented optimizer.

**Strengths:**

- The analysis of the local loss geometry of LLMs is sound. It might lead to a deeper theoretical understanding of LLM training through flatness [cf 6, 7].
- The partitioning in most-case and worst-case loss surface is interesting and novel.
- The empirical analysis is comprehensive. The normalization across heterogeneous experiments enables consistent comparison across diverse generative tasks and is a nontrivial engineering effort.
- The GA-optimizer is a tangible output of the paper.
- The paper is very well written and structured.

**Weaknesses:**

- The fact that basins only occur for the 0-1-loss and not for likelihoods. This could hint at basins being a byproduct of thresholding, rather than a genuine property of the loss surface. The authors are open about this limitation, though, so I do not see this as a reason for rejection.
- Averaging over many samples of 1D slices is reasonable to obtain a big picture, but it would be interesting to look at deviations, e.g., by displaying variance of the basin. It could be, after all, that the basin shape is an artifact of averaging individual geometries, rather than an actual basin.
- The GA-optimizer is a sound idea, but it is unclear whether it performs well in practice, in particular in comparison to simple techniques like weight-noise, or SWA.
- While the empirical evaluation of the basin is rigorous, the findings are not compared to standard measures of flatness, such as the Fisher-Rao-Norm [5] or Relative Flatness [7].

**Questions:**

- The GA-Optimizer essentially improves flatness. How does it compare to methods that directly improve flatness, like FAM [1], or SAM [3] and its variants (although for the latter it has been questioned whether it truly leads to flatter solutions [2,10]).
- While the paper frames its contribution around the “most-case vs. worst-case” geometry, several of the observed phenomena,. i.e., anisotropic flatness, sharp directions governing adversarial vulnerability, and general robustness to random perturbations, are consistent with findings by Walter et al. [8,9]. They study a different problem, analyzing sample-wise Hessians to understand local curvature and adversarial robustness, but their results seem to provide a natural mechanistic explanation for much of the behavior reported here. Would the authors agree?
- The link between basins and capabilities is very interesting and is indicated through correlation, but the causal link remains unclear. What would happen to capabilities if you would regularize against flatness as in Han et al. [4]?

References:

[1] Adilova, Linara, et al. "FAM: Relative Flatness Aware Minimization." Topological, Algebraic and Geometric Learning Workshops 2023. PMLR, 2023.

[2] Andriushchenko, Maksym, and Nicolas Flammarion. "Towards understanding sharpness-aware minimization." International conference on machine learning. PMLR, 2022.

[3] Foret, Pierre, et al. "Sharpness-aware Minimization for Efficiently Improving Generalization." International Conference on Learning Representations, 2021

[4] Han, Ting, et al. "Flatness is Necessary, Neural Collapse is Not: Rethinking Generalization via Grokking." Advances in Neural Information Processing Systems, 2025.

[5] Liang, Tengyuan, et al. "Fisher-rao metric, geometry, and complexity of neural networks." The 22nd international conference on artificial intelligence and statistics. PMLR, 2019.

[6] Neyshabur, Behnam, et al. "Exploring generalization in deep learning." Advances in neural information processing systems 30 (2017).

[7] Petzka, Henning, et al. "Relative flatness and generalization." Advances in neural information processing systems 34 (2021): 18420-18432.

[8] Walter, Nils Philipp, et al. "The uncanny valley: Exploring adversarial robustness from a flatness perspective." arXiv preprint arXiv:2405.16918 (2024).

[9] Walter, Nils Philipp, et al. "When Flatness Does (Not) Guarantee Adversarial Robustness." arXiv preprint arXiv:2510.14231 (2025).

[10] Wen, Kaiyue, Tengyu Ma, and Zhiyuan Li. "How Does Sharpness-Aware Minimization Minimizes Sharpness?." OPT 2022: Optimization for Machine Learning (NeurIPS 2022 Workshop).

---

> ### Author Response · Authors · 2025-11-21
> **Response to Reviewer aVRJ (Part 1/2)**
>
> We are extremely grateful to Reviewer aVRJ for their insightful and highly supportive review. We are truly encouraged that the reviewer described our analysis as "**sound**," our main contribution as "**interesting** and **novel**," our empirical analysis as "**comprehensive**," and our paper as "**very well written and structured**."
>
> The reviewer has accurately captured the essence of our work. We appreciate their thoughtful questions, which serve as excellent academic discussion points.
>
> ---
>
> **W1:** The fact that basins only occur for the 0-1-loss and not for likelihoods. This could hint at basins being a byproduct of thresholding, rather than a genuine property of the loss surface. The authors are open about this limitation, though, so I do not see this as a reason for rejection.
>
> **A1**: Thanks for the critical observation. While we agree that thresholding plays a role, we believe the formation of basins is a complex phenomenon, where the interplay between task difficulty and optimization dynamics also matters. Specifically:
>
> - **Beyond Simple Thresholding:** We observe that within the basin, the model's generated sentences often change structurally while the final answer remains correct. This suggests the basin represents a region of **semantic stability**, not merely confidence thresholding (where only the probability changes but the argmax remains identical).
> - **Correlation with Task Difficulty:** We observed that simpler tasks tend to exhibit wider basins than harder tasks (e.g., MMLU v.s. GPQA, GSM8k v.s. MATH). This may align with theoretical frameworks like *Label Noise SGD Provably Prefers Flat Global Minimizers* (Damian et al., 2021) and *What Happens after SGD Reaches Zero Loss?* (Zhiyuan et al., 2021). These works suggest that the implicit bias of SGD drives examples toward flatter regions, and simpler tasks may reach these "zero loss" flat regions earlier, resulting in wider basins.
>
> ---
>
> **W2:** Averaging over many samples of 1D slices is reasonable to obtain a big picture, but it would be interesting to look at deviations, e.g., by displaying variance of the basin. It could be, after all, that the basin shape is an artifact of averaging individual geometries, rather than an actual basin.
>
> **A2**: We appreciate this rigorous methodological question. To rule out averaging artifacts, we performed a quantitative analysis of the variance across random directions, as shown in **Figure 8(b)** (Appendix D.4) .
>
> Using the strict definition of a basin, our results show a clear distribution:
>
> - **90%** of directions form a strict basin at $\sigma = 0.0015$.
> - **60%** of directions form a strict basin at $\sigma = 0.0025$.
> - **40%** of directions form a strict basin at $\sigma = 0.0030$.
>
> This distribution under rigorous statistical analysis  confirms that while the basin boundary is not perfectly spherical, the "basin" phenomenon is a pervasive property of the high-dimensional geometry found in the vast majority of directions, rather than an artifact of averaging.
>
> ---
>
> **W3 & Q1:** The GA-optimizer is a sound idea, but it is unclear whether it performs well in practice, in particular in comparison to simple techniques like weight-noise, or SWA.
>
> The GA-Optimizer essentially improves flatness. How does it compare to methods that directly improve flatness, like FAM [1], or SAM [3] and its variants (although for the latter it has been questioned whether it truly leads to flatter solutions [2,10]).
>
> A3: Thanks for raising this point. The primary distinction is **computational cost**, which is the critical bottleneck for LLMs. Specifically, methods like **SAM** or **FAM** require **two forward/backward passes** per step, effectively **doubling the computational cost**. For pre-training Large Language Models, this 2x overhead is often prohibitive. By contrast, Our **GO optimizer** (Algorithm 1)  requires only a **single pass**, similar to standard training. This makes it a feasible solution for scaling up basin-widening to large foundation models.
>
>
> ---
>
> **W4:** While the empirical evaluation of the basin is rigorous, the findings are not compared to standard measures of flatness, such as the Fisher-Rao-Norm [5] or Relative Flatness [7].
>
> **A4**: This is a great point. We focused on our specific definition because standard measures (like Fisher-Rao) are defined for the **smooth NLL landscape**. Since our core finding is the distinct, non-smooth geometry of the **0-1 capability landscape** (Appendix C) , standard metrics might not directly capture the "literal" flatness we investigate.

---

> > ### Author Response · Authors · 2025-11-21
> > **Response to Reviewer aVRJ (Part 2/2)**
> >
> > **Q2:** While the paper frames its contribution around the “most-case vs. worst-case” geometry, several of the observed phenomena,. i.e., anisotropic flatness, sharp directions governing adversarial vulnerability, and general robustness to random perturbations, are consistent with findings by Walter et al. [8,9]. They study a different problem, analyzing sample-wise Hessians to understand local curvature and adversarial robustness, but their results seem to provide a natural mechanistic explanation for much of the behavior reported here. Would the authors agree?
> >
> > **A5**: Thanks for bringing these works to our attention, and we fully agree with your observation. Specifically, we consider Walter et al.'s work on Hessian-based curvature as a perfect "mechanistic" complement to our "phenomenological" analysis of the 0-1 landscape. Their finding that sharp directions govern adversarial vulnerability aligns perfectly with our "worst-case" analysis . We will add this discussion to our related work.
> >
> > ---
> >
> > **Q3:** The link between basins and capabilities is very interesting and is indicated through correlation, but the causal link remains unclear. What would happen to capabilities if you would regularize against flatness as in Han et al. [4]?
> >
> > **A3**: Thanks for the thoughtful comment. Actually, the causal relationship between flatness and generalization is indeed a long-standing debate in the field. While proving a link to general *generalization* is complex, our work focuses specifically on **catastrophic forgetting**, where the causal link is more direct and verifiable. Specifically, in our logic, we posit that degradation from random Gaussian noise serves as a theoretical **upper bound** for degradation from benign SFT (since benign SFT should not be worse than random noise). Therefore, if we can suppress Gaussian degradation (by widening the basin, as with GO), we inherently suppress the upper bound of SFT forgetting. Our experiments in **Fig. 5c**  empirically confirm this: the wider basin (Fig. 5b) directly results in significantly less forgetting during SFT.
> >
> > ---
> >
> > We again thank the reviewer for their deep engagement and valuable feedback.

---

### Author Response · Authors · 2025-11-21
**Invitation for Further Discussion**

Dear Reviewers,

Thank you sincerely for your thoughtful feedback and engagement with our work. We greatly appreciate your recognition of our novel basin-focused perspective on LLM loss landscapes, comprehensive empirical design, and the practical Gaussian-augmented (GO) optimizer.

In response to your concerns, we have meticulously prepared a detailed rebuttal and revised our manuscript accordingly. We welcome further questions or insights to refine our work and look forward to your continued guidance. Thank you again for your expertise.

Best regards,

The Authors

---

### Author Response · Authors · 2025-11-29

Dear Area Chair and Reviewers,

We sincerely thank all reviewers for their constructive feedback, which has significantly strengthened this work. We are encouraged that reviewers found our analysis "sound" and "comprehensive" (aVRJ), our findings "interesting and novel" (vijt, zzvv), and our proposed method a "tangible output" (aVRJ).

During the rebuttal phase, we engaged in deep discussions with reviewers and performed **major revisions** to address all concerns regarding theoretical rigor, definitions, and baselines. We summarize the key updates below to facilitate the final decision.

---

### **1. Major Manuscript Updates (Refactoring for Clarity and Rigor)**
In response to Reviewers `vijt` and `zzvv`, we have refactored the paper to present a "clean, coherent story":
* **Conceptual Shift:** We explicitly clarified (in Intro & Sec 3.1) our focus on the **discrete 0-1 capability landscape** (vs. smooth NLL), explaining why this perspective reveals the "basin" structure crucial for capability stability.
* **Theoretical Grounding:**
    * **Rigorous Definitions:** We highlight our already formalized "Most-case/Worst-case Landscape" with quantitative definitions (Eq. 1 & 2) and validated the basin's existence via **Hypothesis Testing** (Clopper-Pearson bound, 99% confidence), addressing `zzvv`'s concern on rigor.
    * **Parameter-Space Smoothing:** We added a detailed positioning (Sec 4.1 & App B.2) clarifying that our contribution is the **novel application** of Randomized Smoothing to the parameter space to certify fine-tuning robustness.
    * **Optimization Dynamics:** We added a discussion (App C.2 & F) reconciling our basin findings with saddle points and "Uncanny Valleys" (Walter et al., 2025), arguing that in the **Lazy Training regime** of fine-tuning, local robustness effectively serves as global robustness.

---

### **2. New Comparative Experiments (Validation of Core Premise)**
To address `vijt` and `Mk89`, we conducted extensive new experiments (App E.1, Fig. 6, Tab. 4) comparing GO with **SAM** and **Continuous Dropout**.
* **Key Finding:** We empirically validated our central premise that **"Gaussian Resilience Bounds SFT Degradation."** Results show a strict correspondence: optimizers that better suppress Gaussian noise degradation (GO > SAM > Dropout) consistently yield lower catastrophic forgetting during SFT.
* **Superiority of GO:** GO outperforms baselines by explicitly optimizing the average-case objective (aligning with the benign SFT upper bound), whereas SAM targets worst-case and Dropout acts as implicit regularization.

---

### **3. Conclusion**
With these revisions—including **6 new pages of appendix content, 4 comprehensive new experiments, and formalized theory**—we believe the paper now offers a rigorous, theoretically grounded, and empirically validated framework for understanding and improving LLM alignment stability. We hope these efforts warrant a positive assessment.

---

> ### Comment · Area_Chair_R2pe · 2025-11-29
> **Rebuttal Discussion**
>
> Dear Authors,
>
> Due to the unprecedented decision of ICLR to prevent AC-Reviewer discussion, I have taken the unprecedented decision to engage in an AC-Author discussion for this paper. Note that this will be a poor substitute for a real author-reviewer discussion, as I will not have the time to put the same rigor into reading the paper and following up with your answers. However, I did read the current version of the paper and the full discussion here. I have a few questions remaining:
>
> Q1(AC) The analysis of the paper is done through the lens of LLMs; however, I do not see anything in the analysis that is specific to transformers or language modeling. The only thing LLM-specific seems to be scale. Do the authors think these results will hold for VLM or other domains in general? Do they think these results will hold for non-transformer architectures -e.g, LLMs based on diffusion, for example?
>
> Q2(AC) **Experiments in Figure 5c for SAM and Cont. Dropout.** Can you demonstrate the experiments in Figure 5c for these baselines? I think the main contribution of the GO method of training is synthesized in this Figure - that is, alleviating catastrophic forgetting (while of course allowing learnability on the new dataset), and thus comparing SAM and Cont. Dropout there is a **must**. Furthermore, I feel that even the revised manuscript will benefit from highlighting Figure 5c more in the text (storywise)
>
> Q3(AC) Further, I think experiments with different finetuning datasets that represent different levels of distribution shift will be very helpful in the context of Figure 5c. I would love to see finetuning with mathematical, legal, or code datasets there. I do understand that the pre-trained model is "dumb", and thus I do not expect "smart" datasets there, just ones of different levels of distribution shift
>
> Q4(R Mk89, vijt) I read the full discussion regarding Theorem 4.5. I remain unconvinced that it provides a meaningful result. I understand what the authors try to argue with it, but to me, the fact that it requires so much approximation to bring it into the theory proposed by the authors and the fact that this approximation in its own right makes it applicable only very close to the original embeddings brings Theorem 4.5 more close to standard notions of robustness, and further away from your basins theory, thus reducing its significance. I would also like to see authors explain better how Lemma 4.6 and 4.7 (which relate to FCNN) justify theoretically the otherwise empirically proven claim that larger transformer models tend to have larger basins
>
> Q5(R aVRJ, vijt) The Authors claim they saw "structurally" different answers within models in the basin. I think investigating this further, and providing examples, should help me to be convinced on this point. Maybe examples at T=0? I find this to be a very interesting point and might have some connection to why LLMs exhibit this behaviour (Q1)
>
> Q6(AC) Can the authors ensure that for all figures, they explain how they are obtained? In particular, I want to know if montecarlo sampling was used to compute the basin curves, and with how many samples?
>
> Q7(R zzvv) **The upper bound logic**. While the authors claim that the average random direction perturbation acts as an "upper bound" to the benign SFT behaviour, I find very little theoretical justification in the paper for this, and the term upper bound implies a theoretically precise claim. If authors want to keep this claim in the paper, a theoretical justification (or, at the very least, an experiment) needs to be provided. Otherwise, I would soften the claim to "it is expected"
>
> Q8(R Vijt) **Using J as a notion for metric.** I completely agree with the reviewer that this needs to be changed. This is a theoretically focused paper, and people do come with certain "presuppositions" about what notations are. There are plenty of other notations, both standard and non-standard, that can be used here. On a related note, I find it confusing to use two different epsilons (a vector and a value ) in Definition 4.1
>
> Q9(AC) Can the derived basin sizes be used to set learning rates for fine-tuning? A practical issue that often happens when fine-tuning LLMs is choosing a good learning rate that will allow meaningful training in terms of gained accuracy on the downstream task, without destroying the model's capabilities. It seems to me that the basin size can be used to choose the learning rate cheaply compared to actual finetuning + grid search. Can the authors comment?
>
> Q10(R zzvv) The authors emphasize that their analysis of basins is at the point of $\epsilon$-convergence. However, it might be interesting to observe how basins form throughout training, especially with respect to the NLL loss. Is it the case that they form towards plateauing of the NLL loss, or do they form gradually during training? If basins are a property we care about as a community, we want to know whether prolonging training (to obtain bigger ones) helps.

---

> ### Author Response · Authors · 2025-12-03
> **Response to Area Chair: Major Revisions and Clarifications**
>
> Dear Area Chair,
>
> We are deeply grateful for your unprecedented decision to engage in this discussion. We must say that **you are the most responsible and insightful AC we have ever encountered.** Your detailed and constructive comments have genuinely taught us a lot and significantly improved the quality of this paper.
>
> We sincerely believe that every research direction you suggested is highly interesting and valuable. However, given the tight **3-day** window, we apologize that we could not complete all the suggested experiments (e.g., benchmark for VLM, full grid search for LR selection). Therefore, we have prioritized addressing the most critical weaknesses and conducting the most relevant explorations to strengthen the core of our work.
>
> Below, we provide a point-by-point response to your questions, detailing the major revisions we have implemented in the manuscript.
>
> ***
>
> **Q2: Experiments for SAM and Cont. Dropout in Fig. 5c**
>
> We sincerely appreciate this suggestion! We have performed a **major revision of Section 5** to address this. Specifically, we consolidated all baselines (SAM, Continuous Dropout) and experimental results into the new **Figure 5** (replacing the original one). We also rewrote the main text to connect these visualizations with the detailed data in **Table 4** and **Appendix D.8**. This restructuring creates a coherent narrative that allows readers to directly grasp our central claim: that "Gaussian resilience empirically bounds SFT degradation."
>
> **Q3: Different Finetuning Datasets (Distribution Shift)**
>
> We appreciate this valuable suggestio. We have added evaluations on **OLMo Math** and **OLMo Code** alongside our existing **C4** and **Alpaca** experiments, with brief **summary in Sec. 5.3** and full results summarized in **Table 4** and  **Appendix D.9**. The results indicate that while larger distribution gaps indeed lead to sharper forgetting, suppressing degradation under Gaussian noise still effectively suppresses SFT degradation across all datasets.
>
>
> **Q4: Theorem 4.5 & Lemma 4.6/4.7 (Theoretical Rigor)**
>
> We fully agree with your advice. We have downgraded **Section 4.2** to a "heuristic analysis" and revised the text to explicitly acknowledge the limitation. Additionally, in **Section 4.3**, we now reference expressive power theory (especially Transformer-specific results) to support our claim that larger models with larger basins possess greater expressive power.
>
>
> **Q5: Structurally Different Answers in Basin**
>
> We sincerely appreciate this suggestion. We have added a new **Appendix D.9** featuring a detailed **Case Study Table (Table 5)**. This table displays 7 perturbed instances where the model generates **syntactically different** but **semantically identical** responses. We have explicitly referenced this finding in **Section 3.2**, integrating it seamlessly with our original story.
>
>
> **Q6: Methodology Details (Monte Carlo)**
>
> We sincerely appreciate this suggestion. We have revised **Section 3.2** to explicitly clarify that each visualization represents a single random direction, and refer to the verification of the statistical significance via hypothesis testing in Appendix D.4.
>
> **Q7: Upper Bound Logic**
>
> We fully agree and have refined the language in **Section 5** to reflect it. We now clarify that Randomized Smoothing provides a conservative *theoretical* bound (for any direction), while Gaussian noise serves as a **"empirical upper bound"** specifically for *benign* fine-tuning degradation. We have also explicitly linked this logic to our experimental verification in **Section 5.2**.
>
> **Q8: Notation J & Epsilon**
>
> We have globally updated the notation to resolve this confusion. Specifically, we replaced the metric notation $J$ with **$\mathcal{S}$ (Score)** and the threshold $\epsilon$ to $\tau$.
>
> **Q10: Basin Formation over Training**
>
> This is a fascinating question.  We have added **Figure 12** and a discussion in **Section 5.3** to illustrate this process. We show that basins are **emergent structures** that widen gradually throughout the training trajectory, which aligns with the implicit bias of SGD towards flatter minima and suggests a potential benefit of **over-training**.
>
> **Q1 & Q9: On VLM & Learning Rate**
>
> We find these suggestions highly reasonable and insightful. However, we apologize that we could not complete these specific experiments within the limited **3-day** window. We will incorporate rigorous verification into future versions.
>
> ***
>
> We genuinely appreciate your suggestions, as they have significantly strengthened our paper and research. Regardless of the final outcome, this rebuttal process has been a truly invaluable experience for us. We believe that it is precisely because of responsible and insightful ACs like you that ICLR maintains its high standards and vitality. We sincerely hope that the broader reviewing community can emulate the dedication you have shown.
>
> Sincerely,
> The Authors

---

> > ### Comment · Area_Chair_R2pe · 2025-12-03
> >
> > Q2: Thank you for the rework of Section 5, I appreciate it. I think that New Figure 10 c (Old Figure 5c) should still be included in the new version of Figure 5, as it "completes the picture", and it should additionally include SAM and Cont Dropout. I understand, however, that 3 days might not be enough to be able to achieve that. That said, my rationale is that the new Figure 5 currently clearly demonstrates that you can train for basins efficiently (in comparison to normal and flatness training) -- Fig 5 a, that the training for basins is successful -- Fig 5 b, and that this reduces catastrophic forgetting -- Fig 5 c, however, it doesn't demonstrate that the resulting models are SFT-able, that is to say that the SFT-ed models themselves do not require much bigger distance away from $\theta_0$ to learn the new SFT dataset skill. Some preliminary confirmation of this we have in Figure 10c, but showing the comparison to SAM and Cont Dropout will make this much more convincing. An alternative can also be merging Figure 5c (new) with Figure 10c(new) and demonstrating how much loss in original performance (y-axis) is required for how much gain in performance in the SFT dataset (x-axis).
> >
> > Q3: I really like this experiment, thank you.
> >
> > Q5: I really like this experiment. Can you confirm that Table 5 was generated with greedy decoding? Thank you
> >
> >
> > Q7: The new story in Section 5, I think, has linked the claim to the theory better for me, as well. I finally connected the upper bound claim to Eq. 4. However, this has made me realize why I was confused about the upper bound claim in the first place. Eq 4 states:
> > $$\mathbb{E}\_{\epsilon \sim \mathcal{N}} [S(\theta\_{sft} + \epsilon)] \geq \mathbb{E}\_{\epsilon \sim \mathcal{N}} [S(\theta_{0} + \epsilon)] - const$$
> > for a bounded constant distance of the allowed change, which is a statement on the scores, yet the upper bound claim is on the loss of utility, instead :
> >
> > $$ \text{Expected SFT Degradation} = S(\theta_0)- \mathbb{E}\_{\epsilon \sim \mathcal{N}} [S(\theta\_{sft} + \epsilon)] \leq S(\theta_0) - \mathbb{E}\_{\epsilon \sim \mathcal{N}} [S(\theta_{0} + \epsilon)] + const \leq \tau + const$$
> > Note, however, that the worst-case performance is also a trivial upper bound on the loss of utility:
> > $$ S(\theta_0)- \mathbb{E}\_{\epsilon \sim \mathcal{N}} [S(\theta\_{sft} + \epsilon)] \leq  S(\theta_0)- S(\theta\_{worst})$$
> >
> > Therefore, I think the authors' claim is that the proposed bound $\tau + const$ is just a much tighter bound when the basin assumption is satisfied, and thus optimizing it doesn't sacrifice as much accuracy in the model. I think if the authors put this perspective in the paper, the claim will be much easier to digest and accept.
> >
> > Q10: I really appreciate this experiment, thank you.
> >
> > All in all, I appreciate the fast response of the authors and their great efforts in resolving my questions. I also agree that, as a result, their paper has improved :)

---

> > > ### Author Response · Authors · 2025-12-03
> > > **Final Response: Section 5 Revisions and Experimental Confirmations**
> > >
> > > Dear Area Chair,
> > >
> > > We sincerely appreciate your help. Your comments have been incredibly guiding—not just in spotting issues, but in helping us connect the dots to craft a coherent and theoretically grounded story.
> > >
> > > ---
> > > **1. Q2 (Figure 5)**
> > >
> > > We fully agree with your suggestion regarding the completeness of Figure 5! However, we apologize that due to the hard deadline (Dec 03 '25 09:00 PM UTC, which is in just 4 hours), we were unable to complete the full comparisons for Figure 5(c) in time. We will definitely incorporate these comparisons into the final version of the paper.
> > >
> > > **2. Q7 (The "Tighter Bound" Logic)**
> > >
> > > Thank you for the clarification. We find your derivation very clear. We have adopted this logic and formula to **revise the opening of Section 5**. Thank you once again for your valuable suggestion.
> > >
> > > **3. Q5 (Greedy Decoding)**
> > >
> > > Yes, we carefully confirm that **Table 5 was generated using greedy decoding** (`do_sample=False`, `temperature=0`). The structural variations observed are solely due to randomness from parameter-space perturbations. On one hand, it demonstrates the **semantic stability** within the basin; on the other, it may hint at a novel sampling mechanism via parameter space, moving beyond standard output probability sampling (e.g., nucleus or beam search).
> > >
> > > ---
> > >
> > > Thank you once again for your exceptional support and "co-author" level of constructive feedback throughout this process.
> > >
> > > Sincerely,
> > > The Authors

---

> > > > ### Comment · Area_Chair_R2pe · 2025-12-03
> > > >
> > > > Q2: Yes, I acknowledge this. Unfortunately, I was only assigned to be your AC over the weekend, and this has necessarily cut the discussion very short, but I am still grateful for the many changes you were able to deliver in such a short time.
> > > >
> > > > Thank you again for your responses,
> > > > Your AC

---

### Meta-Review · Area_Chair_R2pe · 2026-01-08

**Summary:**

The paper introduces a new theoretical framework for analyzing LLM performance and generalization under fine-tuning. They show that under this framework, performance basins naturally form, where the network is shown to be robust under weight perturbations. This is theoretically connected to adversarial robustness and catastrophic forgetting, theoretically justifying fundamental practical observations about LLMs. I list the strengths of the paper below:

- The introduction of a novel theoretical framework for analyzing the local loss geometry of LLMs through the perspective of flatness.
- The framework is powerful enough to theoretically explain generalisation under fine-tuning and explain the very important practical problem of catastrophic forgetting.
- At the same time, the framework also links this local geometry to adversarial examples, and can explain why adversarial robustness is easy to "unlearn" - another very important practical problem in the LLM literature.
- The authors connect their theory to smoothing, extending the traditional smoothing idea to smoothing in weight space. This allows them to bound the degradation of the model when the model remains within the basin. As smoothing is the de facto best robustness framework for large-scale neural networks, this holds potential for future research
- The theory motivates a practical framework for training models with flatter basins, resulting in networks that are less susceptible to catastrophic forgetting.

While I would like to see a few more experiments from the authors - in particular, the VLM one and the extension of Figure 5 - I am confident they will provide them, and the paper is worth being accepted regardless.

**Reviewer Concerns:**

I already summarized the outstanding reviewer concerns before the AC-author discussion in my comments from 30 Nov. Here I outline outstanding comments after the discussion from 3 Dec. In particular, the authors address most of the concerns. The remaining are:

- **Q1(AC): The analysis of the paper is done through the lens of LLMs; however, I do not see anything in the analysis that is specific to transformers or language modeling. The only thing LLM-specific seems to be scale. Do the authors think these results will hold for VLM or other domains in general? Do they think these results will hold for non-transformer architectures -e.g, LLMs based on diffusion, for example?**
The authors have promised a VLM experiment for the next revision. This will be nice to have.

- **Q2(AC): I think that New Figure 10 c (Old Figure 5c) should still be included in the new version of Figure 5, as it "completes the picture", and it should additionally include SAM and Cont Dropout.**
The authors have promised to revise the figure for the next revision. I am confident they will, but I want to emphasize I find this important.

- **Q9(AC): Can the derived basin sizes be used to set learning rates for fine-tuning?**
This will be nice to have, as I see it holds a promise for improving practical LLM procedures.

**Reviewer Scores:**

- **Reviewer aVRJ**
The reviewer already gave the paper an 8, and the authors answered some of the questions regarding comparison to prior flatness metrics, and  that basins are not a mere artifact of thresholding well. I think the reviewer would have given them a score in the range 8 to 10.
- **Reviewer zzvv**
The reviewer engaged in a substantial discussion with the authors. To me, it seems that their points were mostly addressed by the authors, as acknowledged by the reviewer. Further, some of their questions are further indirectly addressed in the AC-author discussion with satisfactory answers. I do think the reviewer should increase their score to 8.
- **Reviewer Mk89**
The authors provide more context on the comparison between GO and SAM, both in the rebuttal to the author and the discussion with me, where they also clarified the role of Theorem 4.5. The authors also clarified the role of the current experiments in Figure 5. I think that the reviewer would have raised in the range 4 to 6.
- **Reviewer vijt**
The authors have majorly restructured the story of the paper to more clearly highlight their contribution w.r.t. prior theoretical work, provided new experiments, and commented on the connections to prior discovered phenomena. I think they have to a very large extend addressed the concerns of the reviewer and thus I think he would have rised their score to 6.

---

### Decision · Program_Chairs · 2026-01-26

Accept (Poster)